# emIAM v1.0: an emulator for Integrated Assessment Models using marginal abatement cost curves

Weiwei Xiong[1,2], Katsumasa Tanaka[1,3], Philippe Ciais[1], Daniel J. A. Johansson[4], Mariliis Lehtveer[4]

[1] Laboratoire des Sciences du Climat et de l'Environnement (LSCE), IPSL, CEA/CNRS/UVSQ, Université Paris-Saclay, Gif-sur-Yvette, 91191, France

[2] School of Economics and Management, China University of Geosciences, Wuhan, 430074, China

[3] Earth System Division, National Institute for Environmental Studies (NIES), Tsukuba, 305-8506, Japan

[4] Division of Physical Resource Theory, Department of Space, Earth, and Environment, Chalmers University of Technology, Gothenburg, 412 96, Sweden

*Correspondence to*: Weiwei Xiong (weiwei.xiong@lsce.ipsl.fr) and Katsumasa Tanaka (katsumasa.tanaka@lsce.ipsl.fr)

**Abstract**. We developed an emulator for Integrated Assessment Models (emIAM) based on a marginal abatement cost (MAC) curve approach. Drawing on the output of IAMs in the ENGAGE Scenario Explorer and the GET model, we derived an extensive array of MAC curves, encompassing ten IAMs, global and ten regions, three gases $CO_2$, $CH_4$, and $N_2O$, eight portfolios of available mitigation technologies, and two emission sources. We tested the performance of emIAM by coupling it with a simple climate model ACC2. Our analysis showed that the optimizing climate-economy model ACC2-emIAM adequately reproduced a majority of the original IAM emission outcomes under similar conditions. This can facilitate systematic exploration of IAMs with small computational resources. emIAM holds the potential to enhance the capabilities of simple climate models as a tool for calculating cost-effective pathways directly aligned with temperature targets.

## 1 Introduction

Integrated Assessment Models (IAMs) combine economic, energy, and sometimes also land-use modeling approaches and are commonly used to evaluate least-cost mitigation scenarios (Weyant, 2017). A variety of IAMs were integrated under common protocols in modeling intercomparison projects (MIPs) (O'Neill et al., 2016; Tebaldi et al., 2021) and provided input to the series of the Intergovernmental Panel on Climate Change (IPCC) Assessment Reports. However, simulating computationally expensive IAMs developed and maintained at different research institutions around the world requires large coordination efforts. Therefore, here we propose a new methodological framework to i) emulate the behavior of IAMs (i.e., emission abatement for a given carbon price) through MAC curves and then ii) reproduce the behavior of IAMs by using the MAC curves coupled with a simple climate model. We show that the MAC curves can be systematically applied to reproduce the behavior of IAMs as an emulator for IAMs (emIAM), paving a way to generate multi-IAM scenarios more easily than before, with small computational resources.

In the context of climate change mitigation, a MAC generally represents the incremental cost of reducing an additional

unit of emissions; a MAC curve illustrates these costs as the level of emission reductions increases relative to the baseline. There is a burgeoning literature on MAC curves (Jiang et al., 2020) that can broadly fall into two categories (Kesicki and Ekins, 2012): i) data-based MAC curves (bottom-up) and ii) model-based MAC curves (top-down). First, a data-based MAC curve provides a relationship between the emission abatement potential of each mitigation measure considered and the associated marginal costs, in the order of low- to high-cost measures based on individual data. A prominent example of such data-based MAC curves is McKinsey & Company (2009). Second, a model-based MAC curve provides a relationship between the amount of emission abatement and the system-wide marginal costs based on simulation results of a model (e.g., an energy system model and a computational general equilibrium (CGE) model) perturbed under different carbon prices or carbon budgets. Our work takes the second approach, building on previous studies (Nordhaus, 1991; Ellerman and Decaux, 1998; van Vuuren et al., 2004; Johansson et al., 2006; Klepper and Peterson, 2006; Johansson, 2011; Morris et al., 2012; Wagner et al., 2012; Tanaka et al., 2013; Su et al., 2017; Tanaka and O'Neill, 2018; Yue et al., 2020; Tanaka et al., 2021; Bossy et al., 2024; Su et al., 2024). While data-based MAC curves tend to be rich in the representation of technological details, they do not consider system-wide interactions that are captured by model-based MAC curves. Model-based MAC curves reflect such interactions, however, without much explicit technological detail. Advantages and disadvantages of MAC curves of different categories are discussed elsewhere (Vermont and De Cara, 2010; Kesicki and Strachan, 2011; Huang et al., 2016).

In this study, we derive a large set of MAC curves from the simulation results of IAMs (see Figure 1 and Section 3), couple them with a simple climate model as an emulator (emIAM), and validate the simulation results with the original IAM results under similar conditions. Namely, we look up the ENGAGE Scenario Explorer hosted at IIASA, Austria (https://data.ene.iiasa.ac.at/engage), a publicly available database from the EU Horizon 2020 ENGAGE project (Drouet et al., 2021; Riahi et al., 2021), and extract total anthropogenic $CO_2$, $CH_4$, and $N_2O$ emission pathways until 2100 from nine IAMs under a range of carbon budget constraints. For each IAM, we derive a set of $CO_2$, $CH_4$, and $N_2O$ MAC curves as a function of the respective emission reduction in percentage relative to the baseline at the global and regional (ten regions) levels. We then integrate the sets of MAC curves (i.e., emIAM) into a simple climate model called the Aggregated Carbon Cycle, Atmospheric Chemistry, and Climate (ACC2) model (Tanaka et al., 2007; Tanaka and O'Neill, 2018; Xiong et al., 2022). ACC2-emIAM works as a hard-linked optimizing climate-economy model that can derive an emission pathway to achieve a given climate target or carbon budget at the lowest cost. We validate to what extent the emission pathway derived from ACC2-emIAM under a given carbon budget or a temperature target can reproduce the corresponding pathway from the original IAM in the ENGAGE Scenario Explorer.

We further apply the emIAM approach to the GET model (Lehtveer et al., 2019), an IAM that did not take part in the ENGAGE project. We can directly simulate GET to derive MAC curves under different model configurations, which complements the existing data from IAMs simulated under single configurations for the ENGAGE project. We obtain global energy-related $CO_2$ emission pathways under a range of carbon price projections, but with several different portfolios of

available mitigation technologies (e.g., differentiated Carbon Capture and Storage (CCS) capacity). We then derive a MAC curve for each technology portfolio. Although MAC curves concern only the total emission abatement without distinguishing individual mitigation measures, this approach allows us to explore the role of a particular mitigation measure by comparing MAC curves with and without that mitigation measure. Note that all IAMs emulated in this study take a cost-effectiveness approach, in which the least-cost emission pathways to achieve a climate-related target are calculated in terms of the cost of mitigation without considering climate damage and adaptation.

To our knowledge, this study is one of the first attempts to apply the MAC curve approach extensively for developing an IAM emulator: we consider ten IAMs, global and ten regions, three gases (i.e., $CO_2$, $CH_4$, and $N_2O$), eight technology portfolios, and two broad sources (i.e., total anthropogenic and energy-related emissions). We demonstrate the applicability of emIAM by implementing it to ACC2, but emIAM can also be used with other simple climate models (Joos et al., 2013; Nicholls et al., 2020). Thus, emIAM allows ACC2 and potentially other simple climate models to reproduce approximately global and regional cost-effective emission pathways from multiple IAMs under a range of given carbon budgets or temperature targets. In recent years, there have been efforts to develop emulators of Earth System Models (ESMs) in CMIP6 and the use of ESM emulators was exploited in the IPCC Sixth Assessment Report (AR6) (Leach et al., 2021; Tsutsui, 2022); however, no emulator has yet been developed for IAMs contributing to the IPCC.

In this paper, following the common definitions of terminologies found in the literature (National Research Council, 2012; Mulugeta et al., 2018), we use "emulate" to indicate a process of identifying a reduced-complexity model (i.e., a MAC curve) that approximates the behavior of a complex model (i.e., an IAM), "reproduce" to refer to a process of generating an output (i.e., an emission pathway) from an emulator with the same input and constraints given to an IAM (i.e., a cumulative carbon budget or end-of-century temperature, for example), and "validate" to indicate a process of investigating the extent to which an emulator reproduces an intended outcome in comparison to the corresponding original outcome from an IAM. Regarding the units, we use the original units of each model (i.e., US$2010 and $tCO_2$-eq with 100-year Global Warming Potential (GWP100) for all IAMs emulated here) to keep the comparability with underlying data, unless noted otherwise.

The remainder of the manuscript consists of five sections: Section 2 introduces the IAMs under consideration and their experiments used to derive MAC curves. Section 3 describes the methodology to derive MAC curves and presents the MAC curves that are derived (i.e., emIAM). Section 4 shows the validation results for ACC2-emIAM. Section 5 discusses a specific aspect of our emulation approach: the time-independency and the time-dependency of MAC curves. The paper is concluded in Section 6 with general remarks on the utility of emIAM. Given the substantial amount of MAC curves generated in our analysis, results are presented only selectively in the main body of the paper; a more extensive and systematic presentation of the results can be found in Supplement and our Zenodo repository.

## 2 IAMs to emulate

95 Our study uses the output from a total of ten IAMs: nine IAMs used in the ENGAGE project and another IAM GET. The subsections below describe these IAMs and their data used to derive MAC curves.

### 2.1 IAMs from the ENGAGE project

We selected the following nine IAM versions available in the database of the ENGAGE Scenario Explorer: AIM/CGE V2.2, COFFEE 1.1, GEM-E3 V2021, IMAGE 3.0, MESSAGEix-GLOBIOM 1.1, POLES-JRC ENGAGE, REMIND-MAgPIE 2.1-

100 4.2, TIAM-ECN 1.1, and WITCH 5.0 (thereafter, shorter labels indicated in Table 1 will be used). These IAMs are diverse in terms of solution concepts (general equilibrium and partial equilibrium models) and solution methods (intertemporal optimization and recursive dynamic models) (Table 1), among many other perspectives (Guivarch et al., 2022). A series of scenarios following a carbon budget ranging from 200 to 3,000 GtCO$_2$ (for the period of 2019-2100), as well as baseline scenarios, are available from each IAM. All scenarios incorporate second marker baseline scenario from the Shared

105 Socioeconomic Pathways (SSP2), which reflect middle-of-the-road socioeconomic conditions (Riahi et al., 2017). The ENGAGE Scenario Explorer is now part of the larger IPCC Sixth Assessment Report (AR6) Scenario Explorer (Byers et al., 2022), which was not available at the time of our analysis. Although the use of the entire AR6 scenario dataset could be advantageous in terms of the number of IAMs and scenarios available for analyses (189 IAMs (including different model versions) and 1389 scenarios in the AR6 Scenario Explorer; 20 IAMs (including different model versions) and 231 scenarios

110 in the ENGAGE Scenario Explorer), an advantage of using the ENGAGE Scenario Explorer is that the data from IAMs were obtained under a common experimental protocol, allowing consistent analyses.

There are two types of scenarios in the ENGAGE Scenario Explorer: i) end-of-century budget (ECB) scenarios (with "f" in the original scenario name) and ii) peak budget (PKB) scenarios (without "f" in the original scenario name) (Riahi et al., 2021). While the former type of scenarios is defined with a carbon budget till the end of this century, including a possibility

115 of temporarily overspending it before (i.e., a possibility of achieving net negative CO$_2$ emissions), the latter type of scenarios is defined with a carbon budget without allowing temporal budget overspending (i.e., a possibility of achieving net-zero CO$_2$ emissions, but not net negative CO$_2$ emissions). The distinction of the two sets of scenarios may have important near-term implications (Johansson, 2021) and are considered when MAC curves are derived. For each type of scenarios, there are another two types of scenarios: i) scenarios without INDC, which only consider currently implemented national policies (indicated as

120 "NPi2020" in the original scenario name); ii) scenarios with INDC, which further consider national emission pledges until 2030 (indicated as "INDCi2030" in the original scenario name). The availability of scenarios depends on the types of scenarios and varies across IAMs (Table S7). For each IAM, we used the NPi2100 scenario, a scenario assuming a continuation of current stated policies until 2100, as the baseline scenario for all carbon budget scenarios in our analysis. The NPi2100 scenarios, which are available for all IAMs considered here, are only slightly different from the NoPolicy scenarios assuming

125 no climate policies at all.

The ENGAGE Scenario Explorer contains emission data for many greenhouse gases (GHGs) and air pollutants from each IAM, including $CO_2$, $CH_4$, and $N_2O$ emissions analyzed in our study. Emission data are available at global and regional levels (for nine and five IAMs, respectively). There are two sets of regionally aggregated emission data, with one for five regions and the other for ten regions, the latter of which was used in our study: that is, China (CHN), European Union and Western Europe (EUWE), Latin America (LATAME), Middle East (MIDEAST), North America (NORAM), Other Asian countries (OTASIAN), Pacific OECD (PACOECD), Reforming Economies (REFECO), South Asia (SOUASIA), and Sub-Saharan Africa (SUBSAFR). Although all ENGAGE IAMs are regionally disaggregated, only a subset of the IAMs provides data for ten regions in the ENGAGE Scenario Explore as shown in Table 1. Note that the GEM model provides emissions for Rest of World (ROW), one more region in addition to the ten regions, in the ENGAGE Scenario Explorer. In other IAMs, we also allocated emissions for ROW to account for the discrepancy between global emissions and the sum of regional emissions (e.g., 3% difference in $CO_2$ emissions in AIM/CGE V2.2). Regarding emission sources, total anthropogenic emissions and energy-related emissions (e.g., energy and industrial processes) were separately used to derive global MAC curves for three gases (only total anthropogenic emissions for regional MAC curves due to computational requirements for validating regional MAC curves). Non-energy-related emissions (e.g., agriculture, forestry, and land-use sector), the differences between the two, were not used to generate MAC curves because non-energy-related emissions did not appear to be strongly correlated with carbon prices in most IAMs in the ENGAGE project.

## 2.2 GET model

GET is a global energy system model designed to study climate mitigation and energy strategies to achieve long-term climate targets under exogenously given energy demand scenarios (Azar et al., 2003; Hedenus et al., 2010; Azar et al., 2013; Lehtveer and Hedenus, 2015; Lehtveer et al., 2019). It is an intertemporal optimization model that, with perfect foresight, minimizes the total cost of the energy system discounted over the simulation period till 2150 (default discount rate of 5%). To do so, various technologies for converting and supplying energy are evaluated in the model. The model considers primary energy sources such as coal, natural gas, oil, biomass, solar, nuclear, wind, and hydropower. Energy carriers considered in the model are petroleum fuels (gasoline, diesel, and natural gas), synthetic fuels (e.g., methanol), hydrogen, and electricity. End-use sectors in the model are transport, feedstock, residential heat, industrial heat, and electricity. We employed GET version 10.0 (Lehtveer et al., 2019) with the representation of ten regions.

To develop global energy-related $CO_2$ MAC curves reflecting different sets of available mitigation measures, we constructed the following eight technology portfolios: i) Base, ii) Optimistic, iii) Pessimistic, iv) No CCS+Carbon Capture and Utilization (CCU)+Direct Air Capture (DAC) (No_cap), v) Large bioenergy (L_bio), vi) Large bioenergy + Small carbon storage (L_bio/S_str), vii) Small bioenergy + Large carbon storage (S_bio/L_str), and viii) No nuclear (No_nc). The Base portfolio uses the default set of assumptions associated with mitigation options available in the model. The Optimistic portfolio combines the assumptions of Large bioenergy supply, Large carbon storage potential, CCS+CCU+DAC, and Nuclear power.

The Pessimistic portfolio, in contrast, combines those of Small bioenergy supply, Small carbon storage potential, No CCS+CCU+DAC, and No nuclear power. Large and Small bioenergy cases assume 100% more and 50% less bioenergy, respectively, than the default level (134 EJ/year globally). Large and Small carbon storage cases assume 8,000 GtCO$_2$ and 1,000 GtCO$_2$, respectively (2,000 GtCO$_2$ by default). With each of these portfolios, we simulated the model under 22 different carbon price scenarios. In all carbon price scenarios, the carbon price grows 5% each year with a range of initial levels in 2010 (1, 2, 3, 5, 7, 10, …, 140 US\$2010/tCO$_2$) (see Table S1 for details), following the principle of the Hotelling rule where there is a limit on the cumulative emissions (Hof et al., 2021). We assumed a discount rate of 5% for all portfolios and carbon price scenarios. Our analysis used a scenario with zero carbon prices as the baseline scenario. We derived only global energy-related CO$_2$ MAC curves from GET since the model does not explicitly describe processes related to non-energy-related emissions.

# 3 Development of emIAM

## 3.1 Deriving MAC curves

Our MAC curve approach aims to capture the relationship between the carbon price and the emission abatement in IAMs. For each IAM (i.e., ENGAGE IAMs and GET), we calculated the emission reduction level relative to the respective baseline level at each time step. Emission reductions can be expressed either in absolute terms (for example, in GtCO$_2$) or in percentage terms (in percentage relative to the baseline level) (Kesicki, 2013; Jiang et al., 2022), the latter of which is used in our analysis. When the emission is at the baseline level, the relative emission reduction is, by definition, 0%. When it is 100%, which can occur for CO$_2$, the emission is (net) zero. When it exceeds 100%, the emission becomes (net) negative. The carbon price for each case is also the relative level to the baseline scenario. If there are non-zero carbon prices in the baseline scenarios (small carbon prices can be found in baseline scenarios from some IAMs), we subtracted them from the carbon prices in the mitigation scenarios. The MAC curves were derived from the data for the period 2020-2100 in the case of ENGAGE IAMs and GET (we did not consider the data from GET after 2100).

There are three key assumptions in our approach: i) MAC curves are assumed to be time-independent, ii) abatement levels are assumed to be independent across gases, and iii) abatement levels are assumed to be independent across regions. While MAC curves are more commonly time-dependent or for a specific point in time, time-independent MAC curves have also been used for long-term pathway calculations (Johansson et al., 2006; Tanaka and O'Neill, 2018; Tanaka et al., 2021) and short-term assessments (De Cara and Jayet, 2011). The implications of the first assumption are discussed later in this section and Section 5. The second assumption implies that co-reductions of GHG emissions (e.g., CO$_2$ and CH$_4$ emission reductions from an early retirement of a coal-fired power plant (e.g., Tanaka et al., 2019)) are not explicitly considered in our MAC curve approach. The third assumption implies that GHG abatements occur exclusively in each region without relying on other regions. The validity of these assumptions can be seen in Section 4. Additional conditions were applied to derive MAC curves from each model, as summarized in Table 1. These conditions were identified based on visual inspection of the data from each IAM.

**Table 1. Models and data considered for emIAM.** This table describes the features of models (including the versions used) and the data (gases, regions (ten regions)) that were used to derive our MAC curves. "Solution concept" and "solution method" for ENGAGE IAMs (first nine IAMs in the table) are based on Riahi et al. (2021), Guivarch et al. (2022), and IAMC_wiki (2022). Total anthropogenic (and separately energy-related and non-energy-related) $CO_2$, $CH_4$, and $N_2O$ emissions were taken from ENGAGE IAMs; only energy-related $CO_2$ emissions were used from GET.

| Model | Label | Solution concept | Solution method | Spatial resolution | Gas | Range of carbon budget (GtCO$_2$) | Number of scenarios | Data range for MAC curve fitting |
|---|---|---|---|---|---|---|---|---|
| AIM/CGE V2.2 | AIM | General equilibrium | Recursive dynamic | Global Regional | $CO_2$ $CH_4$ $N_2O$ | 300-1,800 | 33 | Carbon prices lower than \$100/tCO$_2$ before 2040 and all data after 2040 |
| COFFEE 1.1 | COFFEE | Partial equilibrium | Intertemporal optimization | Global Regional | $CO_2$ $CH_4$ $N_2O$ | 400-2,500 | 52 | Carbon prices lower than \$50/tCO$_2$ with abatement levels below 100% under scenarios without negative emissions |
| GEM-E3 V2021 | GEM | General equilibrium | Recursive dynamic | Global Regional | $CO_2$ $CH_4$ $N_2O$ | 400-1,800 | 23 | All scenarios |
| IMAGE 3.0 | IMAGE | Partial equilibrium | Recursive dynamic | Global Regional | $CO_2$ $CH_4$ $N_2O$ | 600-3,000 | 21 | All data except: EN_INDCi2030_800f, EN_NPi2020_600f, EN_INDCi2030_1000f, EN_Npi2020_800 |
| MESSAGEix-GLOBIOM 1.1 | MESSAGE | General equilibrium | Intertemporal optimization | Global Regional | $CO_2$ $CH_4$ $N_2O$ | 200-3,000 | 51 | All scenarios except: EN_NPi2020_450, EN_NPi2020_500 |
| POLES-JRC ENGAGE | POLES | Partial equilibrium | Recursive dynamic | Global | $CO_2$ $CH_4$ $N_2O$ | 300-3,000 | 53 | Carbon price below \$1,000/tCO$_2$ before 2050 and below \$5,000/tCO$_2$ thereafter |
| REMIND-MAgPIE 2.1-4.2 | REMIND | General equilibrium | Intertemporal optimization | Global | $CO_2$ $CH_4$ $N_2O$ | 200-3,000 | 57 | All scenarios except: EN_INDCi2030_700, EN_INDCi2030_800, EN_NPi2020_400, EN_NPi2020_500 |
| TIAM-ECN 1.1 | TIAM | Partial equilibrium | Intertemporal optimization | Global | $CO_2$ $CH_4$ $N_2O$ | 800-3,000 | 35 | All scenarios |
| WITCH 5.0 | WITCH | General equilibrium | Intertemporal optimization | Global | $CO_2$ $CH_4$ $N_2O$ | 400-3,000 | 53 | All scenarios |
| GET 10.0 | GET | Partial equilibrium | Intertemporal optimization | Global | Energy $CO_2$ | - | 22 | Carbon prices lower than \$5,000/tCO$_2$; excluded data for very high abatements with disproportionally low costs (found typically after 2100) |

We fit a mathematical function $f(x)$ to the data from each IAM as a MAC curve to capture the emission abatement level for a given carbon price. In selecting the functional form of MAC curves, we had to balance the competing requirements of i) capturing complex nonlinear relationships between the carbon price and the abatement level and ii) keeping the functional form at low complexity. We therefore tested the performance of several functional forms to fit the data, some of which were based on previous studies (Johansson, 2011; Su et al., 2017; Tanaka and O'Neill, 2018). The candidate functions are summarized in Table S2, along with the ranges of parameters considered. To infer a good functional form, we further tried the symbolic regression approach by using the software HeuristicLab, but we were unable to obtain a functional form that is more satisfactory than those suggested in Table S2. Our results indicated that the polynomial function with two algebraic terms (equation (1)) gave the highest $r^2$ and adjusted $r^2$ among the equations tested in more than 50% of the cases, consistently performing best for all IAMs (see the Zenodo repository and Table S3). A polynomial function with only one algebraic term was insufficient: two distinct algebraic terms are generally needed to capture the trend of our data (sometimes with a kink like a "reversed L" shape or with a plateau as shown later).

Therefore, we used a common functional form of equation (1) to generate MAC curves for all cases (i.e., models, gases, regions, and sources in ENGAGE IAMs, and portfolios in GET) for consistency, comparability, and simplicity of use.

$$f(x) = a \times x^b + c \times x^d \tag{1}$$

$a$, $b$, $c$, and $d$ are the parameters to be optimized in each case. $x$ is the variable representing the emission abatement level in percentage relative to the assumed baseline level. The carbon price (i.e., $f(x)$ in equation (1)) is expressed in per ton of $CO_2$-equivalent emissions, using GWP100 (28 and 265 for $CH_4$ and $N_2O$, respectively (IPCC, 2013)) to convert $CH_4$ and $N_2O$ emissions, as assumed in the IAMs emulated here (Harmsen et al., 2016). GWP100 is effectively the default emission metric used to convert non-$CO_2$ GHG emissions to the common scale of $CO_2$ and has been used for decades in multi-gas climate policies and assessments, including the Paris Agreement (Lashof and Ahuja, 1990; Fuglestvedt et al., 2003; Tanaka et al., 2010; Tol et al., 2012; Levasseur et al., 2016; UNFCCC, 2018; UNFCCC, 2023). Furthermore, we calculate the confidence intervals of the fitted curves using $\widehat{y_i} \mp t_{\frac{\alpha}{2}} \times S_\varepsilon \times \sqrt{1 + \frac{1}{n} + \frac{(x_i - \bar{x})^2}{\sum_{i=1}^n x_i^2 - \frac{(\sum_{i=1}^n x_i)^2}{n}}}$ (Thomson and Emery, 2014), where $S_\varepsilon = \sqrt{\frac{\sum_{i=1}^n (y_i - \widehat{y_i})^2}{n-2}}$, $n$ is the sample size, $t_{\frac{\alpha}{2}}$ is the critical value of t-distribution, $\bar{x}$ is the mean of samples, $\widehat{y_i} = f(x_i)$, and $x_i, y_i$ are the original abatement level and carbon price from the IAM, respectively. Uncertainty is reported in all MAC curves derived in this study. While such uncertainty is useful to indicate the confidence level of the MAC curve, it is not necessarily very obvious how to make use of the uncertainty range in reproducing scenarios by optimization from the IAM emulator (Figure S241).

In addition to deriving the MAC curves, we derived the maximum abatement level from each IAM, which reflected, for example, the limit of CCS capacity and hard-to-abate sectors. The minimum abatement level is, by definition, zero in all simulation periods, as inter-sectoral emission trading that can increase emissions is irrelevant here. We also estimated upper limits of the first and second derivatives of temporal changes in abatement levels, which account for the limits of the rate of

technological change and the socio-economic inertia (e.g., barriers to the diffusion of new technologies (Schwoon and Tol, 2006)), respectively. The limits on the first and second derivatives of abatement changes will prohibit the use of deep mitigation levels in the MAC curve in early periods. These barriers to rapid emission reductions and the associated costs could also be introduced by more complex functional forms internally in the MAC curves (Ha-Duong et al., 1997; Schwoon and Tol, 2006; De Cara and Jayet, 2011; Hof et al., 2021), but we applied such limits externally on the MAC curves. Processes and factors that can cause inertia in IAMs, including capital stock, growth rate constraints on technology expansion, availability of new technologies, learning by doing, and learning with time (Gambhir et al., 2019; Krey et al., 2019; Tong et al., 2019; Shiraki and Sugiyama, 2020), are not explicitly considered in our MAC curve approach, but are partially captured in our approach, which describes percentage reduction rates relative to rising baseline scenarios. For example, constant emission reductions in absolute terms can appear smaller over time in relative terms and thus become less costly in our approach.

For each IAM, we computed the rate of change in the abatement level at each time step from the previous time step (i.e., first derivatives) over the entire available period. We then approximated such data with a *log-normal* distribution and assumed the three-sigma level (upper side) as the maximum of the first derivative of abatement changes. Likewise, we computed the rate of change of the change in the abatement level (i.e., second derivatives), approximated the data with a *normal* distribution, and assumed the three-sigma level as the maximum of the second derivative of abatement changes. We further assumed that the minimum of the first and second derivatives were at the opposite sign of the maximum of the first and second derivatives, respectively. These limits are applied when MAC curves are coupled with ACC2 to generate cost-effective pathways (Section 4).

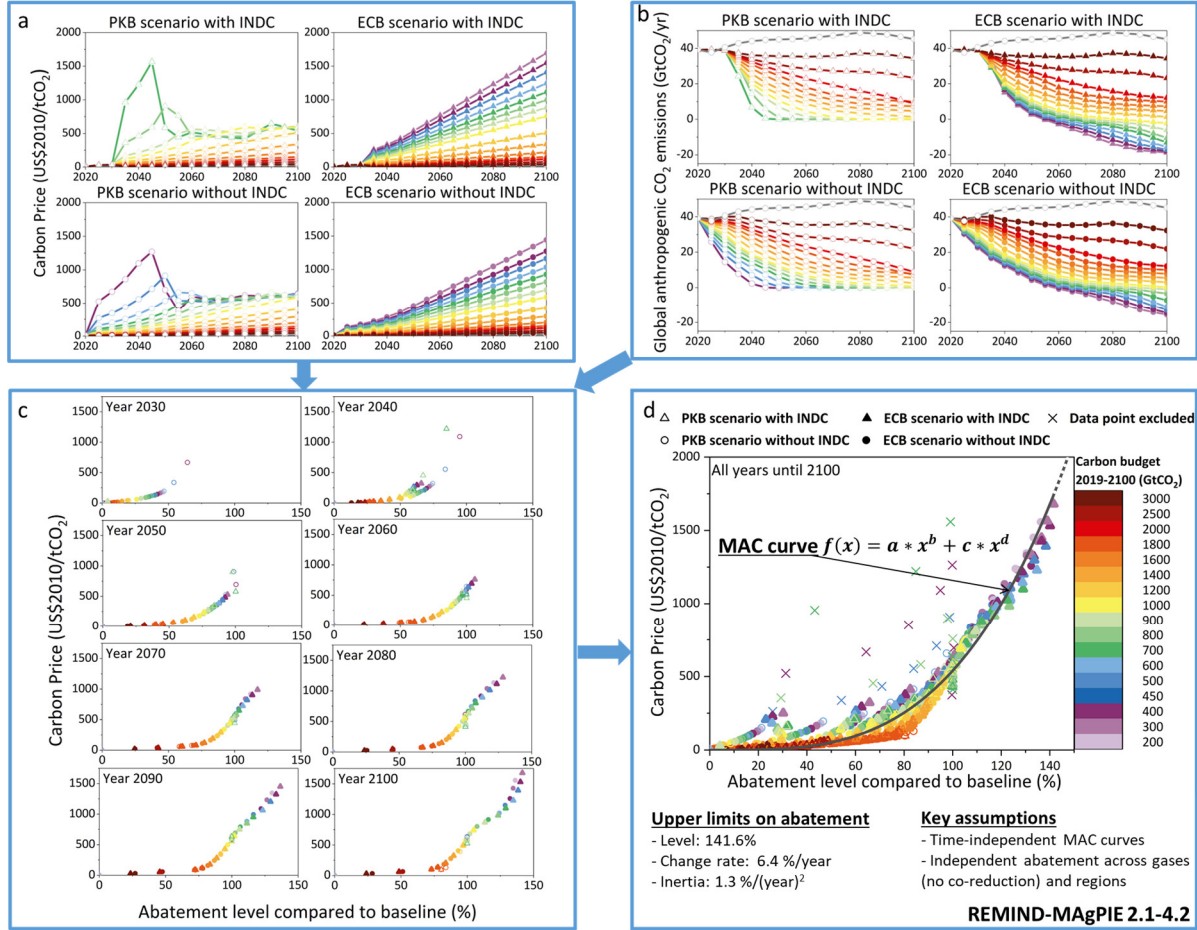

**Figure 1. Overview of the methods to derive MAC curves and limits on abatement (upper limits on abatement levels and their first and second derivatives).** The figure uses the data for global total anthropogenic $CO_2$ emissions from REMIND for illustration. The chromatic colors indicate the respective carbon budgets for the period 2019 – 2100 in $GtCO_2$. The gray color indicates the baseline scenario ("NPi2100" in the original scenario name). Scenarios without INDC consider currently implemented national policies (circle; indicated as "NPi2020" in the original scenario name); scenarios with INDC further consider national emission pledges until 2030 (triangle; indicated as "INDCi2030" in the original scenario name). ECB scenarios consider carbon budgets till the end of this century, with a possibility of temporal budget overspending (filled symbols; with "f" in the original scenario name); PKB scenarios consider carbon budgets without allowing temporal budget overspending (open symbols; without "f" in the original scenario name). Crosses indicate data points from scenarios that were not considered in the derivation of the MAC curve (i.e., EN_INDCi2030_700, EN_INDCi2030_800, EN_NPi2020_400, and EN_NPi2020_500 for REMIND (see Table 1)). In the equation of the MAC curve, $a$, $b$, $c$, and $d$ are the parameters to be optimized; $x$ is the variable representing the abatement level in percentage relative to the assumed baseline level. Note that Panel (c) shows data only for every ten years for the sake of presentation.

In summary, we combined a MAC curve with the upper and lower limits on abatement levels and their first and second derivatives to emulate the behavior of an IAM, as illustrated in Figure 1 using the output from REMIND as an example (corresponding figures for AIM and MESSAGE in Figures S1 and S2 of Supplement). The upper two Panels show the original data from REMIND: the carbon price pathways corresponding to the series of carbon budgets (Figure 1a) and the global anthropogenic $CO_2$ emissions (Figure 1b) from the four types of scenarios (PKB scenarios with INDC, ECB scenarios with

INDC, PKB scenarios without INDC, and ECB scenarios without INDC). These data are rearranged to show the relationship between the carbon price and the abatement level in percentage relative to baseline every ten years (original data every five years before 2060) (Figure 1c). In the near term, data points can only be seen at low abatement levels. With time, data points proceed to deeper abatement levels. Taken together over all years, Figure 1d shows a consistent relationship, providing a basis for a time-independent MAC curve. Outliers arising from very low carbon budget scenarios (crosses in Figure 1d) were identified and manually excluded from the derivation of the MAC curve (Table 1), although excluding such scenario(s) limits the range of applicability of the MAC curve.

The stable MAC curve is an interesting finding in itself because, despite the presence of time-dependent processes in this intertemporal optimization model (Campiglio et al., 2022), the same relationship persists over time between the carbon price and the abatement level. But why does this time-independent approach work so well to capture IAMs that include time-dependent processes? The use of percentage reductions in our MAC curve approach goes some way to explaining this. Since most of the baseline scenarios are rising as noted above, the same amount of emission abatement in absolute terms can become smaller with time in percentage terms, which inadvertently but effectively captures the influences from time-dependent processes in IAMs. When the underlying data are presented in absolute terms, the data distribution appears more dispersed (Figure S3 for AIM, MESSAGE, but to a lesser extent for REMIND). Limits associated with the time-independent approach will be further explored in Section 5.

## 3.2 MAC curves from ENGAGE IAMs

### 3.2.1 Carbon price and abatement level

Figure 2 shows the relationships between the carbon price and the abatement level for global total anthropogenic $CO_2$ emissions obtained from nine ENGAGE IAMs. Overall, the relationships between the carbon price and the $CO_2$ abatement level are well captured by time-independent MAC curves for most IAMs here. The results vary in terms of the range of carbon prices, the range of abatement levels, and the dispersion of data points. For example, the carbon prices of AIM and COFFEE remain below \$500/tCO$_2$, while the carbon prices of POLES and MESSAGE can exceed \$5,000/tCO$_2$. The maximum abatement levels of COFFEE, POLEs and REMIND are over 140%, while most of the others are in the range of 110%-130%. AIM provides a limited amount of data at low abatement levels. IMAGE and POLES produce more dispersed data distributions than other models, which may be related to the fact that these models are recursive dynamic models (Table 1); however, the other recursive dynamic models, AIM and GEM, produce less dispersed data distributions that can be well captured by MAC curves. POLES can be seen as an example where our time-independent MAC curve approach does not work well (See Section 5 for further discussion). The MAC curve, if taken every five years, shifts to the right over time (Figure S4). Visual inspection of the data distributions reveals little difference between the ECB scenarios and PKB scenarios (except for WITCH), indicating that the MAC curves are generally consistent for both types of scenarios in these IAMs. Note that the MAC curves are not very sensitive to the underlying sets of scenarios considered, at least for the five IAMs (COFFEE, MESSAGE, POLES,

REMIND, and WITCH), which provide comparable carbon budget ranges and similar numbers of scenarios, while the distributions of scenarios are generally not homogeneous (Table S7). The MAC curves of the five IAMs are only slightly affected when we consider only a subset of scenarios whose carbon budgets are available for all five models (19 scenarios) (Figure S37). Results for other gases and for energy-related emissions are shown in Figures S5-S36.

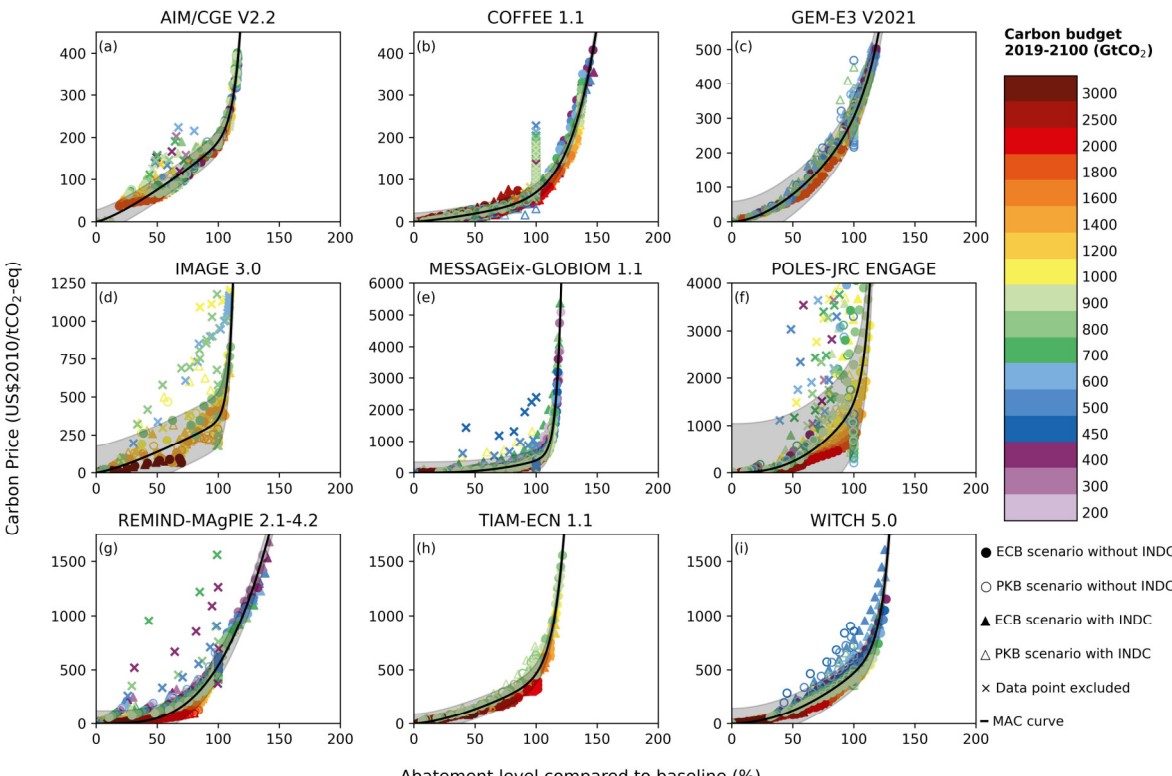

**Figure 2. Relationships between the carbon price and the global total anthropogenic CO₂ abatement level obtained from nine ENGAGE IAMs.** Each Panel shows the results from each ENGAGE IAM. Data were obtained from the ENGAGE Scenario Explorer and are shown in colors and markers as designated in the legend. Black lines are the MAC curves. Crosses are the data points that were not included in the derivation of MAC curves (Table 1). The shaded bands are the 95% confidence intervals of the fitted curves.

### 3.2.2 First and second derivatives of abatement changes

The first and second derivatives of temporal changes in abatement levels for global total anthropogenic $CO_2$ emissions from each ENGAGE IAM are shown in Figure 3. Data for the first derivatives primarily distribute on the positive side and can be best captured by log-normal distributions, among other distributions tested. On the other hand, data for the second derivatives spread on both the positive and negative sides and can be approximated by normal distributions. Based on visual inspection, we found that three-sigma ranges of distributions can largely capture data ranges. We therefore use three-sigma ranges as the limits on the first and second derivatives of abatement changes. There are outliers (now shown) originating from PKB scenarios, which we speculate were caused by sudden declines in carbon prices around the period of achieving net zero $CO_2$ emissions (Figure SI 1.1-6 of Riahi et al. (2021)). These outliers were effectively removed by considering three-sigma ranges (rather than the maxima and minima of the original data points). For other gases and for energy-related emissions, see Figures S38-

315

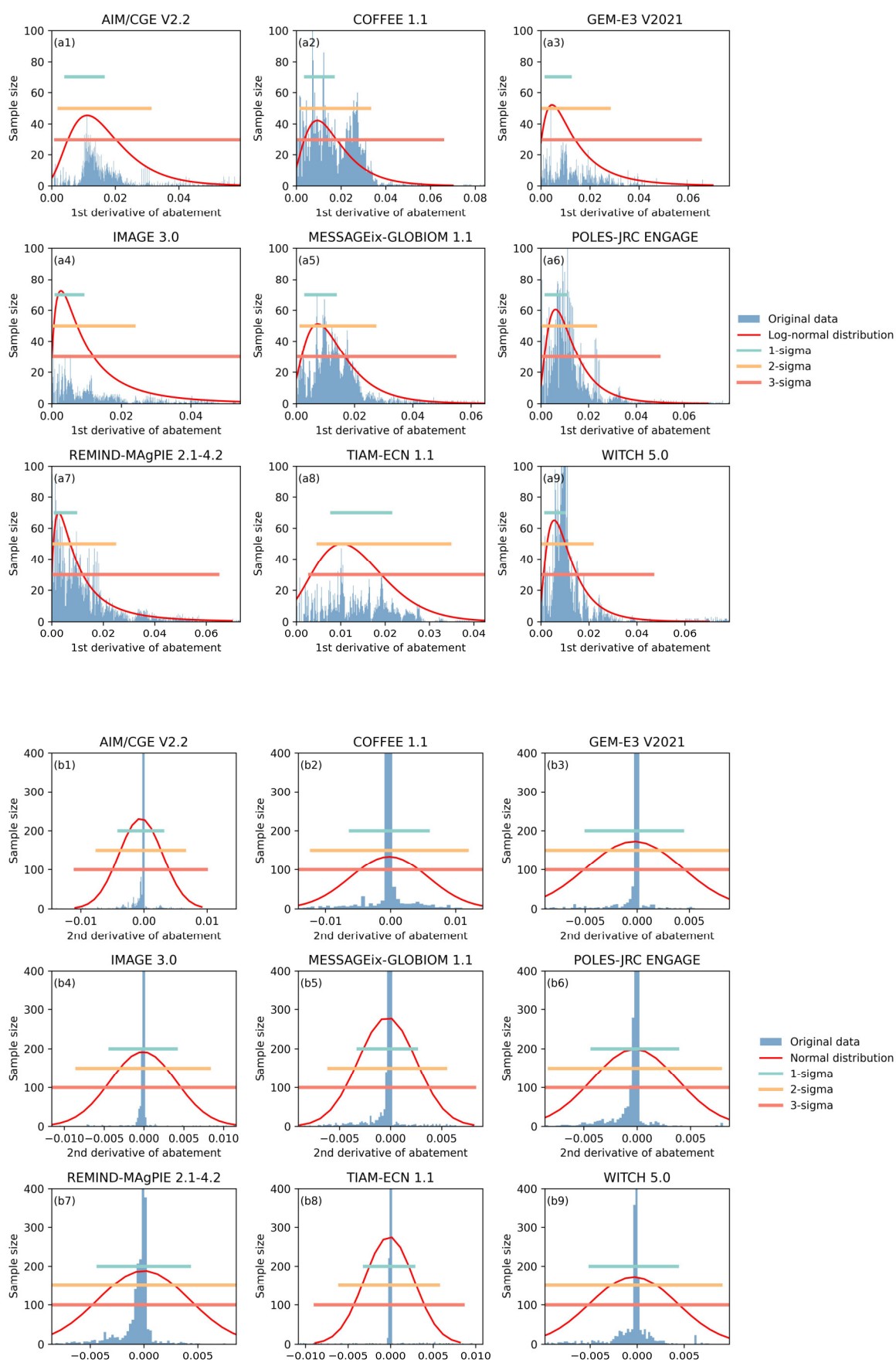

**Figure 3. The first and second derivatives of temporal changes in abatement levels for the global total anthropogenic CO₂**

**emissions from each ENGAGE IAM.** A log-normal distribution is applied to the data for the first derivatives of abatement changes obtained from each IAM (Panels (a1) to (a9)). A normal distribution is applied to the data for the second derivatives of abatement changes obtained from each IAM (Panels (b1) to (b9)).

The upper limits on the first and second derivatives of abatement changes estimated for ENGAGE IAMs are summarized in Table 2. Those for ACC2 were assumed to be 4.0 %/year and 0.4 %/(year)$^2$, respectively, for all three gases ($CO_2$, $CH_4$, and $N_2O$) (Tanaka and O'Neill, 2018; Tanaka et al., 2021). ENGAGE IAMs give higher upper limits on the first and second derivatives than ACC2 for $CO_2$. For the other two gases, ENGAGE IAMs also give higher upper limits on the second derivatives but tend to indicate lower upper limits on the first derivatives.

The upper limits on the first and second derivatives of $CO_2$ abatement can determine the earliest possible year of achieving net zero $CO_2$ emissions (i.e., 100% abatement) for each IAM. In the case of ACC2, it is the year 2050 when net zero $CO_2$ emissions become first possible, if the abatement can start in 2020. Figure S88 compares the earliest possible net zero years implied by the upper limits on the first and second derivatives with the years of net zero in carbon budget scenarios from each ENGAGE IAM. The figure shows that the former precedes the latter in all IAMs, indicating that the upper limits based on three-sigma ranges are large enough to allow pathways to achieve net zero as shown by each IAM.

### 3.2.3 Global MAC curves

Figure 4 shows the global MAC curves for total anthropogenic and energy-related $CO_2$, $CH_4$, and $N_2O$ emissions from nine ENGAGE IAMs and other studies. The parameter values of these global MAC curves and associated limits on abatement are shown in Table 2 (for total anthropogenic emissions) and Table S4 (for energy-related emissions).

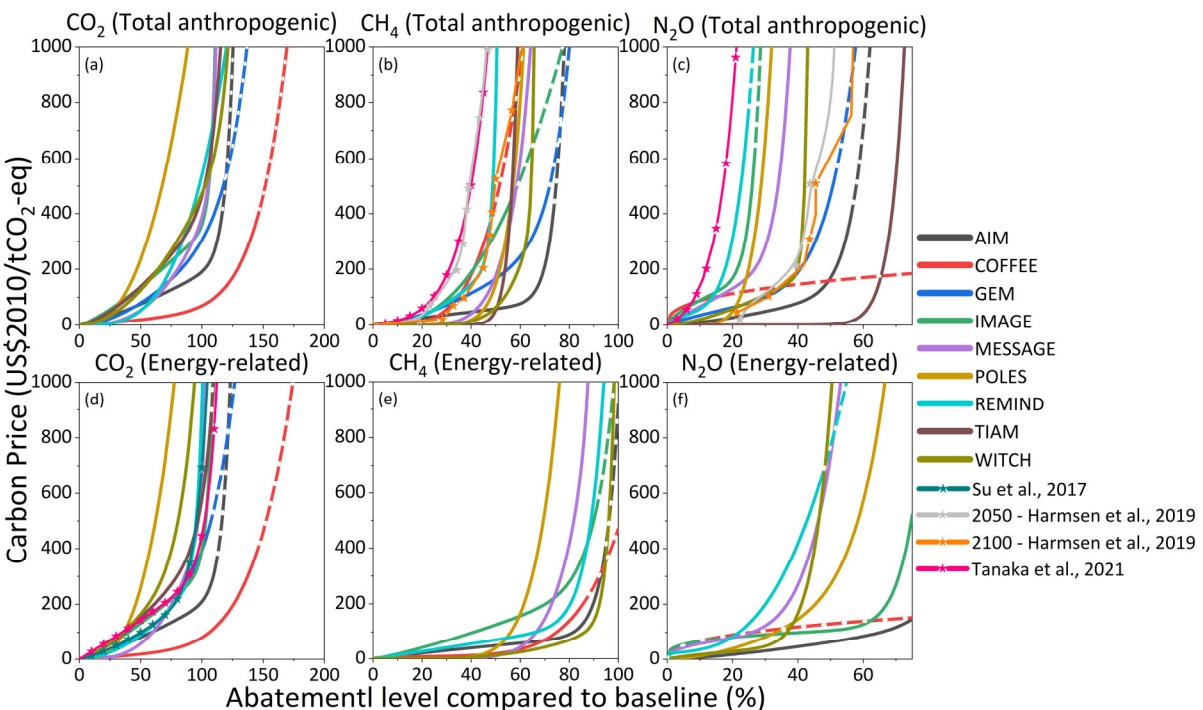

**Figure 4. Global MAC curves for total anthropogenic and energy-related CO$_2$, CH$_4$, and N$_2$O emissions derived from nine ENGAGE IAMs.** In Panels (a) to (f), the solid line indicates that the MAC curve is within the applicable range; the dashed line means that it is outside the applicable range (i.e., above the maximum abatement level indicated from underlying IAM simulation data or above the range of carbon prices considered for fitting the MAC curve; see Tables 1 and 2). Different colors indicate different IAMs. The MAC curves from selected previous studies (Su et al., 2017; Harmsen et al., 2019; Tanaka et al., 2021) are shown for comparison. The MAC curves from Harmsen et al., (2019) are time-dependent and the figure shows those for the years 2050 and 2100.

**Table 2. Parameter values of global MAC curves for total anthropogenic CO$_2$, CH$_4$, and N$_2$O emissions derived from nine ENGAGE IAMs and associated limits on abatement.** See equation (1) for parameters $a$, $b$, $c$, and $d$. MaxABL denotes the maximum abatement level (%) of each gas indicated from IAM simulation data. The units for $a$ and $c$ are US\$2010/tCO$_2$. Max1st and Max2nd represent the maximum first and second derivatives (%/year and %/(year)$^2$), respectively, of abatement changes of each gas also derived from IAM simulation data. For those of global MAC curves for energy-related CO$_2$, CH$_4$, and N$_2$O emissions, see Table S4. For those of regional MAC curves, see the Zenodo repository.

| Model | Gas | $a$ | $b$ | $c$ | $d$ | MaxABL | Max1st | Max2nd |
|---|---|---|---|---|---|---|---|---|
| AIM | CO$_2$ | 182.14 | 1.27 | 8.68 | 19.71 | 116.2 | 5.9 | 1.0 |
| | CH$_4$ | 108.99 | 0.91 | $7.868 \times 10^4$ | 17.91 | 73.6 | 6.1 | 1.3 |
| | N$_2$O | 282.34 | 1.46 | $2.436 \times 10^5$ | 11.84 | 56.1 | 4.5 | 1.0 |
| COFFEE | CO$_2$ | 46.66 | 1.29 | 22.59 | 7.01 | 147.2 | 6.5 | 1.8 |
| | CH$_4$ | 3,658.91 | 4.05 | 3,658.91 | 4.05 | 47.7 | 2.3 | 1.3 |
| | N$_2$O | 102.75 | 0.37 | 102.75 | 0.37 | 20.2 | 3.9 | 1.4 |
| GEM | CO$_2$ | 267.14 | 1.76 | 36.85 | 8.53 | 118.2 | 6.5 | 1.4 |
| | CH$_4$ | 486.16 | 1.59 | 7,133.48 | 10.70 | 72.0 | 4.6 | 1.1 |
| | N$_2$O | 240.14 | 0.83 | $3.107 \times 10^4$ | 6.54 | 51.1 | 4.0 | 0.9 |
| IMAGE | CO$_2$ | 330.58 | 1.27 | 28.57 | 29.83 | 110.1 | 6.3 | 1.2 |
| | CH$_4$ | 959.11 | 2.53 | 959.11 | 2.53 | 58.3 | 3.1 | 0.6 |
| | N$_2$O | 426.52 | 0.68 | $1.541 \times 10^8$ | 9.70 | 26.3 | 2.4 | 0.5 |
| MESSAGE | CO$_2$ | 368.79 | 2.78 | 18.30 | 30.24 | 120.9 | 5.4 | 0.8 |
| | CH$_4$ | 16789 | 6.57 | $3.292 \times 10^7$ | 29.08 | 73.3 | 3.5 | 0.6 |
| | N$_2$O | 610.67 | 0.97 | $7.910 \times 10^6$ | 9.47 | 45.2 | 1.9 | 0.3 |
| POLES | CO$_2$ | 1,347.98 | 2.52 | 144.57 | 21.87 | 147.1 | 5.3 | 1.2 |
| | CH$_4$ | $4.816 \times 10^4$ | 9.36 | $4.816 \times 10^4$ | 9.36 | 75.9 | 4.3 | 1.0 |
| | N$_2$O | $1.513 \times 10^6$ | 6.42 | $1.513 \times 10^6$ | 94.73 | 37.3 | 2.3 | 0.5 |
| REMIND | CO$_2$ | 269.52 | 3.38 | 269.52 | 3.38 | 141.6 | 6.4 | 1.3 |
| | CH$_4$ | 1,002.16 | 2.11 | $1.610 \times 10^{11}$ | 28.11 | 51.2 | 3.4 | 1.2 |
| | N$_2$O | 224.21 | 0.65 | $6.334 \times 10^5$ | 4.92 | 24.8 | 1.6 | 1.0 |
| TIAM | CO$_2$ | 384.32 | 1.48 | 78.52 | 13.31 | 121.7 | 5.6 | 0.9 |
| | CH$_4$ | $1.23 \times 10^7$ | 17.81 | 157.83 | 100 | 59.5 | 3.9 | 1.0 |
| | N$_2$O | $2.151 \times 10^5$ | 16.79 | 99.08 | 100 | 73.3 | 4.3 | 2.3 |
| WITCH | CO$_2$ | 462.12 | 1.89 | 10.13 | 18.05 | 128.2 | 4.7 | 1.4 |
| | CH$_4$ | 6,658.29 | 6.72 | $2.781 \times 10^{15}$ | 69.59 | 66.7 | 3.7 | 2.1 |
| | N$_2$O | 681.73 | 1.52 | $9.130 \times 10^{18}$ | 43.78 | 42.8 | 3.1 | 1.1 |

MAC curves for total anthropogenic and energy-related CO$_2$ emissions resemble each other since total anthropogenic CO$_2$ emissions are predominantly energy-related CO$_2$ emissions. COFFEE gives the lowest carbon prices among all IAMs

over a wide range of abatement levels; POLES shows the highest carbon prices. AIM has the second-lowest carbon prices at abatement levels of 63% and above. REMIND gives higher carbon prices than AIM above the abatement level of 60%. The functional form of the MAC function used by Su et al. (2017) is consistent with our study and Tanaka et al. (2021) used equation (2) in Table S2. Harmsen et al. (2019) considered time-dependent MAC curves and no explicit function is provided. Despite some differences in the form of the functions, the MAC curves for energy-related $CO_2$ used in Su et al. (2017) and

Tanaka et al. (2021) are within the range of the MAC curves from ENGAGE IAMs, but the MAC curves for $CH_4$ and $N_2O$ used in Tanaka et al. (2021) are higher. The $CH_4$ MAC curve in 2050 of Harmsen et al. (2019) is also higher than the range of the $CH_4$ MAC curves from ENGAGE IAMs, but that in 2100 are close to that range. Harmsen's $N_2O$ MAC curves are within the corresponding range of ENGAGE IAMs and not much different between 2050 and 2100.

The difference between MAC curves for total anthropogenic and energy-related emissions is more pronounced for

$CH_4$ and $N_2O$ than for $CO_2$ because of greater mitigation opportunities outside of the energy sector. $CH_4$ MAC curves generally rise sharply at lower abatement levels than $CO_2$ MAC curves. All MAC curves for energy-related $CH_4$ emissions are low up to about 50% abatement level, presumably reflecting low-cost abatement opportunities. AIM and WITCH give a low carbon price up to 80-90% abatement level for energy-related $CH_4$ emissions. Due to limited $N_2O$ abatement opportunities, $N_2O$ MAC curves rise steeply at low abatement levels, with the one from REMIND rising earliest.

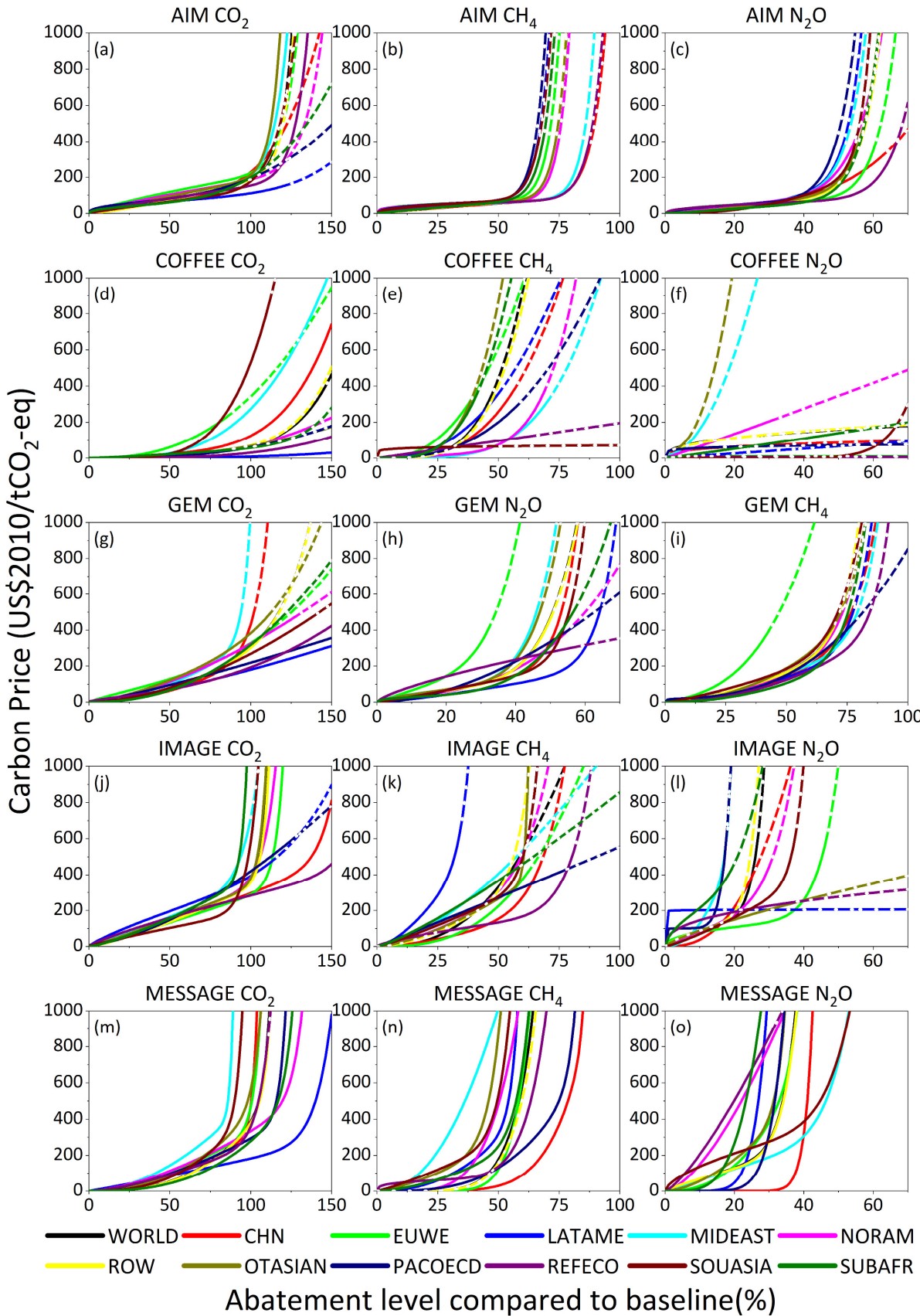

**Figure 5. Regional MAC curves for total anthropogenic CO₂, CH₄, and N₂O emissions derived from five ENGAGE IAMs.** The

solid line indicates that the MAC curve is within the applicable range; the dashed line means that it is outside the applicable range (i.e.,

above the maximum abatement level indicated from underlying IAM simulation data or above the range of carbon prices considered for

fitting the MAC curve; see Tables 1 and 2). Different colors indicate different regions: China (CHN), European Union and Western

 Europe (EUWE), Latin America (LATAME), Middle East (MIDEAST), North America (NORAM), Other Asian countries (OTASIAN), Pacific OECD (PACOECD), Reforming Economies (REFECO), South Asia (SOUASIA), Sub-Saharan Africa (SUBSAFR), and Rest of World (ROW).

### 3.2.4 Regional MAC curves

Figure 5 shows the regional MAC curves for total anthropogenic $CO_2$, $CH_4$, and $N_2O$ emissions from five ENGAGE IAMs. The parameter values of the regional MAC curves and associated limits on abatement can be found in our Zenodo repository. While various inter-model and inter-regional differences can be seen in Figure 5, the regional variations of the AIM MAC curves appear to be the smallest for all three gases.

MIDEST generally shows a high $CO_2$ MAC curve relative to other regions. LATAM gives the lowest MAC curve at abatement levels above approximately 79% in all IAMs considered here, except for the IMAGE model with SOUASIA and REFECO being the lowest MAC curve at abatement levels of above and below 90%, respectively. LATAM also indicates very deep $CO_2$ abatement potentials exceeding 150% in some models. AIM's $CH_4$ MAC curves indicate low-cost $CH_4$ abatement opportunities up to abatement levels of approximately 50% in all regions, while such opportunities appear less abundant in the $CH_4$ MAC curves from other models. REFECO exhibits a very low $CH_4$ MAC curve in all five models. MIDEST gives either a high or a low $CH_4$ MAC curve, depending on the IAM. The $N_2O$ MAC curves generally rise sharply earlier than the $CH_4$ MAC curves.

### 3.3 MAC curves from GET

Figure 6 shows the relationships between the carbon price and the abatement level of global energy-related $CO_2$ emissions and their dependency on the underlying technology portfolios considered in GET. MAC curves from different technology portfolios are compared in Figure 7. They are further compared with the global MAC curves for energy-related $CO_2$ emissions from ENGAGE IAMs and other studies. The parameter values of these global MAC curves and associated limits on abatement are shown in Table 3. Further details of the first and second derivatives of abatement changes from GET can be found in Figures S38 and S39.

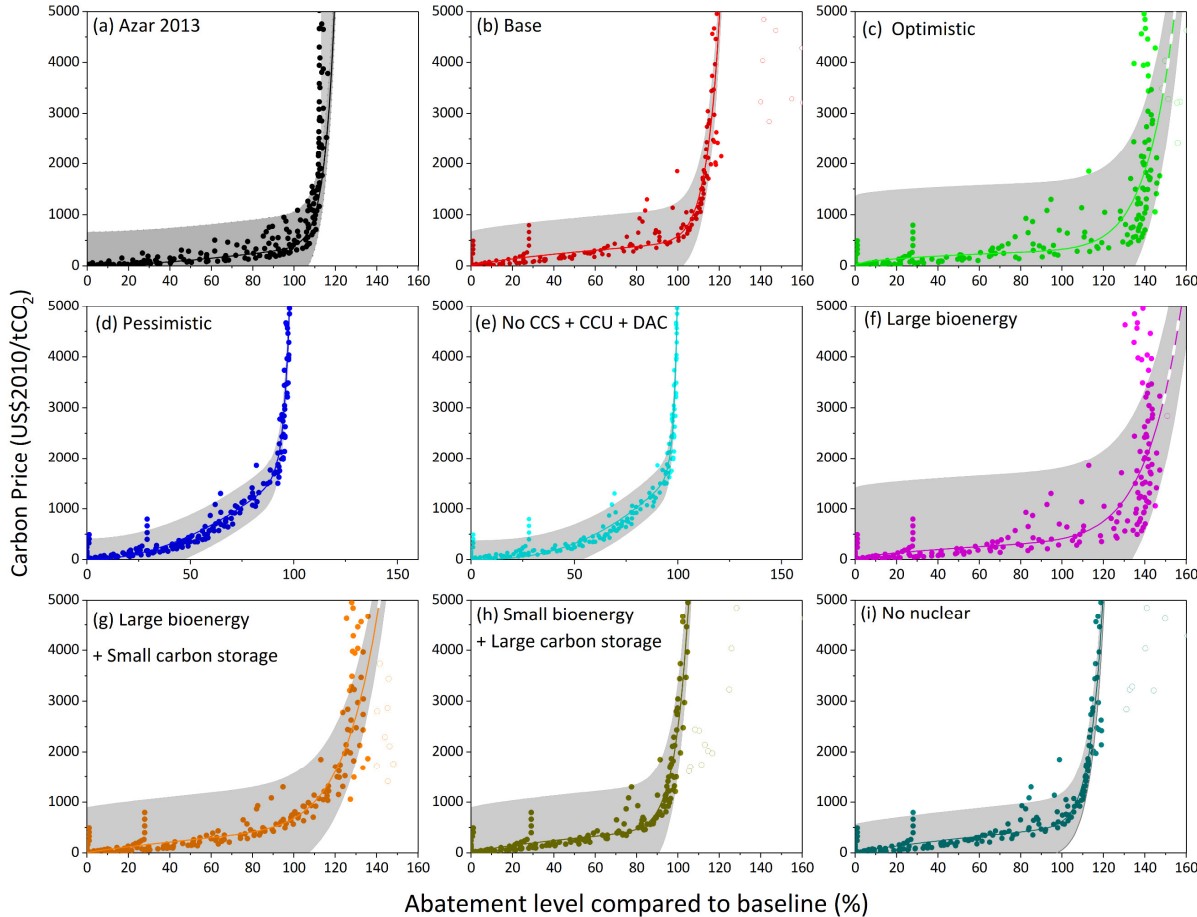

**Figure 6. Relationships between the carbon price and the global energy-related CO₂ abatement level obtained from GET with different portfolios of available mitigation technologies.** Panel (a) shows the results obtained from an older version of GET (Azar et al., 2013) for the sake of comparison. Panels (b) to (i) show the results from GET (Lehtveer et al., 2019) with different technology portfolios. See Section 2.2 for the definitions of technology portfolios. Points are the data obtained from GET; lines are the MAC curves calculated based on our approach. Open circles are the data that were not considered in the derivation of MAC curves (Table 1) and are typically found after 2100, in some cases above the abatement level of 160% (not shown). Note that we have converted the unit in Panel (a) from US$2010/tC, which is used in the older version of GET, to US$2010/tCO₂, the commonly used unit here. The shaded bands are the 95% confidence intervals of the fitted curves calculated.

Global MAC curves for energy-related CO₂ emissions from different technology portfolios cover a wide range. The range is almost as wide as that from ENGAGE IAMs (i.e., inter-portfolio range ≈ inter-model range), if we disregard the MAC curve from COFFEE (Figure 2d). The MAC curve from the Base portfolio is generally higher than the MAC curve based on the previous version of the model (Azar et al., 2013; Tanaka and O'Neill, 2018), reflecting the biomass supply potential being smaller in the GET version used in our analysis (i.e., 134 EJ/year) than in the previous version (approximately 200 EJ/year), among other reasons. The maximum abatement level of the Base portfolio is about 120%, which is slightly higher than the estimate of 112% based on the previous model version. The Optimistic portfolio generally gives lower carbon prices and deeper mitigation potentials than the Base portfolio. Conversely, the Pessimistic portfolio shows higher carbon prices and more limited mitigation potential than the Base portfolio. The Optimistic and Large bioenergy portfolios yield more than 150%

CO$_2$ abatement levels at maximum. The Large bioenergy + Low carbon storage portfolio gives lower maximum abatement levels than the previous two portfolios due to the assumed lower carbon storage potential. The Low bioenergy + Large carbon storage portfolio limits the maximum CO$_2$ abatement levels at only slightly above 100%. With the Pessimistic portfolio, the maximum CO$_2$ abatement levels do not exceed 100% (i.e., no net negative CO$_2$ emissions) primarily because no carbon capture technologies such as CCS, CCU, and DAC are available. Likewise, the No CCS+CCU+DAC portfolio also gives a maximum abatement level below 100%. The No nuclear portfolio gives a similar relationship to the one from the Base portfolio, indicating a limited role of nuclear energy here. Finally, the results are somewhat, but not strongly, sensitive to the choice of discount rate (5% by default), as indicated by the results based on alternative discount rates of 3% and 7%, where the growth rate of carbon price is fixed at the value of the respective discount rate based on the Hotelling rule (Figure S89). The deployment of some technologies leads to a rapid increase and then a saturating increase rate of abatement price, as reflected in values of $b$ less than 1 and values of $d$ greater than 1 (Table 3). In general, a policy mix with more technologies leads to lower carbon costs, despite the relatively high upfront costs of technology deployment and use.

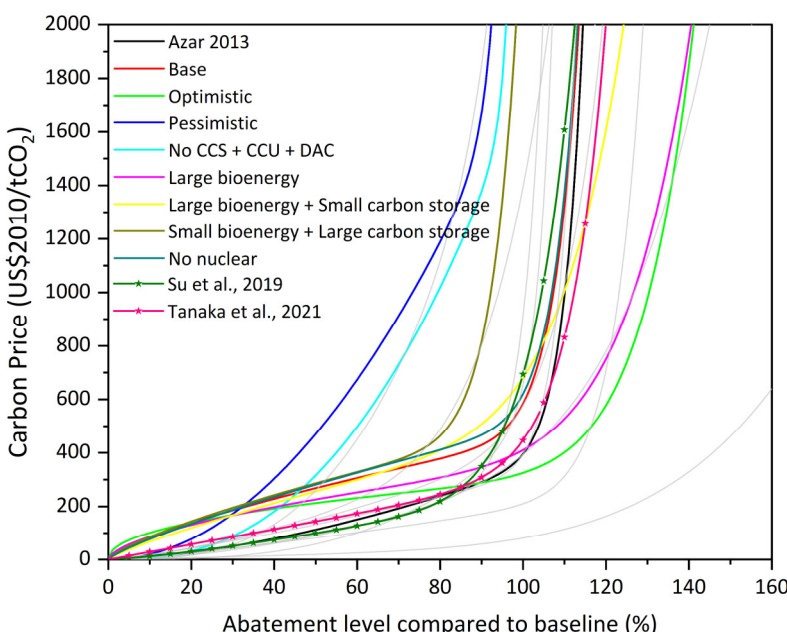

**Figure 7. Global MAC curves for energy-related CO$_2$ emissions derived from the GET model with different portfolios of available mitigation technologies.** Different colors indicate different technology portfolios (see Section 2.2 for details). Global MAC curves for energy-related CO$_2$ emissions from ENGAGE IAMs are shown as a comparison in gray lines, and the MAC curves from selected previous studies (Su et al., 2017; Tanaka et al., 2021) are shown in lines with stars.

**Table 3. Parameter values of global MAC curves for energy-related $CO_2$ emissions derived from GET and associated limits on abatement.** See equation (1) for parameters $a$, $b$, $c$, and $d$. The units for $a$ and $c$ are US$2010/tCO_2$. MaxABL denotes the maximum abatement level (%) of $CO_2$ indicated from GET simulation data. Max1st and Max2nd represent the maximum first and second derivatives (%/year and %/(year)$^2$), respectively, of abatement changes.

| Technology portfolio | Gas | $a$ | $b$ | $c$ | $d$ | MaxABL | Max1st | Max2nd |
|---|---|---|---|---|---|---|---|---|
| Azar 2013 | $CO_2$ | 338.61 | 1.58 | 57.08 | 24.59 | 112 | 5.6 | 0.9 |
| Base | $CO_2$ | 441.86 | 0.72 | 142.54 | 18.73 | 121 | 7.4 | 1.3 |
| Optimistic | $CO_2$ | 292.67 | 0.46 | 32.43 | 11.41 | 148 | 11.5 | 2.1 |
| Pessimistic | $CO_2$ | 1,839.19 | 1.97 | 6,716.35 | 34.62 | 100 | 4.5 | 0.8 |
| No CCS + CCU + DAC | $CO_2$ | 1,775.74 | 2.49 | 3,707.48 | 53.90 | 100 | 5.4 | 0.9 |
| Large bioenergy | $CO_2$ | 340.99 | 0.59 | 69.68 | 9.17 | 148 | 11.3 | 2.0 |
| Large bioenergy + Small carbon storage | $CO_2$ | 452.10 | 0.82 | 229.12 | 8.52 | 140 | 7.6 | 1.5 |
| Small bioenergy + Large carbon storage | $CO_2$ | 480.65 | 0.75 | 1,992.76 | 15.93 | 105 | 6.1 | 1.1 |
| No nuclear | $CO_2$ | 489.97 | 0.80 | 131.23 | 19.52 | 120 | 7.2 | 1.3 |

## 4. Validation of ACC2-emIAM

### 4.1 ACC2 model

To validate the performance of our MAC curves emulating IAM responses (i.e., emIAM), we coupled emIAM with the ACC2 model (ACC2-emIAM). ACC2 dates back to the impulse response functions of the global carbon cycle and climate system (Hasselmann et al., 1997; Hooss et al., 2001; Bruckner et al., 2003). The model was later developed to a simple climate model with a full set of climate forcers (Tanaka et al., 2007) and then to the current form (Tanaka et al., 2013; Tanaka and O'Neill, 2018; Tanaka et al., 2021): a simple climate-economy model[1] that consists of i) carbon cycle, ii) atmospheric chemistry, iii) physical climate, and iv) mitigation modules.

The representations of natural Earth system processes in the first three modules of ACC2 are at the global-annual-mean level as in other simple climate models (Joos et al., 2013; Nicholls et al., 2020). The carbon cycle module falls into the category of box models (Mackenzie and Lerman, 2006) and the physical climate module is a heat diffusion model DOECLIM (Kriegler, 2005). ACC2 covers a comprehensive set of direct and indirect climate forcers: $CO_2$, $CH_4$, $N_2O$, $O_3$, $SF_6$, 29 species of halocarbons, OH, $NO_x$, CO, VOC, aerosols (both radiative and cloud interactions), and stratospheric $H_2O$. The model captures key nonlinearities, for example, those associated with $CO_2$ fertilization, tropospheric $O_3$ production from $CH_4$, and ocean heat diffusion. Uncertain parameters are optimized (Tanaka et al., 2009a; Tanaka et al., 2009b; Tanaka and Raddatz, 2011) based on an inverse estimation theory (Tarantola, 2005). The equilibrium climate sensitivity is assumed at 3 °C, the best

---

[1] ACC2-emIAM (and ACC2 with the previous version of MAC curves) can be broadly viewed as an IAM, that is, a simple cost-effective IAM that considers global mitigation costs relative to an assumed baseline. In terms of the level of simplicity, ACC2-emIAM is similar to the DICE model (Nordhaus, 2017) and other simple cost-benefit IAMs that inform the social cost of carbon (Errickson et al., 2021; Rennert et al., 2022). However, ACC2-emIAM does not have an economic growth model and does not account for climate damage. In this study, ACC2-emIAM is characterized as a climate-economy model, but not an IAM, to distinguish it from the more complex IAMs emulated by the MAC curves. ACC2-emIAM also differs from these complex IAMs, which are typically not directly coupled with a climate model, with some versions of GET (Azar et al., 2013; Gaucher et al., 2023) being exceptions.

estimate of IPCC (2021). The mitigation module contains a set of global MAC curves for $CO_2$, $CH_4$, and $N_2O$ (Johansson, 2011; Azar et al., 2013), which is a previous version of MAC curves to be replaced with the MAC curves derived in this study. ACC2 can be used to optimize $CO_2$, $CH_4$, and $N_2O$ emission pathways based on a cost-effectiveness approach. That is, the model can calculate least-cost emission pathways for the three gases from the year 2020, while meeting a specified climate

target (e.g., 2 °C warming target) with the lowest total cumulative mitigation costs in terms of the net present value. The model is written in GAMS and numerically solved using CONOPT3 and CONOPT4, the solvers for nonlinear programming or nonlinear optimization problems available in GAMS.

More specifically, ACC2 uses equation (2) to calculate the abatement costs ($ABC$) of years, regions (or global total), and gases.

$$ABC_{t,r,g} = Eb_{t,r,g} \times \int_0^x f_{t,r,g}(x)dx \qquad (2)$$

where $t, r, g$ represent year, region, and gas, respectively. $x$ is the abatement level compared to the baseline scenario. $f_{t,r,g}(x)$ is the MAC function. $Eb$ is the baseline emission level for the IAM. The objective of the model is to minimize the net present value of the total abatement cost ($TABC$) such that the climate target is achieved (e.g., the temperature change is kept below at a certain level such as the 2 °C level), that is:

$$\min TABC = \sum_{t,r,g} \frac{ABC_{t,r,g}}{(1+DSC)^{t-t0}} \qquad (3)$$

where $DSC$ is the discount rate and $t0$ represents the base year used for abatement cost calculations (2010 in this study).

In this study, we replace the existing set of MAC curves in ACC2 with the global and regional MAC curves obtained in this study. We also replace the limits on abatement (i.e., upper limits on abatement levels and their first and second derivatives) with those obtained from this study. We assume a 5% discount rate in the validation tests, a rate commonly

assumed in IAMs (Emmerling et al., 2019), which is also consistent with some of the IAMs analyzed here such as MESSAGE and GET (Figures SI 1.2-1 and 1.2-2 of Riahi et al. (2021)). But we were unable to find the discount rates used in the other IAMs. Note that a 4% discount rate was used as default in recent studies using ACC2 (Tanaka and O'Neill, 2018; Tanaka et al., 2021). We consider the mitigation costs through 2100 in scenario optimizations.

**4.2 Experimental setups for the validation tests**

The emission pathways of ENGAGE IAMs were generated under a series of cumulative carbon budgets (or corresponding carbon price pathways) (Section 2.1). Those of GET were calculated under a series of carbon price pathways (Section 2.2). All these pathways are not directly linked to a temperature target, which is typically used as a constraint for ACC2. Therefore, we successively validated the performance of ACC2-emIAM by applying a constraint first on the cumulative emission budget (Test 1) and then on the global-mean temperature (Tests 2 to 4). Four types of experiments were progressively performed as

summarized in Table 4. Test 1 mimics the condition under which the ENGAGE IAM simulations were carried out (for $CO_2$) and can thus be regarded as a direct validation of MAC curves. Tests 2 to 4 are more applied validations to check how MAC

curves can work with a simple climate model. Tests 2 to 4 can also be seen as applications, rather than validations, of emIAM for temperature targets because the ACC2-emIAM setup takes into account the individual gas characteristics such as the short lifetime of $CH_4$ in deriving least-cost emission pathways, which the original IAM setups do not take into account (i.e., using

GWP100 weighting instead).

**Table 4. Experimental designs of the validation tests for ACC2-emIAM.** See text for details.

|  | Test 1 | Test 2 | Test 3 | Test 4 |
|---|---|---|---|---|
| **Target** | Emission budget | 2100 temperature | 2100 temperature | 2100 temperature Peak temperature |
| **Abatement** | Separately gas by gas | Separately gas by gas | Simultaneously all three gases | Simultaneously all three gases |

- Test 1: Constraint on the cumulative emission budget of each gas. We generate least-cost emission pathways with a cap

on the cumulative emissions of each gas separately (total anthropogenic $CO_2$, $CH_4$, and $N_2O$ emissions for ENGAGE IAMs; energy-related $CO_2$ emissions for GET). The cap on $CO_2$ for an ENGAGE IAM is equal to the cumulative carbon budget as specified in each ENGAGE IAM simulation. The cap on $CO_2$ for GET was calculated from the output of GET, which was simulated under carbon price pathways. The caps on $CH_4$ and $N_2O$ for ENGAGE IAMs were obtained by calculating the respective cumulative emissions from 2019 to 2100. Note that the cumulative $CH_4$ budget, or an emission

budget of short-lived gases in general, does not offer any useful physical interpretation, while the cumulative $CO_2$ budget, or an emission budget of long-lived gases, can be an indicator of the global-mean temperature change (Matthews et al., 2009; Allen et al., 2022). It should also be noted that this experiment does not directly make use of the carbon cycle, atmospheric chemistry, and physical climate modules of ACC2 (i.e., simple climate models), as these modules do not affect the results. Test 1 evaluates how the cumulative emission budget can be distributed over time, which depends on

the MAC curves and the limits on abatement (i.e., upper limits on abatements and their first and second derivatives), while minimizing the total abatement costs.

- Test 2: Constraint on the end-of-century warming for one gas at a time. We first use ACC2 to calculate the temperature pathway from each carbon budget scenario of each IAM. The calculated temperature at the end of the century is used as

a constraint on ACC2-emIAM. This test does not use the temperature data found in the ENGAGE Scenario Explorer, which were calculated using different simple climate models (Xiong et al., 2022). We calculate least-cost emission pathways for only one gas at a time ($CO_2$, $CH_4$, or $N_2O$ for ENGAGE IAMs). For example, when calculating a least-cost emission pathway for $CO_2$, we assume the $CH_4$ and $N_2O$ emissions to follow the respective pathways from the corresponding carbon budget scenario in the ENGAGE Scenario Explorer. This test validates the temporal distribution of emissions under an end-of-century warming target with global MAC curves. It also validates the trade-off among

different regions with regional MAC curves; however, it does not address the trade-off among different gases.

- Test 3: Constraint on the end-of-century warming for three gases simultaneously. This test is the same as Test 2, except that it calculates least-cost emission pathways for three gases simultaneously ($CO_2$, $CH_4$, and $N_2O$ for ENGAGE IAMs). This test validates not only the aspects described for Test 2 but also the trade-off among different gases. Note that we do not use GWP100 in ACC2-emIAM to generate least-cost pathways for $CO_2$, $CH_4$, and $N_2O$. In other words, abatement levels among the three gases are determined directly by the MAC curves without being constrained by GWP100. It is well-known that the use of GWP100 in an IAM leads to a deviation from the cost-effective solution (O'Neill, 2003; Reisinger et al., 2013; van den Berg et al., 2015; Tanaka et al., 2021). Although the deviation is unlikely to be very large, this can be a small source of discrepancy between the original and reproduced pathways.

- Test 4: Constraint on the end-of-century warming and the mid-century peak warming for three gases simultaneously. This test is the same as Test 3, except that the maximum temperature in mid-century is used as an additional constraint on ACC2-emIAM. The peak temperature was taken from the temperature calculation using ACC2 performed for Test 2. The constraint of the mid-century peak warming is intended to control near-term $CH_4$ emissions, which are known to have a strong effect on peak temperatures in mid-century but little effect on end-of-century temperatures (Shoemaker et al., 2013; Sun et al., 2021; McKeough, 2022; Xiong et al., 2022).

There are other technical notes that apply to all four tests above. For PKB scenarios, we impose a condition that prohibits net negative $CO_2$ emissions on ACC2-emIAM. For ECB scenarios (for Test 1 only), we assume that a carbon budget can be interpreted simply as a net budget when it is related to the final temperature through the property of the Transient climate response to cumulative carbon emissions (TCRE), as commonly assumed in the IAM community. It should, however, be noted that such an assumption may not hold for large temperature overshoot scenarios (Tachiiri et al., 2019; Melnikova et al., 2021; Zickfeld et al., 2021; Mastropierro et al., 2023). For scenarios with INDC, which follow NDC up to 2030, we impose the original scenarios up to 2030 and perform the optimization from 2030 onwards. For scenarios without INDC, on the other hand, we perform the optimization starting in 2020. Emissions scenarios for all GHGs and air pollutants other than the three gases are assumed to follow the corresponding scenarios from the ENGAGE Scenario Explore or the most proximate SSP in the case of GET. The original scenarios from GET are available from 2010, but we reproduced the GET scenarios from 2020 (as done for ENGAGE IAMs) and adopted the GET scenarios from 2010 to 2020 in ACC2-emIAM. When a scenario was removed from the MAC curve fitting (Table 1), the scenario was also removed from the validation.

It is important to note that the outcome of the tests described above needs to be interpreted differently, depending on whether the IAM is an intertemporal optimization model or a recursive dynamic model (Table 1) (Babiker et al., 2009; Guivarch and Rogelj, 2017; Melnikov et al., 2021). While the temporal distribution of emission abatement is internally

calculated in an intertemporal optimization model, it is typically a priori assumption in a recursive dynamic model and determined by a given carbon price pathway. In a recursive dynamic model, the underlying economic and energy-related relationships that determine the temporal distribution of emission abatement are not necessarily consistent with those used to allocate emission abatement across sectors and regions at each time step.

## 4.3 Results from the validation tests

Figure 8 provides an overview of the validation results, using REMIND as an example. Overall, ACC2-emIAM has closely reproduced the original $CO_2$ emission pathways from REMIND in the series of four tests. The outcomes for $CH_4$ and $N_2O$ were also generally satisfactory, although not as successful as those for $CO_2$. For Test 1, the results were good for all three gases. The results were similarly good for Test 2, except for a minor discrepancy due to a small rise in emissions at the end of the century. A small increase in emissions is known to occur in ACC2 before a temperature target is reached after an overshoot due to the inertia of the system (Tanaka et al., 2021). However, discrepancies were found in Test 3 for the near-term $CH_4$ pathways in low budget cases and the late-century $CH_4$ and $N_2O$ pathways in high budget cases. The discrepancy for near-term $CH_4$ emissions was reduced in Test 4. $CH_4$ abatements tend to be incentivized later in the century in the cost optimization of ACC2 with the discount rate of 5% (Tanaka et al., 2021). This effect can be offset by the additional constraint on the mid-century peak temperature, as near-term $CH_4$ emissions can strongly influence mid-century temperatures (Shoemaker et al., 2013; Sun et al., 2021; McKeough, 2022; Xiong et al., 2022). When interpreting the validation tests, it is useful to keep in mind that only Test 1 can be strictly considered as a pure validation; certain levels of discrepancies can be expected from Tests 2 to 4 due to the difference in the model setup between the original IAMs and ACC2-emIAM.

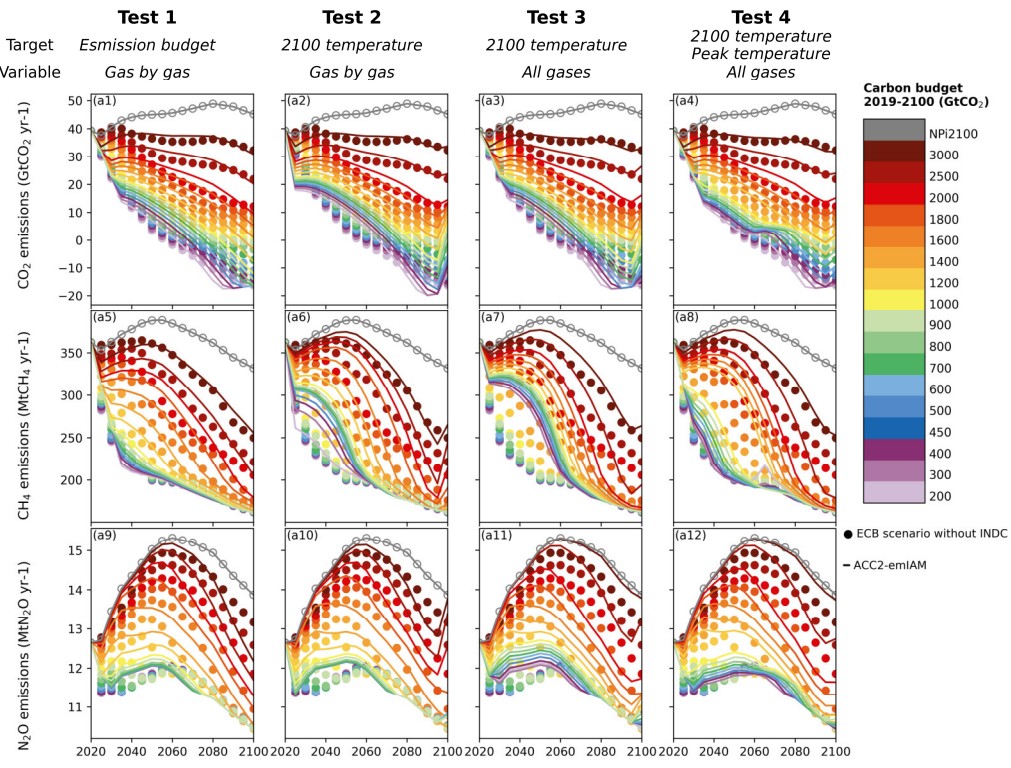

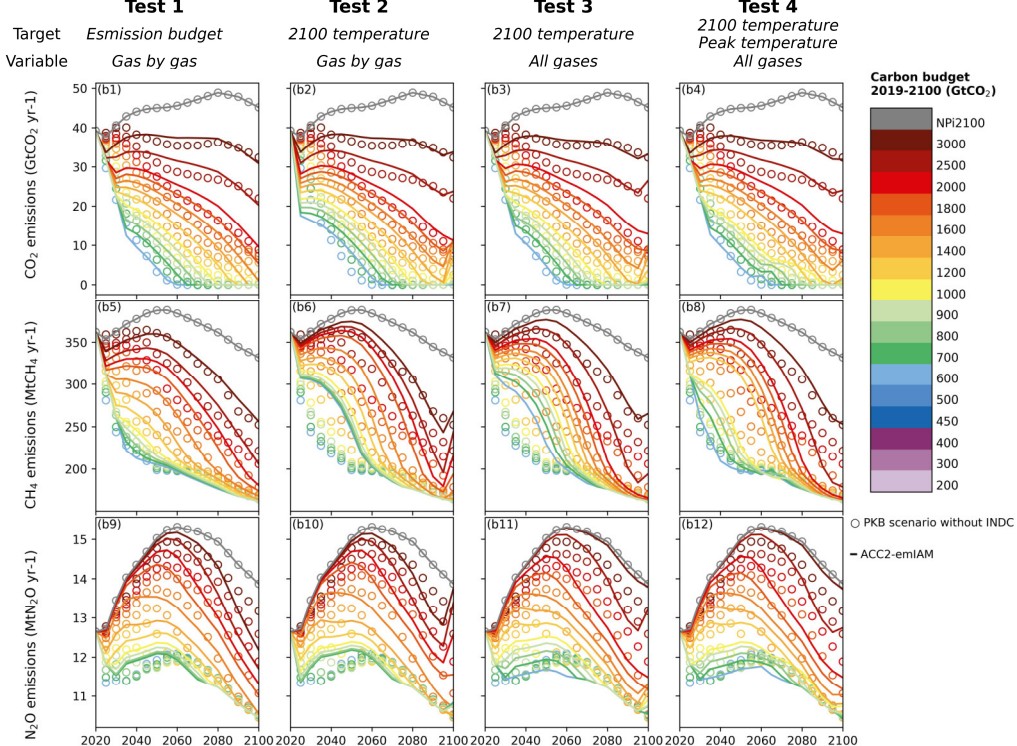

**Figure 8. Overview of the validation results for ACC2-emIAM with REMIND as an example.** The outcomes for ECB scenarios (filled circles) are shown in the upper set of Panels (a1) to (a12); those for PKB scenarios (open circles) are in the lower set of Panels (b1) to (b12). The points show the original emission pathways from REMIND obtained from the ENGAGE Scenario Explorer; the lines show the emission pathways reproduced from ACC2-emIAM. The same color is used for each pair of original and reproduced pathways. For the sake of presentation, only the outcomes of scenarios without INDC are presented; the outcomes of scenarios with INDC are not shown here. The outcomes of the full set of scenarios can be seen in Figure S90.

Figure 9 shows the validation results from Test 4 for all nine ENGAGE IAMs (global total anthropogenic $CO_2$, $CH_4$, and $N_2O$ emissions) and GET with different technology portfolios (global energy-related $CO_2$ emissions). The full set of validation results from Tests 1 to 4 can be found in Figures S91-S109, S129-S147, S167-S184, and S203-S221, respectively. $CO_2$ emission pathways were generally well reproduced through ACC2-emIAM for all ENGAGE IAMs. The outcomes for $CH_4$ and $N_2O$ were not as good as those for $CO_2$: only a subset of ENGAGE IAMs such as REMIND and WITCH was adequately captured by ACC2-emIAM. Some of the mismatches can be explained, for example, by the poor fits of $N_2O$ MAC curves from COFFEE and TIAM (Figure S10). The general difficulty in capturing IMAGE through MAC curves (Figure S16) can be seen in the mismatches in these tests for IMAGE in Figure 9. It is also worth noting that, despite very good fits of MAC curves from GEM (Figure S15), $CH_4$ and $N_2O$ emission pathways were not well reproduced. The results for GET were also generally good, but the Large bioenergy + Small carbon storage portfolio gave a relatively poor result. This may be due to the relatively poor fit of the MAC curve for this technology portfolio, compared to those from other portfolios (Figure 6).

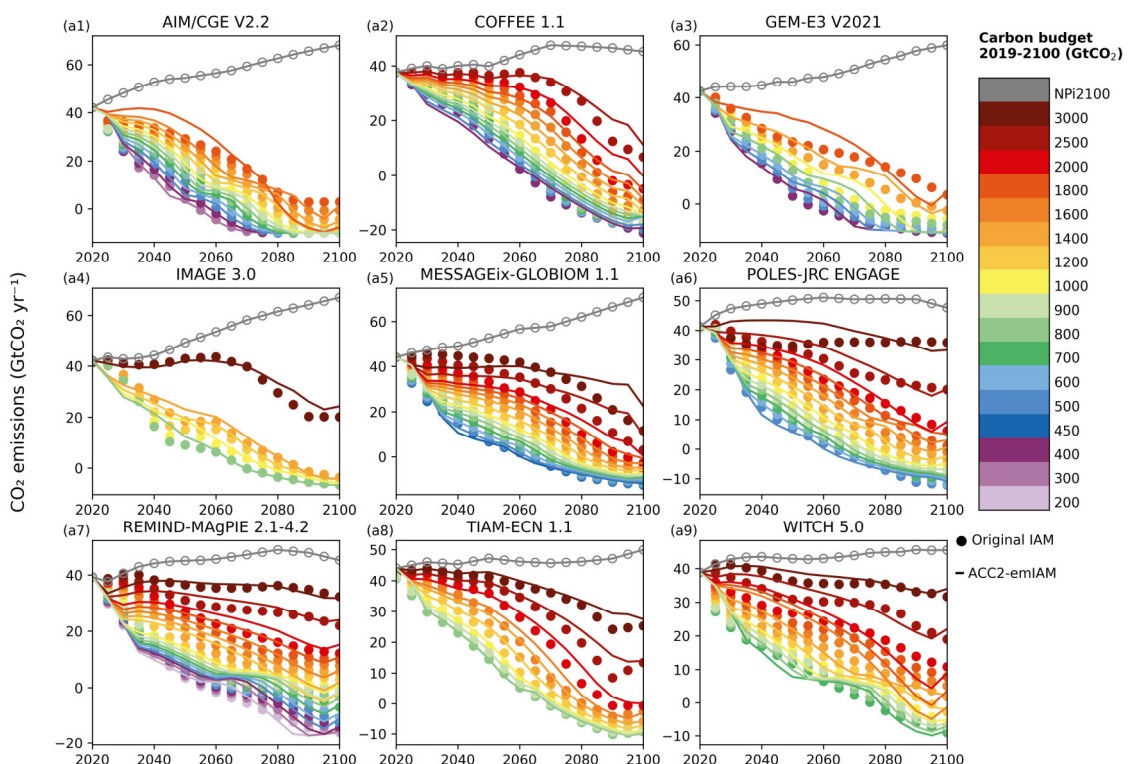

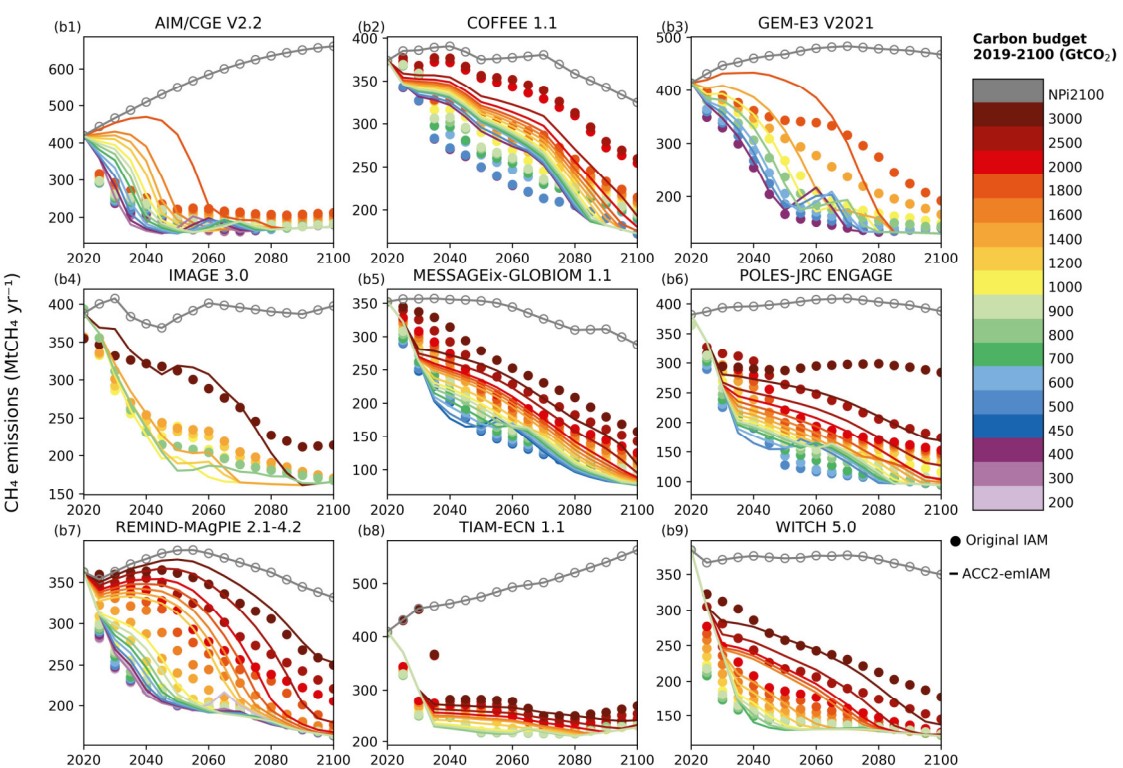

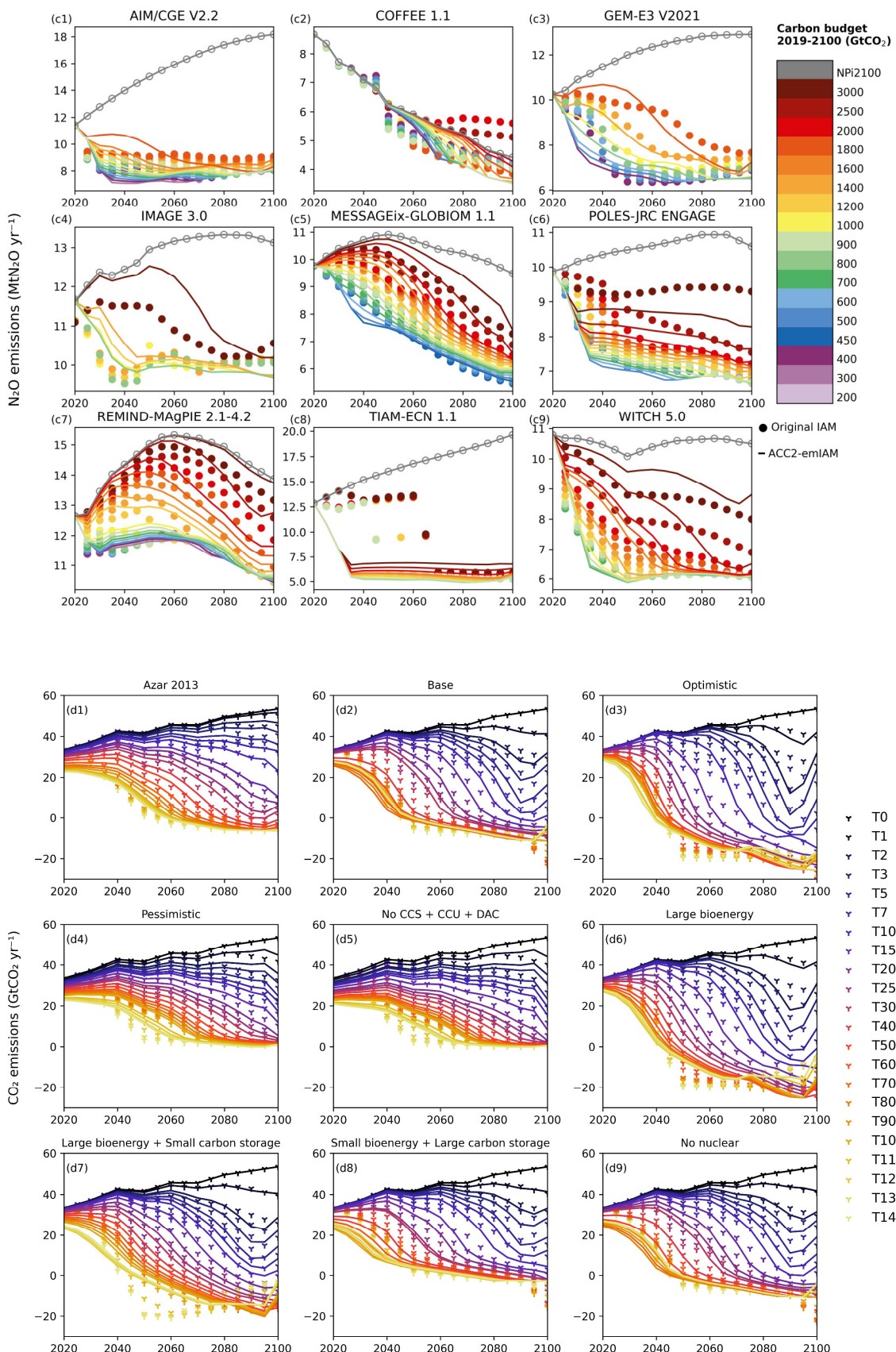

**Figure 9. Original and reproduced global emission pathways from Test 4 for nine ENGAGE IAMs (total anthropogenic CO₂,**

**CH₄, and N₂O emissions) and GET (energy-related CO₂ emissions) with different technology portfolios.** The first three sets of

Panels (a1) to (a9), (b1) to (b9), and (c1) to (c9) are from the nine ENGAGE IAMs for total anthropogenic CO₂, CH₄, and N₂O emissions,

respectively. For the sake of presentation, only the outcomes of ECB scenarios without INDC are presented; those of the full scenarios

can be seen in Figures S204 to S206. The last set of Panels (d1) to (d9) is from GET with different technology portfolios. The points show the original emission pathways from ENGAGE IAMs and GET; the lines show the emission pathways reproduced from ACC2-emIAM. The same color is used for each pair of original and reproduced pathways. For the legend of Panels for GET, the number indicates the initial carbon price (US$2010/tCO$_2$), from which the carbon price grows 5% each year.

Furthermore, we examine several selected features of the original and reproduced emission pathways from Test 4 (ECB scenarios without INDC only), such as CO$_2$ emissions in 2030, 2050, and 2100, cumulative negative CO$_2$ emissions from 2020 to 2100, the year to net zero for CO$_2$, and that for GHG. Figure 10a-c indicates that the reproducibility of CO$_2$ emissions for three different points in time varies across models and carbon budgets, but it is worth noting that ACC2-emIAM nearly consistently overestimates and underestimates 2030 CO$_2$ emissions from AIM and REMIND, respectively. Cumulative negative CO$_2$ emissions are negatively underestimated for COFFEE (Figure 10d), which is related to the general overestimation of 2100 CO$_2$ emissions for COFFEE (Figure 10c). The year to net zero for CO$_2$ tends to be overestimated (later than the original year) for REMIND with the carbon budget at or below 800 GtCO$_2$.

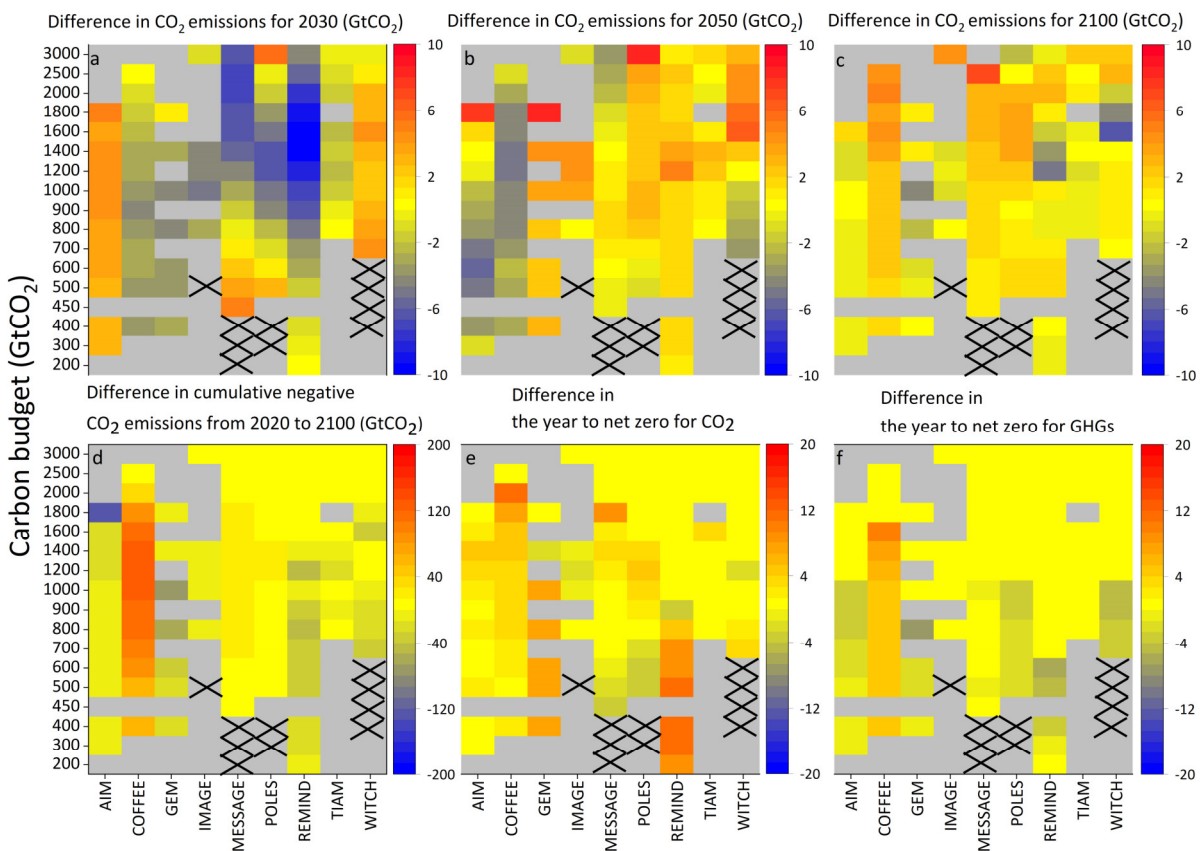

**Figure 10. Differences in the pathway features between ENGAGE IAMs and ACC2-emIAM.** This figure presents the results from Test 4 for ECB scenarios without INDC. Panels (a) to (c) show the difference in CO$_2$ emissions for 2030, 2050, and 2100, respectively, between ENGAGE IAMs and ACC2-emIAM. Panel (d) shows the difference in cumulative negative CO$_2$ emissions. Panels (e) and (f) show the difference in the year to net zero for CO$_2$ and GHG (for CO$_2$, CH$_4$, and N$_2$O), respectively. Positive values indicate that ACC2-emIAM overestimates the pathway feature (i.e., the emulator gives larger emissions (Panels (a) to (c)), less negative cumulative emissions (Panel (d)), or later year (Panels (e) and (f))), while negative values indicate the opposite). Gray boxes without black crosses

indicate that the corresponding scenarios were not available in the ENGAGE Scenario Explorer, while those with black crosses indicate

that the corresponding scenarios were available but not successfully reproduced by ACC2-emIAM (i.e., infeasible solutions).

## 4.4 Statistics of the validation tests

To measure to what extent emission pathways obtained from ACC2-emIAM, denoted as $y$, agree with original pathways from ENGAGE IAMs and GET, denoted as $x$, we calculate the following two different indicators: i) ordinary Pearson's correlation coefficient $r_P$ and ii) Lin's concordance coefficient $r_C$. Each of these indicators is discussed below.

First, because of the prevalent use of $r_P$ and its square form (i.e., coefficient of simple determination, so-called $r^2$) in numerous applications, we use $r_P$ as a reference for comparison, although $r_P$ is known to be inappropriate for testing agreement: it is suited to test the strength of linear relationship, but not the strength of agreement (Bland and Altman, 1986; Cox, 2006). More specifically, $r_P$ (and $r^2$) shows the strength of linear regression line $\acute{y} = \alpha\acute{x} + \beta$, not necessarily $\acute{y} = \acute{x}$, a special case of agreement. Note that it is possible to calculate $r^2$ based on $\acute{y} = \acute{x}$ by using the sum of square of residuals and the total sum of squares (i.e., not equation (2)); however, if $\acute{y} = \acute{x}$ is a very poor regression line, $r^2$ can become negative (page 21 of Hayashi (2000)) and cannot be interpreted as a square of $r_P$. Other arguments that suggest a more restricted use of $r_P$ can be found elsewhere (Ricker, 1973; Laws, 1997; Tanaka and Mackenzie, 2005). For our application, $r_P$ is defined as below.

$$r_P = \frac{\sum_{i=1}^{l}\sum_{j=1}^{m}(x_{i,j}-\bar{x})(y_{i,j}-\bar{y})}{\sqrt{\sum_{i=1}^{l}\sum_{j=1}^{m}(x_{i,j}-\bar{x})^2}\sqrt{\sum_{i=1}^{l}\sum_{j=1}^{m}(y_{i,j}-\bar{y})^2}} \tag{4}$$

where $x_{i,j}$ and $y_{i,j}$ are the original and reproduced emission, respectively, for year $i$ (for $i = 1, \dots, l$) under scenario $j$ (for $j = 1, \dots, m$). $\bar{x}$ and $\bar{y}$ are the mean of $x_{i,j}$ and $y_{i,j}$, respectively, over $i$ and $j$. $r_P$ can change between -1 and 1. When it is 1, the samples have a perfect linear relationship, which is a necessary condition for a perfect agreement. When it is 0, there is no linear relationship in the samples.

Second, $r_C$ is a more appropriate indicator for measuring agreement than $r_P$ (Lin, 1989; Barnhart et al., 2007; Lin et al., 2012). $r_C$ is defined as follows.

$$r_C = \frac{2s_{xy}}{s_x^2 + s_y^2 + (\bar{x}-\bar{y})^2} \tag{5}$$

where $s_x^2$ and $s_y^2$ are the variance of $x_{i,j}$ and $y_{i,j}$, respectively. That is, $s_x^2 = \frac{1}{l\times m}\sum_{i=1}^{l}\sum_{j=1}^{m}(x_{i,j}-\bar{x})^2$ and $s_y^2 = \frac{1}{l\times m}\sum_{i=1}^{l}\sum_{j=1}^{m}(y_{i,j}-\bar{y})^2$, respectively. $s_{xy}$ is the covariance of $x_{i,j}$ and $y_{i,j}$. That is, $s_{xy} = \frac{1}{l\times m}\sum_{i=1}^{l}\sum_{j=1}^{m}(x_{i,j}-\bar{x})(y_{i,j}-\bar{y})$. $r_C$ also distributes between -1 and 1. When it is 1, 0, and -1, it indicates a perfect concordance, no concordance, and a perfect discordance (or reverse concordance), respectively. $r_C$ is commonly interpreted either similar to $r_P$ or in the following way: >0.99, almost perfect; 0.95 to 0.99, substantial; 0.90 to 0.95, moderate; <0.90, poor (Akoglu, 2018). An underlying assumption for this parametric statistic is that the population follows Gaussian distributions.

Two other indicators (i.e., the root-mean-square-error (*RMSE*) and the mean-average-error (*MAE*)) are computed to provide additional insights into the magnitude of the deviations. All four indicators are reported in Figures S110-S128, S148-

**Table 5. Statistical validation of global emission pathways reproduced from ACC2-emIAM with original emission pathways from nine ENGAGE IAMs and GET.** The upper and lower Panels are the results for ENGAGE IAMs (global total anthropogenic $CO_2$, $CH_4$, and $N_2O$ emissions) and GET (global energy-related $CO_2$ emissions), respectively. The table shows two indicators: i) ordinary Pearson's correlation coefficient $r_P$ and ii) Lin's concordance coefficient $r_C$. The higher the value of the indicator is, the darker the color of the cell is. See text for the details of these statistical indicators. This table presents the results from all scenarios. Results only from the ECB scenarios without INDC can be found in Table S5. The results for Test 3 are not reported for GET because Tests 2 and 3 are, by definition, equivalent for GET.

| | | | AIM | COFFEE | GEM | IMAGE | MESSAGE | POLES | REMIND | TIAM | WITCH |
|---|---|---|---|---|---|---|---|---|---|---|---|
| $r_P$ | Test 1 | $CO_2$ | 0.986 | 0.990 | 0.983 | 0.975 | 0.981 | 0.985 | 0.976 | 0.985 | 0.976 |
| | | $CH_4$ | 0.962 | 0.977 | 0.964 | 0.973 | 0.975 | 0.967 | 0.986 | 0.927 | 0.985 |
| | | $N_2O$ | 0.921 | 0.966 | 0.948 | 0.875 | 0.985 | 0.964 | 0.962 | 0.466 | 0.977 |
| | Test 2 | $CO_2$ | 0.984 | 0.994 | 0.982 | 0.979 | 0.984 | 0.987 | 0.962 | 0.997 | 0.973 |
| | | $CH_4$ | 0.667 | 0.941 | 0.805 | 0.890 | 0.960 | 0.954 | 0.941 | 0.951 | 0.962 |
| | | $N_2O$ | 0.908 | 0.970 | 0.777 | 0.875 | 0.983 | 0.969 | 0.962 | 0.531 | 0.981 |
| | Test 3 | $CO_2$ | 0.983 | 0.991 | 0.981 | 0.974 | 0.974 | 0.986 | 0.975 | 0.879 | 0.973 |
| | | $CH_4$ | 0.678 | 0.911 | 0.852 | 0.886 | 0.968 | 0.929 | 0.907 | 0.740 | 0.933 |
| | | $N_2O$ | 0.933 | 0.956 | 0.966 | 0.600 | 0.977 | 0.961 | 0.948 | 0.549 | 0.956 |
| | Test 4 | $CO_2$ | 0.989 | 0.990 | 0.985 | 0.990 | 0.981 | 0.990 | 0.981 | 0.995 | 0.953 |
| | | $CH_4$ | 0.879 | 0.916 | 0.951 | 0.955 | 0.978 | 0.951 | 0.955 | 0.953 | 0.940 |
| | | $N_2O$ | 0.954 | 0.955 | 0.964 | 0.710 | 0.977 | 0.957 | 0.960 | 0.607 | 0.935 |
| $r_C$ | Test 1 | $CO_2$ | 0.981 | 0.985 | 0.981 | 0.974 | 0.980 | 0.985 | 0.976 | 0.983 | 0.968 |
| | | $CH_4$ | 0.957 | 0.977 | 0.960 | 0.968 | 0.974 | 0.966 | 0.986 | 0.927 | 0.984 |
| | | $N_2O$ | 0.916 | 0.964 | 0.946 | 0.863 | 0.981 | 0.963 | 0.962 | 0.385 | 0.975 |
| | Test 2 | $CO_2$ | 0.979 | 0.991 | 0.980 | 0.978 | 0.982 | 0.986 | 0.962 | 0.997 | 0.968 |
| | | $CH_4$ | 0.549 | 0.926 | 0.730 | 0.833 | 0.956 | 0.942 | 0.919 | 0.949 | 0.953 |
| | | $N_2O$ | 0.901 | 0.968 | 0.764 | 0.863 | 0.981 | 0.969 | 0.961 | 0.454 | 0.980 |
| | Test 3 | $CO_2$ | 0.976 | 0.988 | 0.978 | 0.973 | 0.971 | 0.983 | 0.975 | 0.877 | 0.964 |
| | | $CH_4$ | 0.558 | 0.905 | 0.815 | 0.858 | 0.962 | 0.924 | 0.881 | 0.715 | 0.918 |
| | | $N_2O$ | 0.914 | 0.954 | 0.961 | 0.572 | 0.975 | 0.947 | 0.946 | 0.494 | 0.945 |
| | Test 4 | $CO_2$ | 0.987 | 0.987 | 0.982 | 0.989 | 0.978 | 0.987 | 0.981 | 0.993 | 0.949 |
| | | $CH_4$ | 0.852 | 0.911 | 0.934 | 0.930 | 0.964 | 0.933 | 0.944 | 0.946 | 0.935 |
| | | $N_2O$ | 0.935 | 0.954 | 0.956 | 0.697 | 0.972 | 0.940 | 0.960 | 0.482 | 0.920 |

| | GET technology portfolio | Azar2013 | Base | Optimistic | Pessimistic | No_cap | L_bio | L_bio/S_str | S_bio/L_str | No_nc |
|---|---|---|---|---|---|---|---|---|---|---|
| $r_P$ | Test 1 | 0.992 | 0.983 | 0.973 | 0.991 | 0.981 | 0.964 | 0.964 | 0.984 | 0.984 |
| | Test 2 | 0.985 | 0.978 | 0.976 | 0.982 | 0.968 | 0.966 | 0.964 | 0.968 | 0.978 |
| | Test 4 | 0.985 | 0.979 | 0.976 | 0.981 | 0.967 | 0.977 | 0.973 | 0.969 | 0.978 |
| $r_C$ | Test 1 | 0.992 | 0.983 | 0.972 | 0.991 | 0.979 | 0.963 | 0.964 | 0.983 | 0.984 |
| | Test 2 | 0.985 | 0.978 | 0.975 | 0.981 | 0.963 | 0.966 | 0.964 | 0.966 | 0.977 |
| | Test 4 | 0.985 | 0.979 | 0.976 | 0.980 | 0.963 | 0.977 | 0.973 | 0.966 | 0.978 |

**Table 6. Statistical validation of regional emission pathways reproduced from ACC2-emIAM with original emission pathways from five ENGAGE IAMs.** Ordinary Pearson's correlation coefficient $r_P$ and Lin's concordance coefficient $r_C$ are shown in the table. The higher the value of the indicator is, the darker the color of the cell is. This table presents the results from all scenarios. Results only from the ECB scenarios without INDC can be found in Table S6. Emissions from the ROW were not reproduced in some IAMs due to the small emission values.

| | | | Test 1 | | | | | Test 2 | | | | | Test 3 | | | | | Test 4 | | | |
|---|---|---|---|---|---|---|---|---|---|---|---|---|---|---|---|---|---|---|---|---|---|
| | | | AIM | COFFEE | GEM | IMAGE | MESSAGE | AIM | COFFEE | GEM | IMAGE | MESSAGE | AIM | COFFEE | GEM | IMAGE | MESSAGE | AIM | COFFEE | GEM | IMAGE | MESSAGE |
| $r_P$ | $CO_2$ | SUBSAFR | 0.954 | 0.916 | 0.978 | 0.963 | 0.979 | 0.951 | 0.938 | 0.976 | 0.966 | 0.978 | 0.954 | 0.920 | 0.971 | 0.956 | 0.980 | 0.961 | 0.923 | 0.982 | 0.978 | 0.980 |
| | | CHN | 0.974 | 0.978 | 0.978 | 0.966 | 0.990 | 0.970 | 0.947 | 0.976 | 0.965 | 0.983 | 0.975 | 0.982 | 0.969 | 0.957 | 0.986 | 0.982 | 0.983 | 0.980 | 0.984 | 0.989 |
| | | EUWE | 0.979 | 0.849 | 0.974 | 0.973 | 0.994 | 0.977 | 0.869 | 0.974 | 0.979 | 0.982 | 0.984 | 0.865 | 0.975 | 0.974 | 0.988 | 0.982 | 0.865 | 0.973 | 0.988 | 0.990 |
| | | SOUASIA | 0.965 | 0.848 | 0.964 | 0.957 | 0.958 | 0.961 | 0.893 | 0.960 | 0.956 | 0.954 | 0.964 | 0.890 | 0.956 | 0.957 | 0.956 | 0.976 | 0.886 | 0.968 | 0.971 | 0.965 |
| | | LATAME | 0.964 | 0.935 | 0.962 | 0.909 | 0.964 | 0.962 | 0.947 | 0.958 | 0.906 | 0.959 | 0.968 | 0.944 | 0.952 | 0.904 | 0.960 | 0.964 | 0.944 | 0.966 | 0.920 | 0.962 |
| | | MIDEAST | 0.983 | 0.870 | 0.964 | 0.952 | 0.979 | 0.982 | 0.909 | 0.953 | 0.957 | 0.964 | 0.985 | 0.904 | 0.941 | 0.952 | 0.964 | 0.987 | 0.903 | 0.957 | 0.976 | 0.961 |
| | | NORAM | 0.981 | 0.930 | 0.959 | 0.963 | 0.990 | 0.980 | 0.947 | 0.963 | 0.965 | 0.981 | 0.986 | 0.940 | 0.958 | 0.958 | 0.987 | 0.981 | 0.939 | 0.961 | 0.977 | 0.985 |
| | | PACOECD | 0.969 | 0.956 | 0.966 | 0.945 | 0.993 | 0.967 | 0.967 | 0.970 | 0.944 | 0.986 | 0.973 | 0.964 | 0.964 | 0.937 | 0.990 | 0.976 | 0.964 | 0.968 | 0.954 | 0.992 |
| | | REFECO | 0.986 | 0.939 | 0.944 | 0.969 | 0.993 | 0.985 | 0.951 | 0.946 | 0.971 | 0.985 | 0.987 | 0.946 | 0.943 | 0.974 | 0.987 | 0.988 | 0.946 | 0.950 | 0.982 | 0.991 |
| | | OTASIAN | 0.979 | 0.952 | 0.970 | 0.957 | 0.988 | 0.977 | 0.969 | 0.966 | 0.957 | 0.978 | 0.983 | 0.963 | 0.959 | 0.947 | 0.982 | 0.983 | 0.964 | 0.972 | 0.972 | 0.983 |
| | | ROW | 0.772 | 0.881 | 0.922 | 0.873 | 0.000 | 0.767 | 0.873 | 0.875 | 0.883 | 0.000 | 0.790 | 0.886 | 0.882 | 0.870 | 0.000 | 0.802 | 0.887 | 0.935 | 0.888 | 0.000 |
| | $CH_4$ | SUBSAFR | 0.952 | 0.890 | 0.949 | 0.838 | 0.956 | 0.556 | 0.716 | 0.680 | 0.691 | 0.763 | 0.567 | 0.699 | 0.786 | 0.669 | 0.800 | 0.878 | 0.751 | 0.931 | 0.750 | 0.892 |
| | | CHN | 0.980 | 0.988 | 0.962 | 0.984 | 0.980 | 0.792 | 0.970 | 0.783 | 0.921 | 0.973 | 0.800 | 0.969 | 0.858 | 0.942 | 0.982 | 0.932 | 0.975 | 0.958 | 0.972 | 0.989 |
| | | EUWE | 0.935 | 0.989 | 0.920 | 0.923 | 0.940 | 0.695 | 0.973 | 0.758 | 0.770 | 0.927 | 0.693 | 0.970 | 0.794 | 0.828 | 0.927 | 0.884 | 0.974 | 0.879 | 0.927 | 0.930 |
| | | SOUASIA | 0.906 | 0.533 | 0.947 | 0.938 | 0.930 | 0.566 | 0.407 | 0.717 | 0.782 | 0.780 | 0.567 | 0.420 | 0.810 | 0.829 | 0.832 | 0.810 | 0.497 | 0.947 | 0.901 | 0.903 |
| | | LATAME | 0.969 | 0.975 | 0.962 | 0.900 | 0.981 | 0.752 | 0.952 | 0.739 | 0.811 | 0.925 | 0.760 | 0.967 | 0.838 | 0.813 | 0.953 | 0.913 | 0.976 | 0.962 | 0.846 | 0.976 |
| | | MIDEAST | 0.966 | 0.953 | 0.958 | 0.801 | 0.877 | 0.689 | 0.906 | 0.785 | 0.718 | 0.817 | 0.697 | 0.904 | 0.857 | 0.721 | 0.821 | 0.897 | 0.919 | 0.962 | 0.832 | 0.859 |
| | | NORAM | 0.924 | 0.942 | 0.952 | 0.933 | 0.979 | 0.634 | 0.883 | 0.698 | 0.750 | 0.971 | 0.630 | 0.911 | 0.808 | 0.797 | 0.969 | 0.864 | 0.922 | 0.952 | 0.906 | 0.975 |
| | | PACOECD | 0.905 | 0.909 | 0.849 | 0.882 | 0.958 | 0.547 | 0.781 | 0.471 | 0.766 | 0.897 | 0.544 | 0.790 | 0.574 | 0.784 | 0.932 | 0.818 | 0.818 | 0.825 | 0.857 | 0.958 |
| | | REFECO | 0.957 | 0.801 | 0.968 | 0.939 | 0.968 | 0.737 | 0.595 | 0.823 | 0.749 | 0.826 | 0.736 | 0.637 | 0.883 | 0.787 | 0.881 | 0.906 | 0.690 | 0.967 | 0.861 | 0.944 |
| | | OTASIAN | 0.964 | 0.970 | 0.953 | 0.973 | 0.994 | 0.695 | 0.932 | 0.807 | 0.905 | 0.950 | 0.701 | 0.936 | 0.861 | 0.931 | 0.969 | 0.895 | 0.949 | 0.953 | 0.972 | 0.986 |
| | | ROW | 0.895 | 0.000 | 0.945 | 0.000 | 0.000 | 0.665 | 0.000 | 0.776 | 0.000 | 0.000 | 0.672 | 0.000 | 0.836 | 0.000 | 0.000 | 0.782 | 0.000 | 0.947 | 0.000 | 0.000 |
| | $N_2O$ | SUBSAFR | 0.969 | 0.953 | 0.867 | 0.859 | 0.902 | 0.965 | 0.961 | 0.885 | 0.857 | 0.900 | 0.979 | 0.949 | 0.905 | 0.776 | 0.826 | 0.982 | 0.950 | 0.897 | 0.803 | 0.837 |
| | | CHN | 0.982 | 0.933 | 0.967 | 0.971 | 0.952 | 0.977 | 0.934 | 0.967 | 0.970 | 0.945 | 0.984 | 0.945 | 0.980 | 0.977 | 0.933 | 0.982 | 0.942 | 0.983 | 0.983 | 0.931 |
| | | EUWE | 0.924 | 0.967 | 0.828 | 0.851 | 0.971 | 0.908 | 0.965 | 0.805 | 0.839 | 0.970 | 0.951 | 0.971 | 0.810 | 0.703 | 0.970 | 0.958 | 0.971 | 0.795 | 0.767 | 0.974 |
| | | SOUASIA | 0.913 | 0.603 | 0.947 | 0.863 | 0.930 | 0.900 | 0.801 | 0.946 | 0.860 | 0.933 | 0.948 | 0.761 | 0.966 | 0.841 | 0.899 | 0.939 | 0.756 | 0.977 | 0.843 | 0.920 |
| | | LATAME | 0.951 | 0.807 | 0.953 | 0.068 | 0.930 | 0.940 | 0.810 | 0.954 | 0.090 | 0.920 | 0.968 | 0.801 | 0.969 | -0.153 | 0.906 | 0.957 | 0.801 | 0.973 | -0.122 | 0.899 |
| | | MIDEAST | 0.874 | 0.735 | 0.963 | 0.984 | 0.935 | 0.859 | 0.742 | 0.963 | 0.983 | 0.936 | 0.929 | 0.684 | 0.971 | 0.990 | 0.946 | 0.922 | 0.691 | 0.968 | 0.992 | 0.954 |
| | | NORAM | 0.915 | 0.899 | 0.874 | 0.864 | 0.930 | 0.898 | 0.899 | 0.899 | 0.863 | 0.933 | 0.948 | 0.921 | 0.925 | 0.807 | 0.946 | 0.948 | 0.919 | 0.924 | 0.843 | 0.947 |
| | | PACOECD | 0.848 | 0.811 | 0.873 | 0.038 | 0.971 | 0.811 | 0.808 | 0.906 | 0.013 | 0.963 | 0.916 | 0.735 | 0.913 | -0.106 | 0.954 | 0.925 | 0.729 | 0.917 | -0.023 | 0.950 |
| | | REFECO | 0.926 | 0.631 | 0.389 | 0.430 | 0.774 | 0.910 | 0.630 | 0.452 | 0.431 | 0.780 | 0.956 | 0.521 | 0.518 | 0.207 | 0.852 | 0.956 | 0.522 | 0.531 | 0.263 | 0.830 |
| | | OTASIAN | 0.925 | 0.725 | 0.933 | 0.739 | 0.953 | 0.908 | 0.727 | 0.935 | 0.737 | 0.953 | 0.955 | 0.673 | 0.949 | 0.532 | 0.962 | 0.952 | 0.676 | 0.939 | 0.571 | 0.966 |
| | | ROW | 0.000 | 0.000 | 0.862 | 0.000 | 0.000 | 0.000 | 0.000 | 0.879 | 0.000 | 0.000 | 0.000 | 0.000 | 0.895 | 0.000 | 0.000 | 0.000 | 0.000 | 0.920 | 0.000 | 0.000 |
| $r_C$ | $CO_2$ | SUBSAFR | 0.945 | 0.913 | 0.976 | 0.959 | 0.978 | 0.941 | 0.936 | 0.972 | 0.980 | 0.971 | 0.949 | 0.913 | 0.962 | 0.953 | 0.977 | 0.958 | 0.915 | 0.972 | 0.978 | 0.974 |
| | | CHN | 0.963 | 0.972 | 0.976 | 0.964 | 0.989 | 0.957 | 0.980 | 0.972 | 0.978 | 0.983 | 0.968 | 0.972 | 0.958 | 0.954 | 0.983 | 0.976 | 0.973 | 0.972 | 0.980 | 0.986 |
| | | EUWE | 0.973 | 0.827 | 0.941 | 0.973 | 0.994 | 0.971 | 0.851 | 0.939 | 0.982 | 0.980 | 0.981 | 0.828 | 0.955 | 0.972 | 0.986 | 0.982 | 0.828 | 0.951 | 0.986 | 0.987 |
| | | SOUASIA | 0.958 | 0.737 | 0.957 | 0.957 | 0.952 | 0.953 | 0.765 | 0.953 | 0.986 | 0.936 | 0.959 | 0.774 | 0.941 | 0.956 | 0.950 | 0.972 | 0.769 | 0.951 | 0.971 | 0.956 |
| | | LATAME | 0.958 | 0.902 | 0.951 | 0.898 | 0.952 | 0.956 | 0.914 | 0.945 | 0.962 | 0.953 | 0.964 | 0.908 | 0.933 | 0.891 | 0.940 | 0.963 | 0.907 | 0.949 | 0.912 | 0.945 |
| | | MIDEAST | 0.978 | 0.812 | 0.961 | 0.951 | 0.974 | 0.977 | 0.845 | 0.948 | 0.997 | 0.949 | 0.983 | 0.847 | 0.924 | 0.951 | 0.962 | 0.986 | 0.845 | 0.939 | 0.974 | 0.954 |
| | | NORAM | 0.976 | 0.909 | 0.957 | 0.962 | 0.985 | 0.974 | 0.928 | 0.961 | 0.968 | 0.971 | 0.983 | 0.917 | 0.952 | 0.956 | 0.985 | 0.981 | 0.914 | 0.953 | 0.976 | 0.979 |
| | | PACOECD | 0.962 | 0.940 | 0.958 | 0.943 | 0.993 | 0.959 | 0.951 | 0.963 | 0.941 | 0.985 | 0.968 | 0.942 | 0.957 | 0.935 | 0.988 | 0.975 | 0.941 | 0.964 | 0.952 | 0.988 |
| | | REFECO | 0.982 | 0.905 | 0.940 | 0.961 | 0.993 | 0.980 | 0.916 | 0.941 | 0.957 | 0.983 | 0.985 | 0.908 | 0.932 | 0.965 | 0.986 | 0.987 | 0.907 | 0.941 | 0.978 | 0.989 |
| | | OTASIAN | 0.975 | 0.935 | 0.968 | 0.953 | 0.987 | 0.972 | 0.952 | 0.965 | 0.971 | 0.977 | 0.982 | 0.939 | 0.954 | 0.941 | 0.978 | 0.983 | 0.939 | 0.968 | 0.968 | 0.977 |
| | | ROW | 0.705 | 0.534 | 0.891 | 0.561 | 0.000 | 0.699 | 0.518 | 0.863 | 0.953 | 0.000 | 0.746 | 0.557 | 0.866 | 0.580 | 0.000 | 0.751 | 0.554 | 0.931 | 0.591 | 0.000 |
| | $CH_4$ | SUBSAFR | 0.944 | 0.885 | 0.942 | 0.793 | 0.953 | 0.512 | 0.681 | 0.537 | 0.621 | 0.653 | 0.514 | 0.692 | 0.727 | 0.628 | 0.761 | 0.876 | 0.745 | 0.921 | 0.625 | 0.861 |
| | | CHN | 0.973 | 0.986 | 0.955 | 0.978 | 0.980 | 0.681 | 0.962 | 0.666 | 0.846 | 0.962 | 0.684 | 0.965 | 0.810 | 0.898 | 0.981 | 0.912 | 0.971 | 0.943 | 0.957 | 0.983 |
| | | EUWE | 0.913 | 0.989 | 0.824 | 0.914 | 0.932 | 0.552 | 0.959 | 0.757 | 0.686 | 0.924 | 0.536 | 0.959 | 0.761 | 0.784 | 0.919 | 0.842 | 0.967 | 0.740 | 0.903 | 0.914 |
| | | SOUASIA | 0.894 | 0.351 | 0.938 | 0.927 | 0.910 | 0.428 | 0.239 | 0.595 | 0.682 | 0.662 | 0.424 | 0.262 | 0.755 | 0.783 | 0.760 | 0.772 | 0.315 | 0.929 | 0.848 | 0.862 |
| | | LATAME | 0.961 | 0.972 | 0.955 | 0.877 | 0.981 | 0.630 | 0.934 | 0.614 | 0.724 | 0.886 | 0.629 | 0.954 | 0.783 | 0.757 | 0.941 | 0.889 | 0.966 | 0.948 | 0.787 | 0.973 |
| | | MIDEAST | 0.960 | 0.950 | 0.953 | 0.766 | 0.857 | 0.566 | 0.897 | 0.693 | 0.638 | 0.750 | 0.570 | 0.902 | 0.821 | 0.656 | 0.762 | 0.872 | 0.917 | 0.952 | 0.774 | 0.819 |
| | | NORAM | 0.903 | 0.934 | 0.949 | 0.919 | 0.969 | 0.492 | 0.792 | 0.588 | 0.678 | 0.971 | 0.474 | 0.887 | 0.766 | 0.759 | 0.960 | 0.821 | 0.905 | 0.941 | 0.850 | 0.958 |
| | | PACOECD | 0.888 | 0.907 | 0.841 | 0.841 | 0.957 | 0.409 | 0.709 | 0.413 | 0.697 | 0.859 | 0.394 | 0.774 | 0.568 | 0.724 | 0.925 | 0.776 | 0.802 | 0.799 | 0.749 | 0.939 |
| | | REFECO | 0.947 | 0.721 | 0.965 | 0.919 | 0.962 | 0.615 | 0.444 | 0.733 | 0.621 | 0.740 | 0.607 | 0.521 | 0.851 | 0.700 | 0.834 | 0.878 | 0.585 | 0.958 | 0.829 | 0.926 |
| | | OTASIAN | 0.955 | 0.966 | 0.948 | 0.969 | 0.993 | 0.564 | 0.921 | 0.728 | 0.844 | 0.910 | 0.562 | 0.934 | 0.830 | 0.899 | 0.948 | 0.866 | 0.947 | 0.935 | 0.956 | 0.977 |
| | | ROW | 0.767 | 0.000 | 0.923 | 0.000 | 0.000 | 0.452 | 0.000 | 0.707 | 0.000 | 0.000 | 0.461 | 0.000 | 0.776 | 0.000 | 0.000 | 0.675 | 0.000 | 0.868 | 0.000 | 0.000 |
| | $N_2O$ | SUBSAFR | 0.950 | 0.768 | 0.857 | 0.832 | 0.854 | 0.952 | 0.806 | 0.874 | 0.839 | 0.847 | 0.971 | 0.944 | 0.903 | 0.768 | 0.709 | 0.971 | 0.943 | 0.897 | 0.796 | 0.709 |
| | | CHN | 0.969 | 0.925 | 0.961 | 0.958 | 0.926 | 0.963 | 0.928 | 0.960 | 0.958 | 0.907 | 0.970 | 0.934 | 0.978 | 0.961 | 0.879 | 0.971 | 0.930 | 0.983 | 0.972 | 0.876 |
| | | EUWE | 0.896 | 0.919 | 0.693 | 0.835 | 0.970 | 0.871 | 0.908 | 0.685 | 0.824 | 0.969 | 0.939 | 0.889 | 0.660 | 0.684 | 0.964 | 0.946 | 0.889 | 0.630 | 0.757 | 0.967 |
| | | SOUASIA | 0.910 | 0.545 | 0.939 | 0.834 | 0.909 | 0.898 | 0.756 | 0.938 | 0.837 | 0.915 | 0.920 | 0.615 | 0.963 | 0.817 | 0.888 | 0.920 | 0.611 | 0.975 | 0.820 | 0.914 |
| | | LATAME | 0.939 | 0.618 | 0.948 | 0.067 | 0.903 | 0.927 | 0.625 | 0.948 | 0.089 | 0.885 | 0.949 | 0.597 | 0.967 | -0.152 | 0.840 | 0.941 | 0.597 | 0.972 | -0.121 | 0.832 |
| | | MIDEAST | 0.873 | 0.726 | 0.957 | 0.979 | 0.910 | 0.858 | 0.729 | 0.956 | 0.980 | 0.911 | 0.904 | 0.530 | 0.968 | 0.987 | 0.931 | 0.901 | 0.547 | 0.966 | 0.990 | 0.940 |
| | | NORAM | 0.891 | 0.873 | 0.871 | 0.854 | 0.913 | 0.870 | 0.879 | 0.896 | 0.853 | 0.913 | 0.916 | 0.911 | 0.925 | 0.795 | 0.938 | 0.919 | 0.911 | 0.920 | 0.836 | 0.940 |
| | | PACOECD | 0.836 | 0.800 | 0.830 | 0.024 | 0.958 | 0.791 | 0.795 | 0.879 | 0.008 | 0.942 | 0.901 | 0.667 | 0.879 | -0.069 | 0.916 | 0.910 | 0.659 | 0.881 | -0.014 | 0.913 |
| | | REFECO | 0.901 | 0.484 | 0.354 | 0.370 | 0.691 | 0.877 | 0.464 | 0.414 | 0.371 | 0.691 | 0.947 | 0.280 | 0.485 | 0.174 | 0.811 | 0.947 | 0.280 | 0.509 | 0.231 | 0.788 |
| | | OTASIAN | 0.913 | 0.610 | 0.927 | 0.700 | 0.941 | 0.895 | 0.618 | 0.929 | 0.698 | 0.940 | 0.932 | 0.514 | 0.943 | 0.497 | 0.951 | 0.932 | 0.517 | 0.931 | 0.545 | 0.954 |
| | | ROW | 0.000 | 0.000 | 0.806 | 0.000 | 0.000 | 0.000 | 0.000 | 0.840 | 0.000 | 0.000 | 0.000 | 0.000 | 0.833 | 0.000 | 0.000 | 0.000 | 0.000 | 0.866 | 0.000 | 0.000 |

The statistics of the validation tests for global MAC curves are shown in Table 5. Those for regional MAC curves are in Table 6. The values of $r_C$ are generally lower than the corresponding values of $r_P$, as expected. Reproducibility is generally higher for $CO_2$ than for $CH_4$ and $N_2O$. Certain models tend to have higher values for such indicators than other models. In the global case, AIM tends to show relatively low values for $CH_4$. IMAGE and TIAM tend to show low values for $N_2O$. In the

680 regional results, these models give similar values for $CO_2$ for all Tests. The outcomes for $CH_4$ and $N_2O$ are diverse and difficult to generalize. Finally, ROW is marked with low values in many models and from most of the Tests.

## 5. On the time-dependency of MAC curves

### 5.1 Deriving time-dependent MAC curves: transitional and free-fitting approaches

While the time-independent assumption of MAC curves is key to simplifying our IAM emulation approach, it raises questions
about what this simplification entails. Here, we test time-dependent MAC curves to better understand the implications of our time-independent approach. Of ten IAMs analyzed in our paper, we selected three IAMs (AIM, POLES, and WITCH) for such a test because, based on our visual inspection, these models provide data that appear to be suitable for the use of time-dependent MAC curves (Figure 11). As detailed below, we developed time-dependent MAC curves using two different methods.

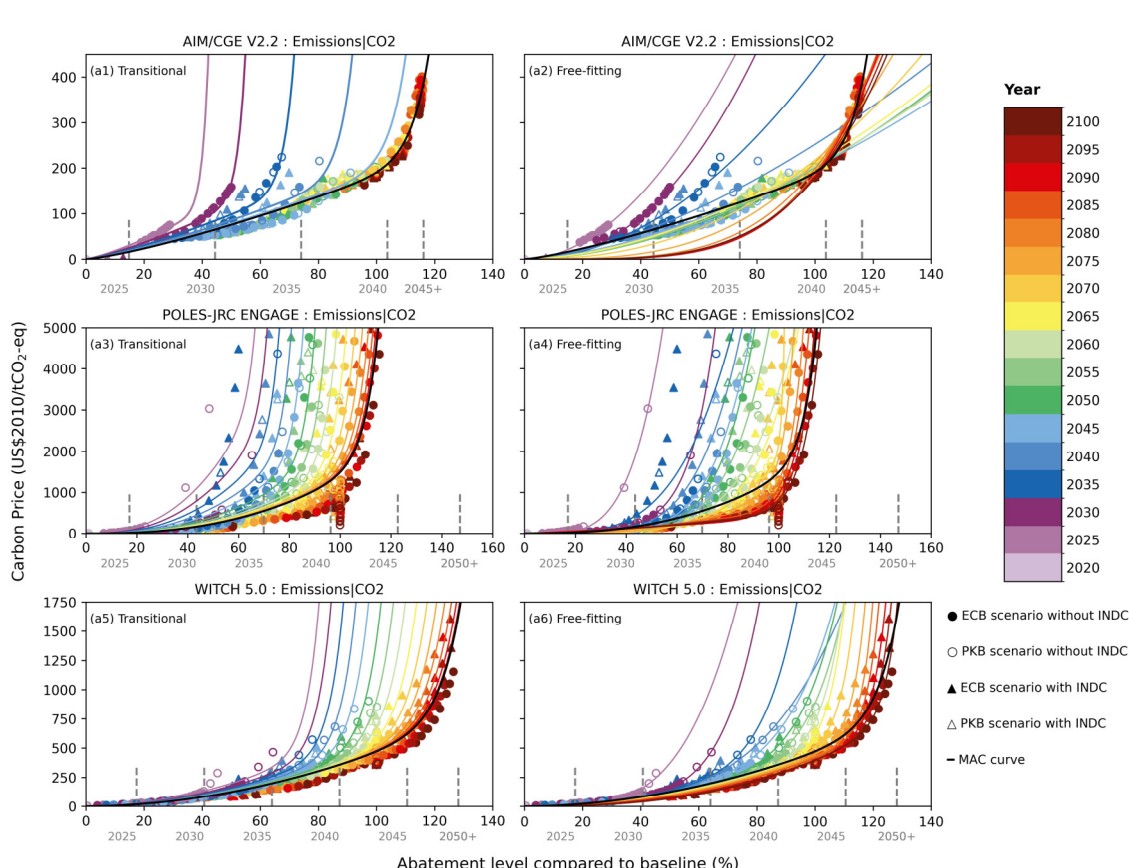

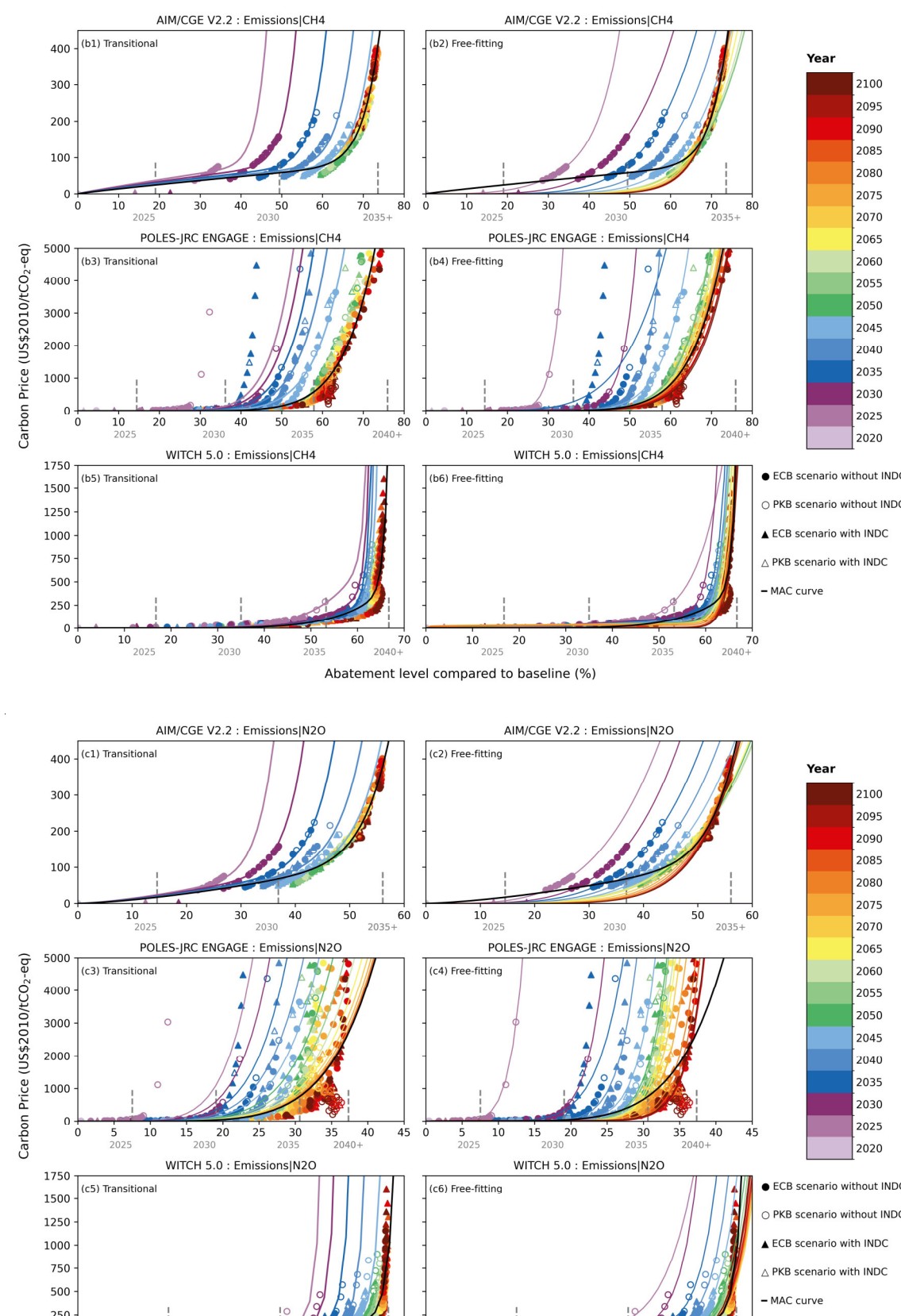

**Figure 11. CO₂, CH₄, and N₂O abatement levels and carbon prices from three IAMs (AIM, POLES, and WITCH) and their time-independent (in black) and transitional and free-fitting time-dependent MAC curves (in chromatic colors).** Panels (a1) to (a6), (b1) to (b6), and (c1) to (c6) show the MAC curves for CO₂, CH₄, and N₂O, respectively. In each set of Panels, data from the three IAMs are presented. Time-independent MAC curves are shown in black lines. Transitional time-dependent MAC curves are in

chromatic color lines on the left column; free-fitting time-dependent MAC curves are in chromatic color lines on the right column. The vertical gray bars indicate the maximum abatement levels that can be potentially achieved at each point in time every five years (gray text), as determined by the upper limits of the first and second derivatives of abatement changes, as well as the upper limit of the abatement level (Table 2). See Table 8 for the goodness of fit (coefficients of simple determination) for the time-independent and time-dependent MAC curves.

First, we introduced the time-dependency to the MAC curves in a way that smoothly extends the time-independent MAC curves and their parameterizations as originally used, referred to as "*transitional* time-dependent MAC curves" (left column of Figure 11). For AIM, the relationships between the relative abatement levels of $CO_2$, $CH_4$, and $N_2O$ and the carbon price are adequately captured by the time-independent MAC curves from 2050 onwards. It is thus sufficient to introduce the time-dependency to the MAC curve only before 2050. Namely, we modified the time-independent functional form by introducing time-dependent terms so that the MAC curves can be shifted to the left (or shifted up) as we go back in time from 2050. Regarding the two other IAMs, we also applied the same approach to $CH_4$ from POLES and $CH_4$ and $N_2O$ from WITCH. For the remaining cases (i.e., $CO_2$ and $N_2O$ data from POLES and $CO_2$ from WITCH), on the other hand, we stretched the time-dependent MAC curve approach all the way to 2100, as it is evident that the data show a temporary shifting trend until 2100.

Hence, we extended the time-dependent MAC curve approach either to 2050 or to 2100, based on the visual inspection of the data for the relationship between the abatement level and the carbon price from each model and gas. For time-dependent MAC curves that shift until 2050, we used the following functional form for each applicable model and gas.

$$f(x_t) = \begin{cases} a \times (x_t)^b + c \times (x_t)^d, 2050 \leq t \leq 2100 \\ a \times (x_t \times (1 + e1 \times (t0 - t)^{e2}))^b + c \times (x_t \times (1 + f1 \times (t0 - t)^{f2}))^d, 2025 \leq t < 2050, t0 = 2050 \end{cases} \quad (6)$$

From 2050 onwards, the equation above (including the parameter values) is equivalent to the time-independent MAC curve originally used for the respective model and gas. Although the time-independent MAC curves are derived using the data for the full period since 2025, outliers in the near term have been removed (Figure 2). As a result, the time-independent MAC curves are largely representative of the data for 2050-2100. For time-dependent MAC curves till 2100, we used the following functional form.

$$f(x_t) = a \times (x_t \times (1 + e1 \times (t0 - t)^{e2}))^b + c \times (x_t \times (1 + f1 \times (t0 - t)^{f2}))^d, 2025 \leq t \leq 2100, t0 = 2100 \quad (7)$$

$x_t$ in equations (6) and (7) is the variable representing the emission abatement level in percentage relative to the assumed baseline level at each point in time $t$. $a, b, c, d$ are the parameters that take the model- and gas-specific values estimated for the respective time-independent MAC curve (Table 2). To represent the time-dependency, we basically shift the MAC curves horizontally by introducing the new terms using the parameters $e1, e2, f1, f2$. We optimized the parameters $e1, e2, f1, f2$ by minimizing the squared deviations from the original price-quantity data between 2025 and 2045 (for equations (6)) or between 2025 and 2095 (for equations (7)) for each model and gas (Table 7). Note that for AIM, $e2$ and $f2$ are assumed to be 2 for the

sake of simplicity (they are optimized for POLES and WITCH), while $e1$ and $f1$ are optimized for all three IAMs.

**Table 7. Values of additional parameters used in the transitional time-dependent MAC curves for the three IAMs.** For the definitions of time-dependent ranges and parameters, see equations (6) and (7) and the related text.

| IAM | Gas | Time-dependent range | Parameter | | | |
|-----|-----|------|------|------|------|------|
| | | | $e1$ | $e2$ | $f1$ | $f2$ |
| AIM | $CO_2$ | Up to 2050 | $9.991 \times 10^{-4}$ | 2.000 | $2.974 \times 10^{-3}$ | 2.000 |
| | $CH_4$ | Up to 2050 | $9.684 \times 10^{-4}$ | 2.000 | $9.610 \times 10^{-4}$ | 2.000 |
| | $N_2O$ | Up to 2050 | $4.099 \times 10^{-4}$ | 2.000 | $9.593 \times 10^{-4}$ | 2.000 |
| POLES | $CO_2$ | Up to 2100 | $8.580 \times 10^{-8}$ | $3.794 \times 10^{0}$ | $4.554 \times 10^{-5}$ | $2.229 \times 10^{0}$ |
| | $CH_4$ | Up to 2050 | $6.353 \times 10^{-2}$ | $6.276 \times 10^{-1}$ | 0.000 | 0.000 |
| | $N_2O$ | Up to 2100 | $1.609 \times 10^{-7}$ | $3.541 \times 10^{0}$ | 0.000 | 0.000 |
| WITCH | $CO_2$ | Up to 2100 | $1.091 \times 10^{-10}$ | $5.038 \times 10^{0}$ | $1.369 \times 10^{-4}$ | $1.953 \times 10^{0}$ |
| | $CH_4$ | Up to 2050 | $6.854 \times 10^{-8}$ | $4.573 \times 10^{0}$ | $1.851 \times 10^{-2}$ | $4.161 \times 10^{-1}$ |
| | $N_2O$ | Up to 2050 | $1.291 \times 10^{-4}$ | $2.390 \times 10^{0}$ | $6.551 \times 10^{-3}$ | $1.192 \times 10^{0}$ |

The transitional time-dependent MAC curves generally well captured the temporary shifting data from the three IAMs,

compared to the time-independent MAC curves. The time-dependent MAC curves maintain shapes comparable to the original time-independent MAC curves and, as the time goes, converge to respective time-independent MAC curves either in 2050 or 2100.

Second, in contrast to the transitional approach discussed above, we also introduced the time-dependency to the MAC curves by optimizing the parameters in the functions of the MAC curves at each time step, referred to as "*free-fitting* time-

740 dependent MAC curves" (right column of Figure 11). More specifically, we maintained the functional form used for the time-independent MAC curves and optimized the four parameters $a, b, c, d$ at each time step (every five years from 2025 to 2100) for each IAM (AIM, POLES, and WITCH) and for each gas ($CO_2$, $CH_4$, and $N_2O$). The free-fitting approach captures the data point as closely as possible at each time step, testing the limit of the time-dependent MAC curves approach, while the transitional approach is more suited for applications as an emulator, as the underlying parameterization is simpler for

implementation. The goodness of fit in terms of the coefficient of simple determination ($r^2$) is summarized for each case in Table 8.

**Table 8. Coefficients of simple determination ($r^2$) of the time-independent and time-dependent MAC curves to the IAM data for the relationship between the abatement level and the carbon price.** The dark blue indicates the highest $r^2$ value and the light blue

the next highest $r^2$ value. See Figure 11 for the MAC curves and IAM data.

| Gas | Type of MAC curve | IAM | | |
|-----|-----|------|------|------|
| | | AIM | POLES | WITCH |
| $CO_2$ | Time-independent | 0.957 | 0.466 | 0.909 |
| | Time-dependent (transitional) | 0.978 | 0.711 | 0.957 |
| | Time-dependent (free-fitting) | 0.971 | 0.812 | 0.989 |

| | | | | | |
|---|---|---|---|---|---|
| | Time-independent | 0.941 | 0.739 | 0.723 |
| CH$_4$ | Time-dependent (transitional) | 0.980 | 0.857 | 0.740 |
| | Time-dependent (free-fitting) | 0.993 | 0.937 | 0.818 |
| | Time-independent | 0.952 | 0.379 | 0.757 |
| N$_2$O | Time-dependent (transitional) | 0.981 | 0.608 | 0.790 |
| | Time-dependent (free-fitting) | 0.991 | 0.816 | 0.774 |

The $r^2$ values from free-fitting time-dependent MAC curves are generally higher than those from transitional time-dependent MAC curves (seven out of the nine cases). For example, near-term data points from WITCH for $CO_2$ are better captured by the free-fitting time-dependent MAC curves than by the transitional time-dependent MAC curves (Panels (a5) and (a6) of Figure 11). On the other hand, the transitional time-dependent MAC curves are more consistent in terms of the way the MAC curves shift over time, as the underlying mathematical functions are formulated to yield such results. The free-fitting time-dependent MAC curves are less consistent because they are more strongly influenced by diverging data points from different scenario assumptions (i.e., end-of-century budget and peak budget; with and without INDC) (for example, Panels (a3) and (a4) of Figure 11).

## 5.2 Reproducing the IAM scenarios with the time-dependent emulator: methods

Now we implement the transitional and free-fitting time-dependent MAC curves to emIAM. For each carbon budget pathway of each IAM, we imposed the same remaining carbon budget to emIAM as a constraint and calculated the least-cost pathway for $CO_2$. Our focus here is on $CO_2$ because of its greatest relevance. This approach is equivalent to Test 1 discussed in Section 4 and is the most direct and simplest way to evaluate the performance of MAC curves, among other Tests in Section 4. In this set of experiments, our emulator derives $CO_2$ emission pathways in the same way as a subset of IAMs: intertemporal optimization models using a remaining carbon budget as the constraint (Table 1).

We also performed an additional set of experiments by prescribing the carbon price pathway directly to emIAM (i.e., without endogenously optimizing it) and calculated the $CO_2$ emission pathway. This is an even more direct way to test the MAC curves than the carbon budget experiments discussed above. The prescribed carbon price pathway uniquely determines the $CO_2$ emission pathway through the MAC curve(s) without any optimization involved (the carbon budget constraints and the change rate and inertia limits for abatement are irrelevant here). Thus, any deviation from the original $CO_2$ emission pathway can be ascribed to the misfit of the MAC curve(s) to the underlying data from the IAM, while in the previous experiments, it can also be ascribed to a deviation of the endogenously optimized carbon price pathway from the original carbon price pathway of the IAM. In this set of experiments, our emulator derives $CO_2$ emission pathways in the same way as another subset of IAMs: recursive dynamic models using a carbon price pathway (exogenously computed from the remaining carbon budget) as the constraint.

We further checked the sensitivity regarding the upper limits of the first and second derivatives of abatement changes (Table 2). The same upper limits are applied to time-independent and time-dependent approaches. These limits can affect the

experiments to test the MAC curves, as they define the segment of MAC curves that can be utilized at each time step (vertical gray bars in Figure 11). That is, in the near term, only a low range of MAC curves can be utilized by emIAM due to the first and second derivative limits.

In sum, we have a total of nine experimental Cases for each IAM as summarized in Table 9. The first three Cases A to C test the extent to which the $CO_2$ emission pathways of each IAM can be reproduced by emIAM under the corresponding carbon budget constraints by using the respective three different types of MAC curves and abatement limits, while also optimizing the carbon price pathways (our default setting). The next three Cases D to F are the same, except that the abatement limits are not used. The last three Cases G to I provide the corresponding tests under the carbon price constraints, instead of the carbon budget constraints. Note that we focus on the ECB scenarios without INDC, among other sets of scenarios. This set of scenarios provides the cleanest data for testing how well the MAC curves reproduce the original scenarios because these scenarios are free of constraints for net-zero emissions and INDC target levels, which cannot be captured by MAC curves.

**Table 9. Statistical validations of $CO_2$ emission pathways reproduced from emIAM against the original emission pathways from the three IAMs.** For the type of MAC curve, "Indepnd." indicates time-independent MAC curve (default), "Depnd./Trans." transitional time-dependent MAC curve, and "Depnd./Free" free-fitting time-dependent MAC curve. For the abatement limits, "Incl." means that the upper limits of the first and second derivatives of abatement changes are included in emIAM (default); "Excl." indicates otherwise. For the carbon price, "Opt." indicates that the carbon price is endogenously optimized in emIAM (default); "Presc." indicates that the carbon price from the original IAM is prescribed to emIAM. Dark blue indicates the highest value; light blue the next highest value. The table shows the results for the ECB scenarios without INDC.

| Experimental case | | A | B | C | D | E | F | G | H | I |
|---|---|---|---|---|---|---|---|---|---|---|
| Type of MAC curve | | Indepnd. | Depnd./Trans. | Depnd./Free | Indepnd. | Depnd./Trans. | Depnd./Free | Indepnd. | Depnd./Trans. | Depnd./Free |
| Abatement limits | | Incl. | Incl. | Incl. | Excl. | Excl. | Excl. | Excl. | Excl. | Excl. |
| Carbon price | | Opt. | Opt. | Opt. | Opt. | Opt. | Opt. | Presc. | Presc. | Presc. |
| AIM | $r_P$ | 0.9859 | 0.9757 | 0.9821 | 0.9856 | 0.9758 | 0.9858 | 0.9784 | 0.9939 | 0.9964 |
| | $r_C$ | 0.9796 | 0.9648 | 0.9716 | 0.9804 | 0.9651 | 0.9779 | 0.9777 | 0.9928 | 0.9961 |
| | MAE | 3.3244 | 4.4760 | 3.4676 | 3.1482 | 4.4452 | 3.1701 | 2.5589 | 1.6386 | 1.1885 |
| | RMSE | 4.3878 | 5.8783 | 5.2018 | 4.2717 | 5.8526 | 4.5156 | 4.3345 | 2.5061 | 1.8183 |
| POLES | $r_P$ | 0.9891 | 0.9862 | 0.9822 | 0.9764 | 0.9835 | 0.9823 | 0.9643 | 0.9831 | 0.9898 |
| | $r_C$ | 0.9891 | 0.9831 | 0.9764 | 0.9738 | 0.9815 | 0.9762 | 0.9606 | 0.9659 | 0.9892 |
| | MAE | 2.0402 | 2.6271 | 2.7632 | 2.8913 | 2.7222 | 3.0789 | 4.2122 | 3.8071 | 1.7276 |
| | RMSE | 2.7512 | 3.5772 | 4.1719 | 4.1323 | 3.7007 | 3.9869 | 5.4772 | 5.0704 | 2.7676 |
| WITCH | $r_P$ | 0.9748 | 0.9725 | 0.9657 | 0.9743 | 0.9724 | 0.9698 | 0.9902 | 0.9958 | 0.9976 |
| | $r_C$ | 0.9625 | 0.9584 | 0.9485 | 0.9654 | 0.9602 | 0.9592 | 0.9893 | 0.9909 | 0.9972 |
| | MAE | 3.7224 | 3.8778 | 4.2143 | 3.4942 | 3.7722 | 3.7820 | 1.6708 | 1.5686 | 0.6386 |
| | RMSE | 4.6483 | 4.9326 | 5.4789 | 4.4011 | 4.7899 | 4.7323 | 2.2389 | 2.0672 | 1.1471 |

**5.3 Reproducing the IAM scenarios with the time-dependent emulator: results**

In the first three experiments with the carbon budget constraints including the abatement limits (Cases A to C), the statistical indicators showed that the use of the transitional and free-fitting time-dependent MAC curves did not improve the

reproducibility of emission scenarios (Table 9). For all three IAMs, the scenario reproducibility was, in fact, slightly decreased with the introduction of the time-dependency to the MAC curves. In the next three experiments also with the carbon budget constraints but excluding the abatement limits (Cases D to F), the use of the time-dependent MAC curves generally only improved the scenario reproducibility for POLES. In contrast, in the last three experiments with the carbon price constraints (Cases G to I), the use of the time-dependent MAC curves unanimously improved the scenario reproducibility, with the free-fitting time-dependent MAC curves being superior to the transitional time-dependent MAC curves. To understand why the use of time-dependent MAC curves improved the scenario reproducibility only under certain conditions, we examine the results separately for the carbon budget simulations (Cases A to F) and the carbon price simulations (Cases G to I) below.

### 5.3.1 Carbon budget simulations

In Cases A to C, both the transitional and free-fitting time-dependent approaches tend to give higher emissions in the near term and lower emissions later in the century than the time-independent approach for all three IAMs (Figure 12). This finding can be explained by the relative positions of the time-independent and time-dependent MAC curves. Because the time-dependent MAC curves are higher (i.e. higher marginal cost for a specific level of abatement) than the time-independent MAC curves in the near term, mitigation becomes more costly, resulting in higher emissions in the near term. The results were opposite later in the century. Because the remaining carbon budget must be conserved, emissions later in the century become lower with time-dependent MAC curves to compensate for the higher emissions earlier. Now, most results from Case A show that the time-independent approach already overestimated the emissions in the near term and underestimated the emissions later. Hence, those deviations were not reduced by the adoption of the time-dependent approach (Cases B and C); it was rather increased, despite the better fit of the time-dependent MAC curves to the price-quantity data from IAMs than the time-independent MAC curves.

Our implicit hypothesis was that the time-dependent approach yields a higher scenario reproducibility than the time-independent approach; however, this hypothesis proved wrong for Cases A to C. To understand the unexpected outcome, it is important to consider the carbon price. There are two different yet associated quantities from the emulator that can be characterized as carbon price: i) value of the MAC curve and ii) shadow price. The shadow price is always higher than or equal to the value of the MAC curve, as the shadow price is not influenced by various model constraints. Although there is no definitive argument to judge which quantity should be compared to the carbon price reported by IAMs, we primarily compare the value of the MAC curve with the IAM carbon price (available in the ENGAGE Scenario Explorer) (Figure 13).

We now ask why both the time-independent and time-dependent approaches overestimated near-term $CO_2$ emissions and underestimated long-term $CO_2$ emissions. Taking AIM as an example, the emission overestimations till mid-century are primarily caused by the difference in carbon price between the emulator and the IAM. The MAC estimates are generally lower than the corresponding carbon prices of AIM, with differences depending on the carbon budget of the scenario. The generally lower MAC estimates largely explain the emission overestimations till mid-century. Later in the century, on the other hand,

the MAC estimates become higher than the AIM carbon prices, resulting in the emission underestimations. The MAC estimates from different carbon budget pathways converge after the emissions reach the lower limit defined by the maximum $CO_2$ abatement level for AIM (116.2% relative to the baseline (Table 2)). An exception is the emission overestimations in 2025, which stem from the upper limits of the first and second derivatives of abatement changes, which do not allow a rapid emission reduction required to follow the original AIM scenarios. If these assumed upper limits are dropped (Cases D to F), the 2025 emissions became substantially lower and better reproduced the original emission levels (e.g., Panels (a1) and (b1) of Figure 12). However, the impact of these abatement bounds is limited to the very near term. The emission overestimations till mid-century are better explained by the carbon price differences discussed above.

Additional descriptions of the results from the other two IAMs follow (Cases A to F). For POLES, the time-independent approach slightly underestimated the emissions in the near term. Similarly to the results from AIM, both time-dependent approaches overcorrected this negative discrepancy and resulted in emission overestimations in the near term. Later in the century, the time-dependent approaches overcorrected the discrepancy in the opposite way and resulted in emission underestimations. When the abatement limits are removed (Cases D to F), the transitional time-dependent approach outperformed (Table 9), which was however primarily the consequence of the excessive drop in 2025 emissions of the time-independent approach (Panels (a4) and (b4) of Figure 12), with high penalty in the statistical indicators for the time-independent approach. For WITCH, the differences in the results between the time-independent and time-dependent approaches are the smallest among the three IAMs. This reflects the fact that the time-independent MAC curve largely captured the relationship between the abatement level and the carbon price in the case of WITCH, except for a limited number of near-term data points representing very high abatement levels (Panel (a5) of Figure 11). The WITCH results also exhibited the general deviation trend seen from other models: emission overestimations in the near term and emission underestimations later in the century. This general trend can be also explained by the carbon price differences. Furthermore, the comparison of the carbon prices indicates that the discount rate in WITCH may be lower than the assumed discount rate of 5% used in our emulator. As discussed earlier, in the absence of information on the discount rate used by all but a few IAMs, our emulator assumes 5% for all IAMs. The discount rate in IAM may follow the Ramsey rule, meaning that the discount rate is time-dependent, depending on the future economic growth.

**5.3.2 Carbon price simulations**

In stark contrast to the results discussed above, the results based on the experiments using prescribed carbon prices (Cases G and I) show that the use of time-dependent MAC curves can improve the reproducibility of $CO_2$ emission scenarios over the use of time-independent MAC curves (Panels (c1) to (c9) of Figure 12). In particular, near-term emission pathways up to mid-century were more closely reproduced with the use of time-dependent MAC curves, following our expectation. This is because time-dependent MAC curves capture the near-term relationship between the abatement level and the carbon price much better than time-independent MAC curves. On the other hand, near-term emissions were underestimated with the use of time-

independent MAC curves because such MAC curves tended to be lower (i.e., lower carbon price for a given level of abatement) than the near-term data points, which led to an underestimation of near-term mitigation costs and thus an overestimation of abatement. The use of free-fitting time-dependent MAC curves yielded higher scenario reproducibility than the use of transitional time-dependent MAC curves.

The superiority of time-dependent MAC curves over time-independent MAC curves discussed above can be confirmed by the statistical indicators in Table 9. This table also indicates that such results can only be found under the simple experimental setup with prescribed carbon prices. Under the more complex (and more applied) setup, in which carbon price pathways are endogenously optimized under given carbon budgets, the superiority of time-dependent MAC curves become less clear. This is due to the effect of carbon price pathways – an important determinant of scenario reproducibility – which

can even negate the benefit of using time-dependent MAC curves.

       Ultimately, emission scenarios will be perfectly reproduced, if the following two conditions are met: first, the original IAM data (the relationship between the abatement level and the carbon price) are perfectly captured by the MAC curve; second, the carbon price pathways are also perfectly reproduced by the emulator. While the first condition can be adequately satisfied with the use of time-dependent MAC curves within limits set by the functional form of the MAC curve, the second condition

cannot necessarily be met due to various constraints in the IAMs that cannot be captured by the emulator. For example, the AIM carbon price pathways have first peaks in the near term, followed by second peaks later in the century. Such complex terrains of carbon price pathways, which are exogenously imposed in recursive dynamic models, cannot be reproduced by our intertemporal optimization emulator. Even the carbon price pathways of the intertemporal optimization model WITCH, which shows a monotonic and exponential increase over time, differ from the carbon price pathways of the emulator. The discussion

here points to the importance of investigating carbon price pathways to further improve the IAM emulator.

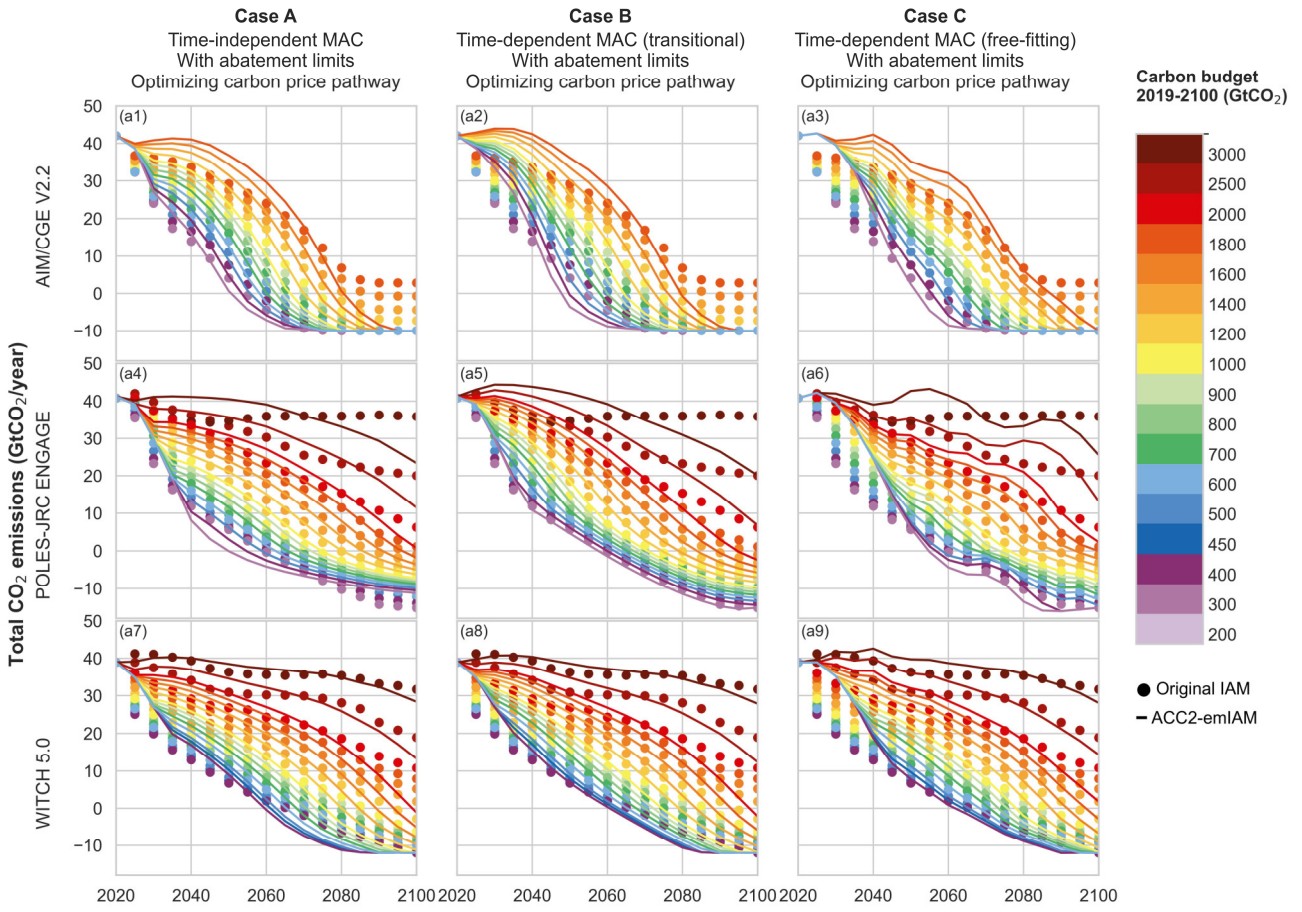

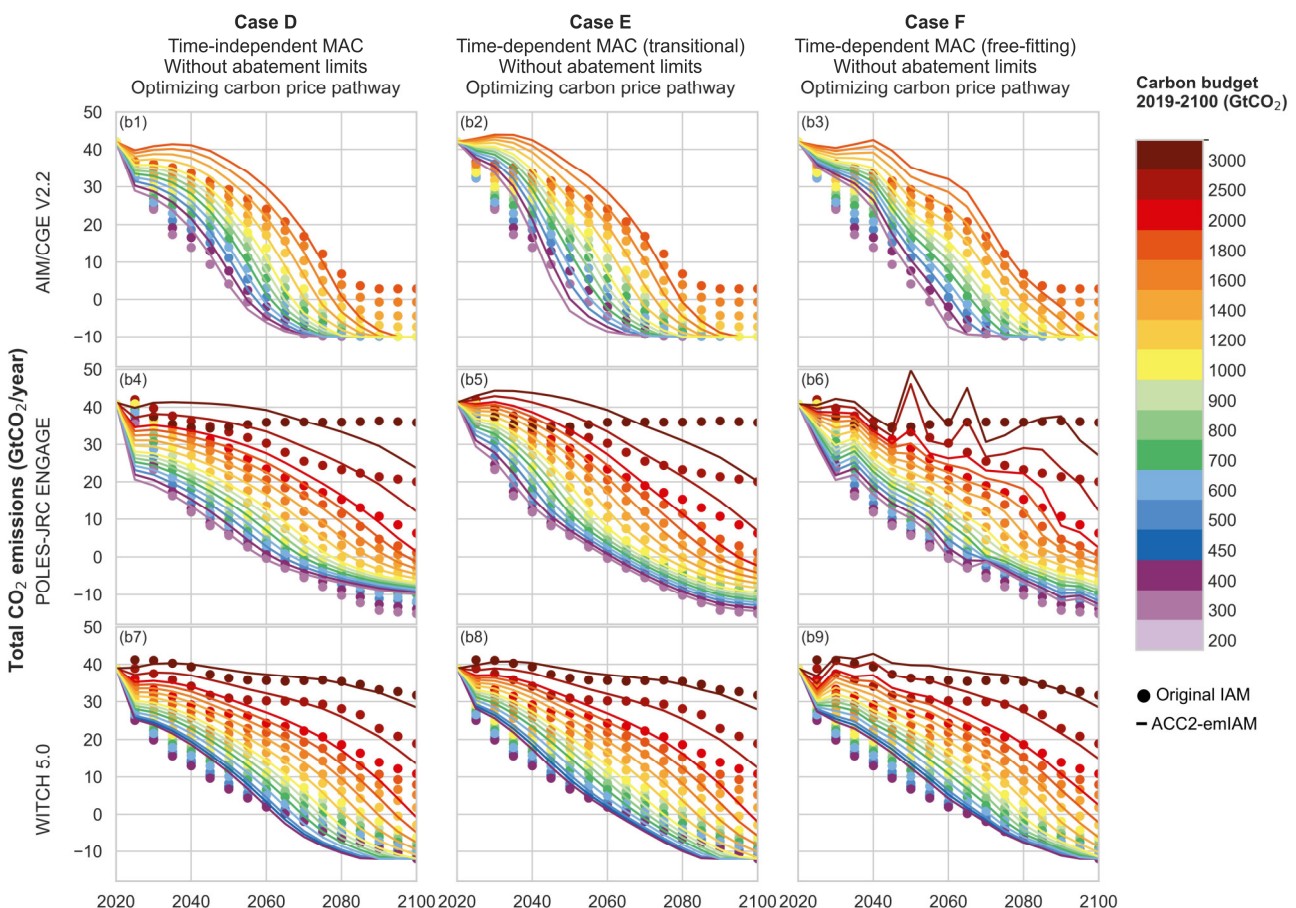

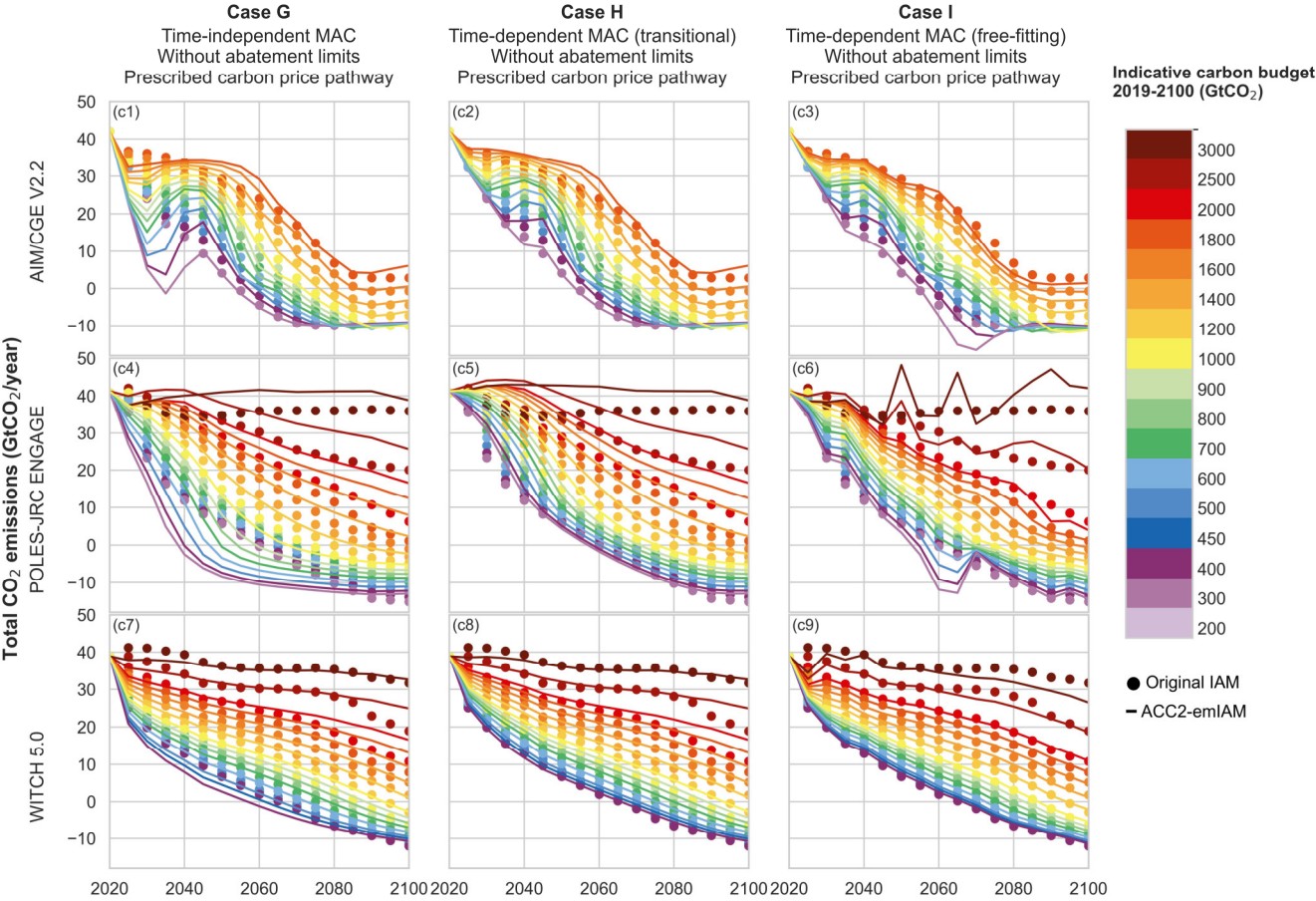

**Figure 12. Comparison between the reproduced CO₂ emissions from emIAM and the original emissions from the three IAMs for the experimental cases summarized in Table 9.** The figure shows the results for the ECB scenarios without INDC. In Panels (c1) to (c9), carbon budgets are only indicative, as the simulations were driven by carbon prices, without using carbon budgets.

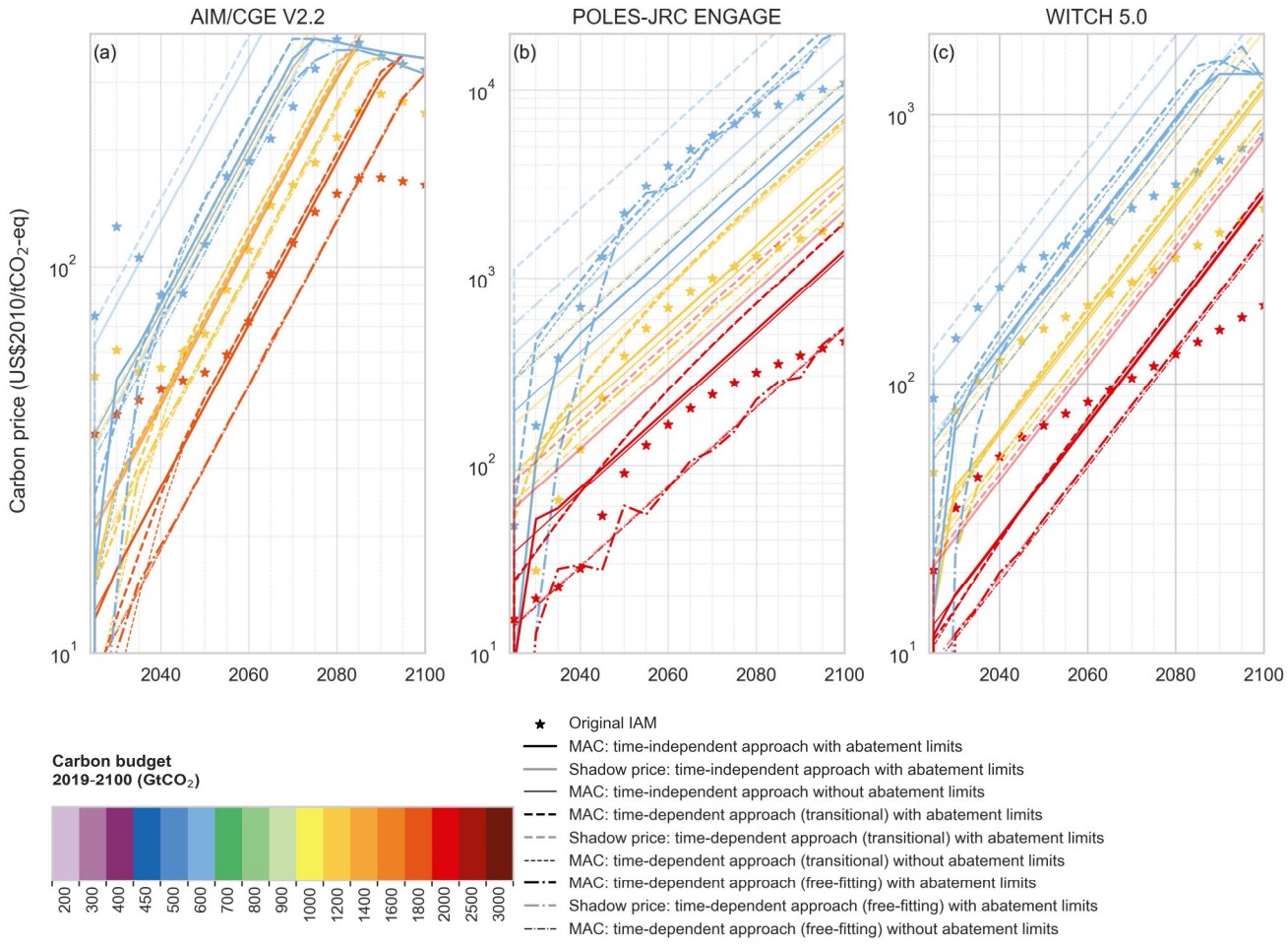

**Figure 13. Carbon price pathways from the time-independent and time-dependent emulators and the three IAMs.** MAC indicates the value of the MAC curve at each period under each scenario. Shadow price indicates the change in the total policy cost (the area of the MAC curves) for an infinitesimal change in emissions from the optimal level. The carbon prices of IAMs are indicated by star symbols. Selected three carbon budget scenarios are shown for each IAM. Vertical axes are on a logarithmic scale.

## 6. Conclusions

We have developed emIAM, a novel modeling approach to emulating IAMs by using an extensive array of MAC curves: ten IAMs (nine ENGAGE IAMs and GET); global and ten regions; three gases ($CO_2$, $CH_4$, and $N_2O$); eight portfolios of available mitigation technologies; and two emission sources (total anthropogenic and energy-related). A series of four validation tests (Table 4) were performed using ACC2-emIAM, the hard-linked optimizing climate-economy model, to reproduce the original IAM outcomes. The results showed that the original emission pathways were reproduced reasonably well in the majority of cases (Tables 5 and 6), although the reproducibility varied depending on the IAM, region, gas, portfolio, source, test, and scenario type as summarized below.

- Certain data points were difficult to capture by MAC curves. In particular, PKB scenarios with low carbon budgets can give very large carbon prices in the near-term. Such data points tend to deviate from the trend of other data points and were manually removed from the MAC curve fitting where appropriate (Figure 1 and Table 1). Except for these "outliers,"

no discernible difference in the data trend was found between ECB scenarios and PKB scenarios, supporting the use of common MAC curves for ECB and PKB scenarios. Note also that certain data points from GET at high abatement levels do not follow the trend of other data points and were also removed from the MAC curve fitting where appropriate. We speculate that these data points are affected by the limit on CCS capacity assumed in GET.

- Some IAMs were more easily emulated than other IAMs, reflecting specific model features such as solution methods, technology assumptions, and abatement inertia. The emulator can usually reproduce the emission pathways of an IAM better if the model response to carbon price is well fitted with a MAC function.

- The validation results for the two long-lived gases $CO_2$ and $N_2O$ did not strongly differ across all four tests, even though for Tests 2 to 4, there is a difference in the model setup between the original IAMs (GHG aggregation using GWP100) and ACC2-emIAM (individual gas cycle modeling without using GWP100). On the other hand, the validation results for the short-lived gas $CH_4$ in Tests 2 to 4 were not as good as those in Test 1. Test 4, with the additional mid-century temperature target, yielded higher reproducibility for $CH_4$ than Tests 2 and 3.

- Overall, the global emissions were better reproduced than the regional emissions. $CO_2$ emission pathways were generally better reproduced than $CH_4$ and $N_2O$ pathways. Specific pathway features such as $CO_2$ emissions in 2030, 2050, and 2100, cumulative negative $CO_2$ emissions from 2020 to 2100, the year to net zero for $CO_2$, and that for GHG were reproduced to varying degrees across models and carbon budgets (Figure 10). While certain biases were found for certain pathway features for some models, as reported earlier, no general conclusions can be drawn.

- The overall good reproducibility of emIAM relies on the use of time-independent MAC curves for percentage emission reductions. The behaviors of IAMs that contain various time-dependent processes were generally well captured by the time-independent MAC curves in the second half of the century, although the goodness of fit varies considerably among IAMs. However, time-independent MAC curves can work only poorly on shorter timescales for many IAMs. A plausible explanation for the overall good reproducibility in the second half of the century is that the use of percentage abatement levels relative to rising baseline can offset the effect of lowering mitigation costs over time. In other words, the higher the baseline scenario is, the larger the absolute amount of emission reduction is (for the same percentage emission reduction). If technology costs will not vary significantly over time, a time-independent MAC curve can be a reasonable assumption (under a stable baseline scenario).

- For certain IAMs (AIM, POLES, and WITCH), time-dependent MAC curves provide a better fit to the price-quantity data generated from the original IAM than time-independent MAC curves. However, the use of time-dependent MAC curves improves the reproducibility of emission scenarios only when the equivalent carbon price pathway is prescribed to the emulator. When the carbon price pathway is endogenously optimized under the equivalent carbon budget, it will differ from the carbon price pathway used for the IAMs. This difference in carbon prices can negate the benefit of using time-dependent MAC curves. The overall performance of the emulator is determined by a complex interplay of various

factors, including the MAC curves, the upper bounds of the first and second derivative limits, and carbon price pathways. Reproducing carbon price pathways will be an important consideration for the future development of IAM emulators.

If one is interested in using emIAM, this could easily be done by combining the MAC curve(s), the limits on the abatement levels and their first and second derivatives, and the baseline scenario of the IAM of interest in an optimization environment such as GAMS. We do not provide specific recommendations on the appropriateness of using each MAC curve and leave it up to the user to decide which MAC curves to use because the required accuracy of the IAM emulator depends on the purpose of the application. However, the goodness of fit of the MAC curves to the original IAM data and the results of

validation tests should be carefully examined. Materials needed to make such decisions are systematically presented in Supplement and our Zenodo repository, in addition to the discussion above.

This study demonstrated 1) a methodological framework to generate MAC curves from multiple IAMs simulated under a range of carbon budgets and carbon price scenarios and 2) another methodological framework to assess the performance of MAC curves with a simple climate model to reproduce original IAM outcomes. Our methods are generic and

transparent, providing an avenue for extending simple climate models to hard-linked climate-economy models. Future studies may emulate specific IAMs with more tailored parameterization approaches. We also open up an avenue for performing a quasi-multiple IAM analysis with low computational cost. Given the variety of IAMs available today, insights from multiple IAMs are indispensable for creating robust findings. Finally, simple models are complementary to complex models; modeling is an art that can shed light into the fundamental laws of complex systems (Yanai, 2009). In similar vein, emIAM can further

pave an avenue for understanding the general behavior of IAMs.

*Code availability.* GAMS code (emIAM v1.0) for deriving MAC curves is archived on Zenode with doi: 10.5281/zenodo.7478234.

*Data availability.* Data for the parameters in MAC curves and associated upper limits on abatement levels and their first and second derivatives are available on Zenodo with doi:10.5281/zenodo.7478234. The supplement related to this article is also available online.

*Author contributions.* Conceptualization of the IAM emulator, W.X. and K.T.; simulations using ACC2-emIAM, W.X. and K.T.; simulations using GET, K.T, D.J., and M.L.; analysis of simulation results, W.X., K.T., and D.J.; writing - original preparation for figures and tables, W.X.; writing - original preparation for text, K.T.; writing – revision and editing, W.X., K.T., P.C., D.J., and M.L. All authors have read and agreed to the submitted version of the manuscript.

*Competing interests.* The authors declare that they have no conflict of interest.

*Acknowledgments.* K.T. dedicates this paper to the memory of Prof. Hiroshi Yanai (1937-2021) of Keio University, Tokyo, Japan, a pioneer in the field of Operations Research and his bachelor thesis advisor, by whom K.T. was taught the fundamentals

of mathematical modeling and academic writing and the joy of intellectual pursuit. We would like to thank, in particular, but not limited to, the following individuals (in alphabetical order) for their valuable comments and related useful discussions: Nico Bauer, Thomas Bossy, Stéphane De Cara, Mark Dekker, Yann Gaucher, Takuya Hara, Xiangping Hu, Stuart Jenkins, Gunnar Luderer, Irina Melnikova, Leon Merfort, Yang Ou, Dale Rothman, Chris Smith, Xuanming Su, Masahiro Sugiyama, Kiyoshi Takahashi, Frank Venmans, Rintaro Yamaguchi, Tokuta Yokohata, Theodoros Zachariadis, and an anonymous reviewer. W.X. acknowledges financial support from the China Scholarship Council. This research was conducted as part of the Achieving the Paris Agreement Temperature Targets after Overshoot (PRATO) project under the Make Our Planet Great Again (MOPGA) program and funded by the National Research Agency in France under the Programme d'Investissements d'Avenir, grant number ANR-19-MPGA-0008. We further acknowledge the European Union's Horizon Europe research and innovation programme under Grant Agreements N° 101056939 (RESCUE – Response of the Earth System to overshoot, Climate neUtrality and negative Emissions) and N° 101081193 (OptimESM – Optimal High Resolution Earth System Models for Exploring Future Climate Changes) and the Environment Research and Technology Development Fund (JPMEERF20202002) of the Environmental Restoration and Conservation Agency (Japan).

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
