# Peer review of "emIAM v1.0: an emulator for Integrated Assessment Models using marginal abatement cost curves"

_EGUsphere, 2022_

## Author Comment (AC1)

**#Reviewer 1**

This study estimated a large set of marginal abatement cost (MAC) curves based on the output of IAMs in the ENGAGE Scenario Explorer and the GET model. The MAC curves were then applied to the emulator for Integrated Assessment Models (emIAM) and coupled to a simple climate model, ACC2. The test results showed that emIAM was able to reproduce the original IAM emission outcomes under similar conditions. The topic provided rich information about MAC curves under various IAMs, as well as different regions evaluated in the manuscript. While I agree with the authors that the analysis provided by the authors is certainly of general interest to climate-economic model developers and climate-focused researchers, I unfortunately cannot recommend publication of the manuscript in its present form. Here are my concerns:

[Response] We thank the reviewer for taking the time to read our manuscript and for providing useful comments. We have carefully revised the manuscript based on the reviewer's comments.

First, the authors reviewed a range of existing literature about the categories of MAC curves and different MAC curves estimated under various backgrounds. However, the results and analysis generally focused on the outcome of this study. I recommend adding a comparison between the estimated MAC curves in this study and those presented in existing studies, including differences in function forms, appropriate interpretation of parameters, and other major differences compared to existing estimates.

[Response] We selected several previous studies using MAC curves and compared them with ours (see Figure 4 and Figure 7). We added the following text to the manuscript:

*The functional form of the MAC function used by Su et al. (2017) is consistent with our study, and Tanaka et al. (2021) used equation (2) in Table S2. Harmsen et al. (2019) considered time-dependent MAC curves and no explicit function is provided. Despite some differences in the form of the functions, the MAC curves for energy-related $CO_2$ used in Su et al. (2017) and Tanaka et al. (2021) are within the range of the MAC curves from ENGAGE IAMs, but the MAC curves for $CH_4$ and $N_2O$ used in Tanaka et al. (2021) show a higher level of marginal carbon price. Harmsen et al. (2019) show that $CH_4$ MAC curve in 2050 is also to the left of our results, but that in 2100 are close to our study. Meanwhile, their results for $N_2O$ are in the middle of our results and not much different between 2050 and 2100.*

[Figure]

**Figure 1. Global MAC curves for total anthropogenic and energy-related CO₂, CH₄, and N₂O emissions derived from nine ENGAGE IAMs.** In panels (a) to (f), the solid line indicates that the MAC curve is within the applicable range; the dashed line means that it is outside the applicable range (i.e., above the maximum abatement level indicated from underlying IAM simulation data or above the range of carbon prices considered for fitting the MAC curve; see Tables 1 and 2). Different colors indicate different IAMs. The MAC curves from selected previous studies (Su et al., 2017; Harmsen et al., 2019; Tanaka et al., 2021) are shown for comparison. The MAC curves from Harmsen et al., (2019) are time-dependent and the figure shows those for the years 2050 and 2100.

[Figure]

**Figure 7. Global MAC curves for energy-related CO₂ emissions derived from the GET model with different portfolios of available mitigation technologies.** Different colors indicate

different technology portfolios (see Section 2.2 for details). Global MAC curves for energy-related $CO_2$ emissions from ENGAGE IAMs are shown as a comparison in gray lines, and the MAC curves from selected previous studies (Su et al., 2017; Tanaka et al., 2021) are shown in star-shaped lines.

Second, the ENGAGE Scenario dataset includes a wide range of outputs from various IAMs and regions. I am not quite sure about the reasons why the output of a separate GET model was also used for estimating the MAC curves. An explanation of the necessity of adding the output of the GET model is needed for readers to understand the framework of this study more clearly.

[Response] It is correct that the ENGAGE scenario database covers a broad range of output from various IAMs. However, we still simulated the GET model and used the output to additionally explore the effect of technological assumptions (e.g. CCS capacity) on the MAC curves. This was possible only with GET because we have a capability to run the GET model as needed, but this was not possible with ENGAGE IAMs because no such output is included in the ENGAGE Scenario Explorer (i.e. technological assumptions are kept the same in each model when it is simulated under different carbon prices). The motivation for using GET was already stated in the initial manuscript:

> *We further apply the emIAM approach to the GET model (Lehtveer et al., 2019), an IAM that did not take part in the ENGAGE project. We can directly simulate GET to derive MAC curves under different model configurations, which complements the existing data from IAMs simulated under single configurations for the ENGAGE project.*

Third, the manuscript mentions that the emIAM-ACC2 model minimized total abatement costs to obtain possible emission pathways for reproducing the outcomes from other IAMs. More information about how this process works is needed, including the necessary equations and the objective function for minimizing.

[Response] We thank the reviewer for the suggestion. We have added a more detailed description of this process in Section 4.1, as in the following statements:

> *More specifically, ACC2 uses equation (2) to calculate the abatement costs (ABC) of regions (or global total), gases, and years.*
>
> $$ABC_{t,r,g} = Eb_{t,r,g} \cdot \int_0^x f_{t,r,g}(x)dx \qquad (2)$$
>
> *where $t, r, g$ represent year, region, and gas, respectively. $x$ is the abatement level compared to the baseline scenario. $f_{t,r,g}(x)$ is the MAC function. $Eb$ is the baseline emission level for the IAM. The objective of the model is to minimize the net present value of the total abatement cost (TABC), that is:*

$$min\,TABC = \sum_{t,r,g} \frac{ABC_{t,r,g}}{(1+DSC)^{t-t0}} \qquad (3)$$

*where DSC is the discount rate and t0 represents the base year used for abatement cost calculations (2010 in this study).*

*In this study, we replace the existing set of MAC curves in ACC2 with the global and regional MAC curves obtained in this study. We also replace the limits on abatement (i.e., upper limits on abatement levels and their first and second derivatives) with those obtained from this study. We assume a 5% discount rate in the validation tests, a rate commonly assumed in IAMs (Emmerling et al., 2019), which is also consistent with some of the IAMs analyzed here such as MESSAGE and GET (Figures SI 1.2-1 and 1.2-2 of Riahi et al. (2021)). But we were unable to find the discount rates used in the other IAMs. Note that a 4% discount rate was used as default in recent studies using ACC2 (Tanaka and O'Neill, 2018; Tanaka et al., 2021) We consider the mitigation costs through 2100 in scenario optimizations.*

Fourth, this study provided many figures (some of which are similar) to present the estimations of the MAC curves and the emulating results, especially in the Supplement. While these figures provide visual information to present relevant results, there are too many figures stacked together, making it difficult for readers to find the information they need. An appropriate way to manage these figures, such as indexing them using tables or other means of relevance, should be added.

[Response] We thank the reviewer for the suggestion. We have added a list of tables and figures to Supplement so that it is easier for readers to find the relevant content.

**#Reviewer 2**

Xiong et al have developed an emulator for integrated assessment models (IAMs) using a "marginal abatement cost (MAC)" approach. The emulator uses a large set of MACs derived from IAM-based scenarios in an existing database to reproduce most original IAM emission outcomes at a much lower computational cost than the original model. Additionally, the emulator can be coupled to a simple climate model to generate emission pathways for a specific temperature target. In general, this is a positive modeling development, as emulators are common in various fields, including climate models, but are currently lacking in the IAM field. As IAMs continue to advance in complexity, emulators could be valuable in scenario discovery.

[Response] We thank the reviewer for recognizing the future potential of our work and for providing comments that were useful for improving the quality of our manuscript.

While this study represents one of the first attempts to develop an IAM emulator, there are three main areas for improvement, as summarized below and discussed in detail.

Firstly, the overall flow should be better. Sometimes, the details are provided before a general overview, creating challenges for readers.

[Response] We thank the reviewer for pointing out this problem. We have substantially revised the paper structure to improve the flow of the paper.

Secondly, the visualization could be improved. Many figures are too busy to deliver the critical message.

[Response] We thank the reviewer for the suggestion. We have made systematic refinement of the visualization of the figures so that they present the desired information more clearly. For example, we improved the layout of Figure 1 (see below), with panel c presenting results for each decade (rather than every five years) in larger subpanels. Panel d, which shows MAC curves, is now shown in a larger format. In addition, we have also improved the legend of the figure. We think the legend is now easier to understand.

[Figure]

**Figure 2. Overview of the methods to derive MAC curves and limits on abatement (upper limits on abatement levels and their first and second derivatives).** The figure uses the data for global total anthropogenic $CO_2$ emissions from REMIND for illustration. The chromatic colors indicate the respective carbon budgets for the period 2019 – 2100 in $GtCO_2$. The grey color indicates the baseline scenario ("NPi2100" in the original scenario name). Scenarios without INDC consider currently implemented national policies (circle; indicated as "NPi2020" in the original scenario name); scenarios with INDC further consider national emission pledges until 2030 (triangle; indicated as "INDCi2030" in the original scenario name). ECB scenarios consider carbon budgets till the end of this century, with a possibility of temporal budget overspending (filled circles; with "f" in the original scenario name); PKB scenarios consider carbon budgets without allowing temporal budget overspending (open circles; without "f" in the original scenario name). Crosses indicate data points from scenarios that were not considered in the derivation of the MAC curve (i.e., EN_INDCi2030_700, EN_INDCi2030_800, EN_NPi2020_400, and EN_NPi2020_500 for REMIND (see Table 1)). In the equation of the MAC curve, $a$, $b$, $c$, and $d$ are the parameters to be optimized; $x$ is the variable representing the abatement level in percentage relative to the assumed baseline level). Note that panel c shows data only for every ten years for the sake of presentation.

Lastly, while I appreciate the massive details and results provided by authors, most of the result text was just purely describing the results, without a high-level generalization or explanation of

the reason behind the findings. There is little discussion about the model structures, which might help explain the results.

[Response] We thank the reviewer for the suggestion. We have substantially expanded Section 5 to provide a high-level generalization. Regarding the model structures, we have the following discussion in Section 3:

*The results vary in terms of the range of carbon prices, the range of abatement levels, and the dispersion of data points. For example, the carbon prices of AIM and COFFEE remain below $500/tCO$_2$, while the carbon prices of POLES and MESSAGE can exceed $5,000/tCO$_2$. The maximum abatement levels of COFFEE and REMIND are over 150%, while others are in the range of 100%-120%. AIM provides a limited amount of data at low abatement levels. IMAGE and POLES produce more dispersed data distributions than other models, which may be related to the fact that these models are recursive dynamic models (Table 1); however, the other recursive dynamic models, AIM and GEM, produce less dispersed data distributions that can be well captured by MAC curves. POLES can be seen as an example where our time-independent MAC curve approach does not work well. The MAC curve, if taken every five years, shifts to the right over time (Figure S4).*

Detailed comments:

1.      Line 108: please explain what's the NPi2100 scenario. Previous sentences mentioned other scenarios like NPi2020 and INDCi2030, but not NPi2100.

[Response] We have added an explanation about what NPi2100 is in that sentence. Here, NPi2100 is our reference scenario that assumes a continuation of the current stated policies until 2100.

2.      Section 2 breaks the entire flow. First, it's unclear what precisely the MAC is in this context. Some experienced readers might generally know a MAC as a function between the carbon price and % emission reductions. Still, different kinds of literature might have different definitions (i.e., carbon price or emission price) or sectoral and gas specifications. This critical "background" information did not show up until Section 3.1. So before diving into the IAM and overwhelming scenarios definitions, this paper could benefit from a high-level schematic showing the entire working flow. (BTW, the current Fig.1 is overwhelming, with many texts and details but somewhat unclear logic).

[Response] We thank the reviewer for the comment. Given the general structure of our paper, we think that the discussion on scenarios (Section 2) should come before the discussion of MAC curves (Section 3). In Section 1, we have a paragraph that introduces the general concept of MAC curves and why the MAC curve approach was used to conduct these studies in the Introduction. We have further added the following text in Section 1: "*In the context of climate change mitigation, a MAC generally represents the incremental cost of reducing an additional unit of emissions; a MAC curve illustrates these costs as the level of emission reductions increases relative to the baseline.*" We have modified Figure 1 to more clearly present the methodological flow. We put the description of the paper structure at the end of Section 1. In revising the manuscript, we kept in mind that the paper structure should be clearer (we further made use of footnotes where necessary). We hope that our revision adequately addresses the reviewer's concern.

3.      From section 2, It's unclear why this paper needs the GET model in addition to the ENGAGE scenario database.

[Response] This point was also raised by reviewer #1. The reason why we use the GET model is that we can directly simulate the GET model to explore the effect of technological assumptions on the MAC curves. Though the ENGAGE Scenario Explorer provides a large number of scenarios that show the carbon price pathways under different carbon budgets, it is not suited for the type of analyses that can be possible with GET. While we have a capability of simulating GET, we cannot directly simulate the IAMs in the ENGAGE Scenario Explorer and can only use existing output from these IAMs. The GET model provides a set of $CO_2$ emission pathways due to the change of carbon prices under different technical portfolios, which can complement the output of the ENGAGE IAMs. Therefore, we use both ENGAGE IAMs and the GET model for this study.

4.      Line 157: "if there are non-zero carbon prices in baseline, we subtracted them from the carbon price in mitigation scenarios", is this implicitly assuming a linear relationship between CO2 price and emission reductions? i.e., a linear MAC?

[Response] We thank the reviewer for the question. All our MAC curves are nonlinear as described in Section 3.1. The carbon price for each case is also the relative level to the baseline scenario. The small corrections that the reviewer pointed out should not influence the functional form of the MAC curve.

5.      Line 163: I know the term "portfolio" is clearly defined in the GET modeling part in Section 2.2, but what does the "portfolio" mean in the ENGAGE scenario database?

[Response] A portfolio is a set of technological assumptions in the GET model. In the ENGAGE Scenario Explorer, there is only one portfolio for each IAM, so the portfolio is irrelevant to ENGAGE IAMs. Therefore, we revised the statement as follows:

*for all cases (i.e., models, gases, regions, and sources in ENGAGE, and portfolios in GET).*

6.      Line 165 and below: what exactly does this functional form mean? Again, this is breaking the flow, as I saw additional explanations 30 lines below in line 192.

[Response] We have revised this section to improve the flow of the argument. Meanwhile, we have further explained this function, and the definition of each parameter has been clarified as well.

7.      Line 165, where is the carbon price in this equation (1)? I guess the carbon price is f(x), but the text below, albeit with many details embedded, did not indicate which term represents the carbon price.

[Response] Yes, the carbon price is f(x), which means that the carbon price is a function of the abatement level. We added a further explanation for this equation to the following statements:

*a, b, c, and d are the parameters to be optimized in each case. $x$ is the variable representing the emission abatement level in percentage relative to the assumed baseline level. The carbon price (i.e., $f(x)$ in equation (1)) is expressed in per ton of $CO_2$-equivalent emissions, using GWP100 (28 and 265 for $CH_4$ and $N_2O$, respectively (IPCC, 2013)) to convert $CH_4$ and $N_2O$ emissions, as assumed in the IAMs emulated here (Harmsen et al., 2016). GWP100 is effectively the default emission metric used to convert non-$CO_2$ GHG emissions to the common scale of $CO_2$ and has been used for decades in multi-gas climate policies and assessments, including the Paris Agreement (Lashof and Ahuja, 1990; Fuglestvedt et al., 2003; Tanaka et al., 2010; Tol et al., 2012; Levasseur et al., 2016; UNFCCC, 2018, 2023).*

8.      Line 199, "performing consistently the best for all IAMs (see the Zenodo repository)". This crucial result needs at least a supplementary table or figure or even a main figure/table.

[Response] We thank the reviewer for the suggestion. We have added the table below in Supplement to show this result:

**Table S3. Statistics for function choices**

| Function | Count | Percentage (%) |
|----------|-------|----------------|
| T1 | 127 | 51.42 |
| T2 | 15 | 6.07 |
| T3 | 45 | 18.22 |
| T4 | 60 | 24.29 |
| **Total** | **247** | **100** |

9.      Line 208-209: why do the maximum first and second derivatives of temporal change in abatement levels correspond roughly to the limit of the technological change rate and the socio-economic inertia?

[Response] We thank the reviewer for the question. The technological change rate of mitigation measures shows limitations in the speed of implementation, while socio-economic inertia interferes with the rate of technology change by revealing that some systems need more time to change and adapt (Schwoon and Tol, 2006; Harmsen et al., 2019; Hof et al., 2021). In our study, we interpret technological change rate as the first derivative of the abatement levels, and socio-economic inertia as the second derivative. Therefore, the upper limits of the first and second derivatives, derived from individual model behavior, respectively represent the peak rate of technological change and socio-economic inertia for the entire sample of the IAM.

10.     Can you show the x- and y-axis in the same scale? (so that we can see how MACs shift in time)

[Response] We thank the reviewer for the comment. We have expanded the sizes of the subpanels in panel c by showing the results for each decade, and we have also used the same scale of the x- and y-axis for all subpanels in panel c. It can now be easier to see how the data shifts over time. The revised figure was copied as part of our response to the second general comments.

11.     Line 248: "crosses in the right panel of Figure 1" --- I cannot find crosses in the right panel because they are too small.

[Response] We thank the reviewer for the comment. In order to improve the clarity of the information displayed in Figure 1, we have adjusted Figure 1 so that the subpanels are larger and the information is easier to see. Please refer to the revised Figure 1 above.

12.     In figure 2, the authors pointed out different models show very different carbon prices for the same level of reduction. For example, when reaching a 100% reduction, the corresponding price is about $150, while POLES is about $1000. However, this could be the masked effect of the single fitted line on a wide range of scenarios. Even the fitted value for POLES indicated an ~$1000 to achieve a 100% reduction, there are individual data points (scenarios) reaching 100% reduction with much lower prices. For this type of data and distribution, perhaps the MAC approach is not suitable because of the nature of some particular models.

[Response] We thank the reviewer for the comment. We examined the data from POLES in more detail. We realized that the POLES model offers a sort of failed example of our MAC curve approach, in which the MAC curve, if taken every five years, shifts to the right over time (see Figure S4). In such cases, a time-independent MAC curve is not a proper approach to capturing the emission behavior of the model. We nevertheless present the results because a motivation of this study is to understand to what extent our general MAC curve approach can emulate the behavior of various IAMs.

[Figure]

**Figure S4. MAC curves of total anthropogenic $CO_2$ emissions per five years for POLES.** The dots are original data from the POLES model, and the lines are MAC curves derived from these data for different years.

13.     Section 3.2.2 discussed the role of the first and second derivatives of the abatement changes, which is interesting, but I still don't fully understand its value. I.e., do they have physical meanings? (see my comment # 8). Also, what if those upper limits for the first and second derivatives were removed? How could that change the fitted models?

[Response] If the first and second derivatives are taken into account, the rate of increase in the abatement level will rise slowly in the near term. It can reach its upper limit when society and technology have adapted to the policy requirements of climate mitigation (see red and blue lines of Figure R1). However, if this constraint is removed, it means that the upper limit of emission reductions (e.g. net zero for $CO_2$ or even negative $CO_2$ emissions) can be reached immediately (see black line of Figure R1), which is clearly not the case in the original model output.

[Figure]

**Figure R1. Abatement level considering the most growth rate of mitigation.** Here we assume the maximum potential of mitigation is 100%. If no social-economic inertia is considered for the MAC curve (black line), then the abatement level can reach 100% quickly. If only technological change inertia is considered for the MAC curve (red line), then the abatement level will grow with a fixed slope before it reaches 100%. If both technological change and social-economic inertia are considered for the MAC curve (green line), the abatement will increase slowly because the technology also needs time to change and adapt.

14.     Fig 4 seems to capture the model differences. However, the true question is to what extent are these differences because of the model's structural differences or differences in the scenarios simulated by different models? Each model may contribute varying numbers of scenarios to the database with unevenly distributed scenario narratives. Thus, the differences here might be driven by the artificial selection of the training sample. I hope the authors can

share some thoughts on this. Also, is there any notable structural differences that might be helpful to explain the observations in line 320-329?

[Response] We thank the reviewer for the question. We aim to extract the relationship between abatement levels and carbon prices for models from a large number of scenarios, so the number of available scenarios can influence how well the MAC curves can be fitted. The larger the number of carbon budget scenario is, the more accurate the fitted curves will generally be. Therefore, the distribution of carbon budgets tested by individual models (Table S7; see below) can be a potential source of bias. Meanwhile, the model structure could also be a reason as well. For example, the five IAMs (COFFEE 1.1, MESSAGEix-GLOBIOM 1.1, POLES-JRC ENGAGE, REMIND-MAgPIE 2.1-4.2, and WITCH 5.0) have very similar carbon budget ranges and number of scenarios while different solution concepts and solution methods (Table 1 and Table S7). Thus, we chose these IAMs and filtered the scenarios that they all provided (19 scenarios in total, see Table S7). The MAC curves for anthropogenic $CO_2$ emissions are given in Figure S37. The MACs between the different IAMs still vary considerably, but the results of the three general equilibrium models are close to each other, while those of the two partial equilibrium models are far apart, although we do not have a further insight into why this occurs. Note that the results of the MAC curves are not very sensitive to the number of scenarios, as the results for the subsample of scenarios we used here are very similar to the results for the full sample.

[Figure]

**Figure S37. Global MAC curves for total anthropogenic CO₂ emissions derived from the same scenarios for five ENGAGE IAMs.** The solid lines are MAC curves derived from the

subsample, and the dotted lines are MAC curves derived from the full sample. No upper limit of abatement level is shown for MAC curves.

**Table S7. Available scenarios for each model in the ENGAGE Scenario Explorer.** 0 means that the model does not provide this scenario, while 1 means that the model provides this scenario.

[Figure]

15.     Table 2, why do some models have huge coefficients for a and c? For example, the "a" parameter for REMIND CH4 or the "c" parameter for WITCH CH4 and N2O? Is this because of the model itself, or were the scenarios chosen for fitting? Also, the main text did not make any comment on Table 2.

[Response] We thank the reviewer for the question. In a single power function $y = a * x^b$, $a$ determines the position of the function curve in the vertical direction, and $b$ determines its shape. That is, when $b>1$, the curve is flatter near the origin and then rises sharply. When $0<b<1$, the curve is steeper near the origin and then flattens out. A large $a$ implies a large $y$ value. The

function $y = a * x^b + c * x^d$, which has two power functions, allows us to capture more complex trends of MAC curves.

Therefore, we think the phenomenon that the reviewer raised is due to a combination of the chosen function and the data distribution of models. The reason for the very high value of $a$ for REMIND $CH_4$ and $c$ for WITCH $CH_4$ and $N_2O$ is that the mitigation price is very low at a low abatement level, but rises sharply when the abatement level is close to the upper limit (nearly vertically for the REMIND and WITCH models (see Figure S7(g), S7(i), and S10(i)).

[Figure]

Figure S7(g) Global total $CH_4$ MAC    Figure S7(i) Global total $CH_4$ MAC

Figure S10(i) Global total $N_2O$ MAC

16.  Line 366: "They are further compared with the Global MAC curves for energy-related CO2 emissions from ENGAGE IAMs."

[Response] This review comment looks incomplete, but we guess that the reviewer asks why we compare the GET model's results with ENGAGE IAMs. We aim to see if there is any significant difference between the two datasets for MAC curves. The results show that the range is nearly as wide as that from ENGAGE IAMs (i.e., inter-technology portfolio range ≈ inter-model range) if we disregard the MAC curve from COFFEE.

17.     Figures 7 and 8: Given the current presentation, there's no way to check the model performance for emIAM-ACC2 visually. Please avoid showing so many lines/dots in one figure; this busy chart provides minimal information.

[Response] We thank the reviewer for the suggestion. We assume that the reviewer refers to Figures 8 and 9. We think that Figure 8 in the original manuscript was clear enough for the comparison because it shows only a subset of simulations (i.e. results of EBC scenarios without INDC). However, Figure 9 in the original manuscript was more difficult to read because all scenarios are shown. Therefore, we have modified Figure 9 to present only EBC scenarios without INDC (consistent with Figure 8).

18.     Technically, the entire validation test (section 4) is performed in the "training set". Ideally, this should be done in a validation set outside the training set. Authors could 1) try to select scenarios from another scenario database, such as IPCC AR6, with the same set of models and selected scenarios as validation, or 2) just randomly choose a part of the ENGAGE scenarios as the training set to fit MACs (if there's enough sample size), then use the remaining ENGAGE scenarios as the validation set.

[Response] We thank the reviewer for the suggestion. It is an interesting idea, but since our framework considers both the MAC function and the baseline scenario, as well as the constraints of the first and second derivative of abatement rates, selecting scenarios from other projects in AR6 can lead to inconsistency between the model used to train the emulator and the model that give test scenarios. The second point is also a useful suggestion, but the number of scenarios used to generate MAC curves in the dataset in this study is limited, so we decided to stick to our current approach. Nevertheless, we thank the reviewer for sharing the thoughts, which could be applied to our future study.

19.     Comparing Figures 10 and 11, I wonder why COFFEE performed well in the global test but poorly for most regions for CO2. Are they consistent?

[Response] We thank the reviewer for the question. We double checked and updated the results. Because of our mistakes, we provided the wrong emission pathways for scenarios with INDC from ACC2-emIAM. Now we have updated these figures with the correct results. The revised results show that the COFFEE model also performed well in reproducing the regional $CO_2$ emissions.

20.     Line 623 "The results showed that the original emission pathways were reproduced reasonably well in a majority of cases." This is oversimplified. The performance depends on

the gas, model, and maybe other features (if Figures 7 and 8 could have been clearer). Here needs a better summary of the findings.

[Response] We thank the reviewer for the suggestion. We have substantially expanded the discussion in Section 5. The new Section 5 has a high-level summary of the findings as follows:

- *The validation results for the two long-lived gases $CO_2$ and $N_2O$ did not strongly differ across all four Tests, even though for Tests 2 to 4, there is a difference in the model setup between the original IAMs (GHG aggregation using GWP100) and ACC2-emIAM (individual gas cycle modeling without using GWP100). On the other hand, the validation results for the short-lived gas $CH_4$ in Tests 2 to 4 were not as good as those in Test 1. Test 4, with the additional mid-century temperature target, yielded higher reproducibility for $CH_4$ than Tests 2 and 3.*
- *Overall, the global emissions were better reproduced than the regional emissions. $CO_2$ emission pathways were generally better reproduced than $CH_4$ and $N_2O$ pathways. Specific pathway features such as $CO_2$ emissions in 2030, 2050, and 2100, cumulative negative $CO_2$ emissions from 2020 to 2100, the year to net zero for $CO_2$, and that for GHG were reproduced to varying degrees across models and carbon budgets (Figure 10). While certain biases for some models were found for certain pathway features, as reported earlier, no general conclusions can be drawn.*
- *Some IAMs were more easily emulated than other IAMs, reflecting specific model features such as solution methods, technology assumptions, and abatement inertia. The emulator can usually reproduce the emission pathways of an IAM better if the model response to carbon price are well fitted with a MAC function.*
- *Certain data points were difficult to capture by MAC curves. In particular, PKB scenarios with low carbon budgets can give very large carbon prices in the near-term. Such data points tend to deviate from the trend of other data points and were manually removed from the MAC curve fitting where appropriate (Figure 1 and Table 1). Except for these "outliers," no discernible difference in the data trend was found between ECB scenarios and PKB scenarios, supporting the use of common MAC curves for ECB and PKB scenarios. Note also that certain data points from GET at high abatement levels do not follow the trend of other data points and were also removed from the MAC curve fitting where appropriate. We speculate that these data points are affected by the limit on CCS capacity assumed in GET.*
- *The overall good reproducibility of emIAM relies on our novel approach: time-independent MAC curves for percentage emission reductions. The behaviors of IAMs that contain various time-dependent processes were generally well captured by the time-*

*independent MAC curves. A plausible explanation is that the use of percentage abatement levels relative to rising baseline can offset the effect of lowering mitigation costs through learning.*

21.     Line 626, "Materials that are required for making such decisions are systematically presented in Supplement and our Zenodo repository." This is essential information; the authors should provide a couple of high-level bullet points.

[Response] As we responded above, the revised manuscript provides several high-level bullet points.

22.     Line 627, "Some IAMs were more easily emulated than other IAMs. The goodness of fit of the MAC curves depends on gases and regions." Again, this is another place that should have provided richer information beyond the current simple comment (which readers would even know before reading this paper).

[Response] This also relates to the two previous comments. We hope that the newly added bullet points address the reviewer's concern.

**#Reviewer 3**

**Summary:**

The paper describes an emulator for Integrated Assessment Models (IAMs) based on an aggregation of MAC curves of different models, regions, time points and greenhouse gases. The idea is interesting and useful, because it allows for quick assessments of abatement given different carbon prices, for which running IAMs may be computationally costly. The paper focuses on the calculation of these MAC curves, on which the authors are thorough, and on the validation of the resulting emulator in comparison to the output of the IAMs that the authors started with.

[Response] We thank the reviewer for taking the time to read our manuscript and for providing useful comments. We also thank the reviewers for recognizing the usefulness and thoroughness of our work.

**General comments**

While the idea of this emulator is interesting and useful, I unfortunately do not recommend publication in the paper's current form and am providing a few suggestions below that may be used for major revision.

1.      The scenarios used as input are merely listed, but little motivation is given why the ENGAGE database is chosen, while I think this is key to the resulting MAC curves in the emIAM. My suggestion would be to at least motivate why the ENGAGE database is suitable for this exercise, and why the authors are not using the full AR6 scenario database that came out last year.

[Response] We thank the reviewer for the comment. The AR6 Scenario database includes a large ensemble of scenarios from different projects, including the ENGAGE project. However, the AR6 Scenario database was not available at the time of our analysis. Meanwhile, we chose only the ENGAGE project because this project adopts the same socioeconomic assumptions (i.e. second marker baseline scenario from the Shared Socioeconomic Pathways (SSP2), which reflect middle-of-the-road socioeconomic conditions (Riahi et al., 2017)) and provides plenty of cases to derive the MAC curves (see Figure 3.2 of (Riahi et al., 2022)). Thus, we argue that the ENGAGE Scenario Explorer is the best dataset for our application as it gives a range of scenarios under different carbon budgets for many models with consistent configurations. We

have added some text to explain why we only used the scenarios from the ENGAGE project instead of the full dataset of AR6:

*The ENGAGE Scenario Explorer is now part of the larger IPCC Sixth Assessment Report (AR6) Scenario Explorer (Byers et al., 2022), which was not available at the time of our analysis. Although the use of the entire AR6 scenario dataset could be advantageous in terms of the number of IAMs and scenarios available for analyses (189 IAMs (including different model versions) and 1389 scenarios in the AR6 Scenario Explorer; 20 IAMs (including different model versions) and 231 scenarios in the ENGAGE Scenario Explorer), an advantage of using the ENGAGE Scenario Explorer is that the data from IAMs were obtained under a common experimental protocol, allowing consistent analyses.*

2.      A major concern is the lack of discussion in this paper. The paper contains a lot of detailed description of results, along with many detailed figures, but lacks broader discussion. For example, where to the gas differences in Fig. 4 or the regional differences in Fig. 5 come from? Could we have expected them beforehand? And what do the significant model differences in Fig. 2 imply for the ultimate results?

[Response] We thank the reviewer for pointing out this problem. However, we are afraid that it is nearly impossible to directly answer these questions because the IAMs we are dealing with are very different from each other and we do not have deep insight into each of these IAMs (we are not taking part in the ENGAGE project. We are merely using the publicly available database of ENGAGE). In this sense, we argue that our study is not designed to provide explanations for the differences found. Rather, our study aims to explore to what extent our generic MAC curve approach works for different models, gases, regions, etc, although we discuss possible reasons of good/poor MAC curve fitting and reproducibility where possible.

3.      The paper can be written more concise and requires a bit more flow to guide the reader throughout the steps. Also, the paper contains too many figures/panels which are not well readable, especially when it comes to symbols (circles/triangles, etc.) and scenario labels. The authors may consider moving some to the SI.

[Response] We thank the reviewer for the comment. We have made efforts to streamline the content and improve the flow throughout the manuscript. We have also polished the manuscript figures for better readability (e.g., font size and color scheme). For example, we have modified Figure 1 to more clearly present the overall methodological flow (e.g. reduced the number of small panels over time; changed the figure legend to something more intuitive). In Figure 9, we reduced the number of scenarios presented so that each scenario can be read more clearly. In

Section 3, we merged the content of the MAC functions from different paragraphs and moved it after the introduction of data processing. We also made use of footnotes where necessary to shorten the text and avoid breaking the flow of the manuscript.

4.      More details on the uncertainty of this approach is needed. Clearly, the results are gas, model, region and time dependent, while some of these things are actually aggregated into a single MAC in emIAM. What does this imply for the end results? Perhaps work with uncertainty bars in a summarizing plots in the end to give the reader a feeling for the uncertainty of emIAM. Similar for the parametric uncertainties in the values of a, b, c and d when fitting, which may require a sensitivity analysis.

[Response] We thank the reviewer for the suggestion. Though the relationship between the carbon price and the $CO_2$ abatement level can be well captured by MAC curves for most IAMs we considered, the results vary in terms of the range of carbon prices, the range of abatement levels, and the dispersion of data points. We have added 95% confidence intervals of the fitted MAC curves in Figure 2 and Figure 6 (and more figures in Supplement).

[Figure]

**Figure 3. Relationships between the carbon price and the global total anthropogenic $CO_2$ abatement level obtained from nine ENGAGE IAMs.** Each panel shows the results from each ENGAGE IAM. Data were obtained from the ENGAGE Scenario Explorer and are shown in colors and markers as designated in the legend. Black lines are the MAC curves. Crosses are the

data points that were not included in the derivation of MAC curves (Table 1). The shaded bands are the 95% confidence intervals of the fitted curves calculated by $\hat{y} \mp t_{\frac{\alpha}{2}} * S_{\varepsilon} *$ $\sqrt{1 + \frac{1}{n} + \frac{(x-\bar{x})^2}{\sum x^2 - \frac{(\sum x)^2}{n}}}$ (Thomson and Emery, 2014), where $S_{\varepsilon} = \sqrt{\frac{\sum(y-\hat{y})^2}{n-2}}$, $n$ is the sample size, $t_{\frac{\alpha}{2}}$ is the critical value of t-distribution, $\bar{x}$ is the mean of samples, $\hat{y} = f(x)$, and $x, y$ are the original abatement level and carbon price result from the IAM, respectively.

[Figure]

**Figure 4. Relationships between the carbon price and the global energy-related CO₂ abatement level obtained from GET with different portfolios of available mitigation technologies.** Panel (a) shows the results obtained from an older version of GET (Azar et al., 2013) for the sake of comparison. Panels (b) to (i) show the results from GET (Lehtveer et al., 2019) with different technology portfolios. See Section 2.2 for the definitions of technology portfolios. Points are the data obtained from GET; lines are the MAC curves calculated based on our approach. Open circles are the data that were not considered in the derivation of MAC curves (Table 1) and are typically found after 2100, in some cases above the abatement level of 160% (not shown). Note that we have converted the unit in Panel (a) from US$2010/tC, which is used in the older version of GET, to US$2010/tCO₂, the commonly used unit here. The shaded bands are the 95% confidence intervals of the fitted curves calculated (see the caption of Figure 2)

5.     The authors have chosen to work with percentage abatement w.r.t. baselines rather than absolute abatement. I understand the reasoning, but it is not trivial that this choice fully counteracts the lack of temporal dependency in the analysis (e.g., in the form of learning by doing), even though this is (perhaps even coincidentally) visible when comparing the percentage versions versus the absolute versions. Moreover, baselines significantly differ among models, which introduces another source of uncertainty. A discussion on this would be helpful in the paper.

[Response] We thank the reviewer for the comment. We compared the data distribution of relative and absolute abatements for three models (AIM, REMIND, and MESSAGE) (see Figure S3). The figure shows the results for relative abatement are more concentrated, supporting the use of relative abatements for our MAC functions.

Although there are large differences in the baselines of the models, many models assume a rising baseline scenario (especially for $CO_2$). Rising baseline scenarios counteract, at least to a certain extent, the increasing abatement level over time at the same carbon price.

We have the following relevant discussions in the manuscript (Sections 3.1 and 5):

*"Learning by doing" and "learning with time," which reduce the mitigation cost with abatement (endogenously) and time (exogenously), respectively (Hof et al., 2021), are not explicitly considered in our MAC curve approach, but are partially captured in our approach, which describes percentage reduction rates relative to rising baseline scenarios. For example, constant emission reductions in absolute terms can appear smaller over time in relative terms and thus become less costly in our approach.*

- *The overall good reproducibility of emIAM relies on the use of time-independent MAC curves for percentage emission reductions. The behaviors of IAMs that contain various time-dependent processes were generally well captured by the time-independent MAC curves. A plausible explanation is that the use of percentage abatement levels relative to rising baseline can offset the effect of lowering mitigation costs through learning.*

[Figure]

**Figure S3. MAC curves defined with relative and absolute abatement for three models.** The left three panels show the relationship between carbon price and relative abatement level, while the right three panels show the relationship between carbon price and absolute abatement. Black lines in the left three panels are MAC curves in percentage used in our study.

6.      The numbers in Fig. 10 and 11 are difficult to judge purely on their numerics. It would be useful to provide an example and focus on a number of key ingredients of emission pathways rather than pure correlations: how do the 2030 emissions differ, the netzero years, and the required negative emissions in overshoot scenarios? I guess that it is almost trivial to have a high correlation in general, because in all scenarios, emissions go down over time. Hence, to convince the reader, focusing on comparisons beyond mere correlation metrics would be useful.

[Response] We thank the reviewers for their comments. This suggestion is very useful, and we have incorporated this idea into our manuscript. We considered several new indicators that would be useful for comparing emission pathways of ACC2-emIAM and EMGAGE IAMs. Specifically, we considered the difference in carbon emissions between our reproduced results

and the original results for 2030, 2050, and 2100, as well as the difference in cumulative negative $CO_2$ emissions during the period 2020-2100, the difference in the year in which net zero of $CO_2$ and net zero of GHGs ($CO_2$ + $CH_4$ + $N_2O$) are achieved. We added Figure 10, which presents the results for the ECB scenarios without INDC from Test 4, as well as associated discussions as follows:

*Furthermore, we examine several selected features of the original and reproduced emission pathways from Test 4 (ECB scenarios without INDC only), such as $CO_2$ emissions in 2030, 2050, and 2100, cumulative negative $CO_2$ emissions from 2020 to 2100, the year to net zero for $CO_2$, and that for GHG. Figure 10a-c indicates that the reproducibility of $CO_2$ emissions for three different points in time varies across models and carbon budgets, but it is worth noting that ACC2-emIAM nearly consistently overestimates and underestimates 2030 $CO_2$ emissions from AIM and REMIND, respectively. Cumulative negative $CO_2$ emissions are negatively underestimated for COFFEE (Figure 10d), which is related to the general overestimation of 2100 $CO_2$ emissions for COFFEE (Figure 10c). The year to net zero for $CO_2$ tends to be overestimated (later than the original year) for REMIND with the carbon budget at or below 800 $GtCO_2$.*

[Figure]

**Figure 5. Differences in the pathway features between ENGAGE IAMs and ACC2-emIAM.**
Panels a to c show the difference in $CO_2$ emissions for 2030, 2050, and 2100, respectively. Panel

d shows the difference in cumulative negative $CO_2$ emissions. Panel e shows the difference in the year to net zero for $CO_2$. Panel f shows the difference in the year to net zero for GHGs (for $CO_2$, $CH_4$, and $N_2O$). Positive values indicate that the features in the original pathways (from ENGAGE IAMs) are larger than those in the reproduced pathways (from ACC2-emIAM), while negative values indicate the opposite. Gray boxes without black crosses indicate that the corresponding scenarios were not available in the ENGAGE Scenario Explorer, while those with black crosses indicate that the corresponding scenarios were available in the ENGAGE Scenario Explorer but not successfully reproduced by ACC2-emIAM (i.e., infeasible solutions).

7.        Perhaps more generally and related to aforementioned points: the MAC curve deductions themselves are interesting and a lot of insights can be obtained from them. However the analysis also reveals that "We do not provide specific recommendations on the appropriateness of the use of each MAC curve and leave the users to decide which MAC curves to apply" (p. 31), suggesting that the many differences between the MAC curves limit the universal applicability of emIAM. Potential users need to be guided better: which results are generalizable, what are the main uncertainties? A discussion section, looking at this question from a helicopter point-of-view may help in this respect, which is currently missing. This paper may be a first step in the direction of IAM-emulators, but then the authors are invited to write a bit more about what the next steps should be.

[Response] We thank the reviewer for the suggestion. Another reviewer expressed the same concern. In the revised manuscript, we have added several bullet points that give a high-level summary of the findings. We hope this generalization can better guide potential users.

- *The validation results for the two long-lived gases $CO_2$ and $N_2O$ did not strongly differ across all four Tests, even though for Tests 2 to 4, there is a difference in the model setup between the original IAMs (GHG aggregation using GWP100) and ACC2-emIAM (individual gas cycle modeling without using GWP100). On the other hand, the validation results for the short-lived gas $CH_4$ in Tests 2 to 4 were not as good as those in Test 1. Test 4, with the additional mid-century temperature target, yielded higher reproducibility for $CH_4$ than Tests 2 and 3.*
- *Overall, the global emissions were better reproduced than the regional emissions. $CO_2$ emission pathways were generally better reproduced than $CH_4$ and $N_2O$ pathways. Specific pathway features such as $CO_2$ emissions in 2030, 2050, and 2100, cumulative negative $CO_2$ emissions from 2020 to 2100, the year to net zero for $CO_2$, and that for GHG were reproduced to varying degrees across models and carbon budgets (Figure 10). While certain biases for some models were found for certain pathway features, as reported earlier, no general conclusions can be drawn.*

- *Some IAMs were more easily emulated than other IAMs, reflecting specific model features such as solution methods, technology assumptions, and abatement inertia. The emulator can usually reproduce the emission pathways of an IAM better if the model response to carbon price are well fitted with a MAC function.*
- *Certain data points were difficult to capture by MAC curves. In particular, PKB scenarios with low carbon budgets can give very large carbon prices in the near-term. Such data points tend to deviate from the trend of other data points and were manually removed from the MAC curve fitting where appropriate (Figure 1 and Table 1). Except for these "outliers," no discernible difference in the data trend was found between ECB scenarios and PKB scenarios, supporting the use of common MAC curves for ECB and PKB scenarios. Note also that certain data points from GET at high abatement levels do not follow the trend of other data points and were also removed from the MAC curve fitting where appropriate. We speculate that these data points are affected by the limit on CCS capacity assumed in GET.*
- *The overall good reproducibility of emIAM relies on the use of time-independent MAC curves for percentage emission reductions. The behaviors of IAMs that contain various time-dependent processes were generally well captured by the time-independent MAC curves. A plausible explanation is that the use of percentage abatement levels relative to rising baseline can offset the effect of lowering mitigation costs through learning.*

**Minor comments**

1.      The portfolios for GET are described only qualitatively. The choices (p. 5) even seem arbitrary – e.g., why did the authors use the numbers of 100% larger and 50% smaller bioenergy constraints in the respective portfolios?

[Response] This is just an arbitrary assumption used to illustrate our purpose.

2.      Unclear: in Section 4, also the regional MAC curves from emIAM are used, while on p. 6 a regional independence is assumed. What is it you are actually using in section 4?

[Response] Thank you for your comment. When we derived MAC curves for regions using the ENGAGE project, we assumed regions are independent. That is, we do not consider the correlation (or inter-dependency) between the abatement level of a region and that of another region. In Section 4, we used the regional MAC curves derived from the ENGAGE project. Here, the trade-off of abatement levels between regions with the least-cost emission pathways

can be seen. The model will decide which region should remove a certain level of gases considering its carbon price. Therefore, the work in section 4 does not conflict with that assumption.

3.      Could you elaborate a bit on Fig. 6 and where these points come from?

[Response] We thank the reviewer for the comment. As stated in the figure caption, different panels in Figure 6 present the relationship between the carbon price and the global energy-related $CO_2$ mitigation level under different technology portfolios. The original results from the GET model are shown in points. The process to calculate the abatement level can be seen in Section 3.1.

4.      Second derivative unit should be % / year^2 I guess, or are the numbers of the fractional order 1e-4?

[Response] We thank the reviewer for pointing out this issue. The second derivative unit is %/(year)^2, and this has been corrected in the text.

**References**

[revised manuscript text omitted]

---

## Referee Report (RR1)

**Title**: emIAM v1.0: an emulator for Integrated Assessment Models using
marginal abatement cost curves

**Authors**: Xiong et al.

**Summary of review**

I think the paper significantly improved, primarily with the revision of the figures and streamlining the content. Also, the new figure 10 and discussion of differences between estimates from the emulator and individual models, as well as the inclusion of confidence intervals and the generalization/discussion at the end improved the manuscript. Again, I do think the idea of this emulator is interesting and useful, but I still have a few concerns. I will leave it to the editor to weigh these concerns for a final decision and I am open to go with the consensus of the other reviewers.

**General comments**

1. Great that you included the confidence intervals in Figs. 2 and 6. However, how do these ranges propagate in the results of the emulator? In other words, could you also provide such ranges in the results of the emulator? Do they mean that the emulator output becomes very uncertain? I would propose, also to allow better comparison between the dots and the lines, that in Fig. 8 or 9 you omit some of the carbon budget levels (i.e., only focus on a few), and then also add confidence intervals of the emulator's output to get a feeling of how uncertain the output is.
2. In your response to my question on time variance of MACs and percentage abatement, you quote the text "*The behaviors of IAMs that contain various time-dependent processes were generally well captured by the time-independent MAC curves. A plausible explanation is that the use of percentage abatement levels relative to rising baseline can offset the effect of lowering mitigation costs through learning.*" I am not an expert on this particular matter, but could you elaborate on this? For example, has it been studied before to what extent, when merely looking at (percentage) abatement levels, time-invariant MACs are fine? I would expect that in the finer details (e.g., lifestyle changes, energy mix), this time invariance does not hold anymore. Also, see comment (3) below on the performance indicators you are using.
3. The correlation metrics in Tab. 5 and 6 should be changed. These are correlations over time, I believe? If so, it makes perfect sense that you get high correlations purely because in both cases the emission pathways drop. I would prefer to use root-mean-square-error, to actually get an idea of the error (in average or cumulative Gt $CO_2$, for example).

**Minor comments**

- I still think that the paper is lengthy. You may want to consider moving some of the text and figures to a supplementary file to streamline it even more.
- Why is only REMIND shown in Fig. 8? Perhaps better to show the averages across all models?
- Table 6 is unreadable. I propose to make a selection of things to show rather than everything.

---

## Author Response (AR2)

**#Topic editor**

The reviewers appreciate the progress made, but also point to needs for further improvement.

[Response] We thank all reviewers for the comments. They are helpful for improving the quality of the manuscript.

In particular, the justification for the use of time-independent MACs is not yet sufficient. For instance, the manuscript does not mention capital stock inertia as a key reason for time-dependence in abatement potential. Given the goal to apply the emIAM for long-term climate analysis, the authors should consider deriving separate MACs for short-term (for, e.g., 2030, when existing fossil capital stocks limit abatement potentials) and long-term (from 2050 onwards, beyond the lifetime of the bulk of current capital stocks).

[Response] We thank the editor for pointing out this issue. In light of the comment from the editor, we have tested time-dependent MAC curves using the data from AIM. We chose AIM because AIM gives markedly different data points in early decades compared to those in later decades, especially in scenarios with low carbon budgets. REMIND is used in the rest of the manuscript as an illustrative case, but we found that REMIND is not suitable for this exploration because the outliers mainly originate from peak-budget scenarios, which may have caused by constraints associated with the net-zero target.

Thus, for AIM, we introduced time-dependent MAC curves for $CO_2$, $CH_4$, and $N_2O$ before 2050, when the data points shift with time. We kept the original MAC curves after 2050 because the data points generally follow the same line after 2050. Specifically, we introduced a new parameter $e$ and an additional term into the MAC curves before 2050, as described by the following equation:

$$f(x) = \begin{cases} a * x^b + c * x^d, \ t \geq 2050 \\ a * \left(x * \left(e * (t - 2050)^2\right)\right)^b + c * \left(x * \left(e * (t - 2050)^2\right)\right)^d, \ t < 2050 \end{cases} \quad \text{(R1)}$$

We kept the values of the parameters $a, b, c$ and $d$, which have already been estimated. We estimated the parameter $e$ by considering all data points from 2020 to 2045, with no data points being excluded ($e$ is 0.00124 for $CO_2$, 0.00098 for $CH_4$, and 0.00076 for $N_2O$, respectively). This revised equation better captures both the near-term time-varying relationship and the longer-term stable relationship in the data points (Figure R1).

[Figure]

**Figure R1. Global total anthropogenic time-dependent MAC curves from the AIM model as an example.** The points are the data obtained from AIM in the ENGAGE Scenario Explorer and are shown with colors and markers as designated in the legend; the lines show the MAC curves derived for specific periods. The same color for the points and lines is for the same year. The time-independent MAC curve (black line) is derived from the approach described in the main text. This figure is included as Figure S241 in the Supplement.

Using these time-dependent MAC curves in ACC2-emIAM, we conducted Test 1 (i.e., the constraint on the cumulative emission budget of each gas) to examine the performance of the emulator. As discussed below, we also varied (retained or discarded) the assumptions on the upper bounds of the first and second derivatives of abatement changes (Table 2). Because these upper bounds strongly influence the near-term mitigation levels, we performed an analysis with and without such upper bounds. The results are shown in Table R1 and Figure R2.

**Table R1. Validation results of different MAC curve approaches for total anthropogenic CO₂, CH₄, and N₂O emissions derived from AIM.** All scenarios are shown here. For the definitions of the statistical indicators, see Section 4.4 of the main paper. This table is included as Table S8 in the Supplement.

| Upper bounds of 1st and 2nd derivatives | | Time-independent MAC curves | | Time-dependent MAC curves | |
|---|---|---|---|---|---|
| | | Included | Excluded | Included | Excluded |
| $r_P$ | $CO_2$ | 0.986 | 0.986 | 0.974 | 0.979 |
| | $CH_4$ | 0.962 | 0.962 | 0.94 | 0.939 |
| | $N_2O$ | 0.921 | 0.922 | 0.885 | 0.884 |
| $r_C$ | $CO_2$ | 0.981 | 0.981 | 0.964 | 0.964 |
| | $CH_4$ | 0.957 | 0.957 | 0.927 | 0.927 |
| | $N_2O$ | 0.916 | 0.918 | 0.874 | 0.873 |
| MAE | $CO_2$ | 2.595 | 2.512 | 3.628 | 3.609 |
| | $CH_4$ | 19.47 | 19.282 | 25.5 | 25.461 |
| | $N_2O$ | 0.501 | 0.486 | 0.624 | 0.619 |
| RMSE | $CO_2$ | 3.715 | 3.643 | 5.234 | 5.212 |
| | $CH_4$ | 27.411 | 27.014 | 36.713 | 36.755 |
| | $N_2O$ | 0.653 | 0.638 | 0.822 | 0.822 |

[Figure]

[Figure]

**Figure R2. Validation results for ACC2-emIAM with AIM MAC curves.** In the top panel, the points show the original REMIND emission pathways obtained from the ENGAGE Scenario Explorer; the lines show the emission pathways reproduced by ACC2-emIAM. The same color is used for each pair of original and reproduced pathways. The bottom panel shows the errors between the outputs of AIM and ACC2-emIAM, which use two types of MAC curves with/without the maximum first and second derivatives, respectively. For the sake of presentation, only the outcomes of the PKB scenarios w/o INDC are shown. This figure is included as Figure S242 in the Supplement.

The results indicate that the revised emulator using the time-dependent MAC curves is not superior to the original emulator with the time-independent MAC curves in terms of the reproducibility. With the time-dependent MAC curves, which penalize more the near-term mitigation, the near-term abatement became more limited for all three gases. This was expected, but in fact, the mitigation up to 2040 became too limited (relative to the output from AIM) because the time-dependent MAC curves are too high at low abatement levels for each period, which makes the near-term abatement more costly than the original model. The time-dependent MAC curves play the dominant role in shaping the near-term mitigation pathways. While the results from the original emulator with time-independent MAC curves showed a high sensitivity to the upper limits of the 1st and 2nd derivatives, those from the revised emulator with time-dependent MAC curves showed almost no sensitivity.

Overall, the time-dependent MAC curves did not improve the reproducibility of the IAM emulator in our example based on AIM. The results seemed puzzling to us at first because we expected an improved reproducibility with the time-dependent MAC curves. However, we

came to the realization that the overall performance of the emulator is determined by a complex interplay of various factors, including the MAC curves and the upper bounds of the first and second derivative limits. We agree that the time-dependent MAC curve approach can potentially improve the reproducibility of the IAM emulator (despite our rather negative results) and should be further pursued if the reproducibility of the IAM emulator is the main goal. However, we speculate that the actual advantages of using the time-dependent MAC curves can be model- and scenario-dependent, requiring further analysis.

Finally, we note that the use of the emulators for developing extended scenarios till 2300 is beyond the scope of the current paper (in fact, the paper does not say anything about that). As the editor is aware, we are indeed applying this method to extend emissions scenarios as part of Horizon Europe RESCUE and OptimESM projects. For these projects, we are further developing and fine tuning the emulator specifically to a newer and different version of REMIND-MAgPIE by taking into account the needs for the projects (e.g., explicit treatment of CDR technologies in the MAC curves to produce individual CDR pathways explicitly in our extended scenarios). On the other hand, our current paper aims to develop a more general approach. We intend to test a simple and common approach that could be applied to different models consistently and understand how well the simple MAC representation works for different IAMs under different scenarios.

Nevertheless, we once again thank the editor for suggesting the idea of time-dependent MAC curves. We have gained better understanding for some more complexity behind time-independent and time-dependent MAC curves. Since this issue of time-indepedency is important and also raised by the reviewers, we added a new subsection (4.5) to include the discussion above in the main text in a shorter form so that follow-up studies can be conducted potentially using different scenario data from different models.

Please also carefully consider the other remaining comments raised by the reviewers.

[Response] We further revised the manuscript based on the reviewers' comments. Please see below our point-by-point response. We hope that we have addressed all reviewers' comments and that our manuscript will be accepted for publication.

**#Reviwer 1**

The manuscript was substantially improved after the first-round revision and addressed most of the issues raised by the reviewers. I recommend that a version close to this one be accepted for publication. However, I still have one concern and three minor comments, as follows:

[Response] We appreciate the reviewer for the comments. They were very useful for improving the quality of our manuscript.

The authors simplified the MACCs by ignoring the temporal effects along the evaluation period, such that it can make the analysis more tractable and easier to communicate. In some cases, the assumptions necessary for a time-dependent MACC may introduce complexities that do not significantly improve the accuracy of the assessment. But it is not so clear to me in lines 686-687 (for the time-independent MAC curves): "A plausible explanation is that the use of percentage abatement levels relative to rising baseline can offset the effect of lowering mitigation costs through learning." It might relate to the learning costs; on the other hand, if there is a high degree of confidence that technology costs will not vary significantly over time, a time-independent MACC may be a reasonable assumption.

[Response] The issue discussed above is in some way explained by learning. As the reviewer pointed out, it is also true that if technology costs will not vary significantly over time, a time-independent MACC can be a reasonable assumption (under a stable baseline scenario). Given the reviewer's comment, we elaborated the text to the following:

*"A plausible explanation is that the use of percentage abatement levels relative to rising baseline can offset the effect of lowering mitigation costs through learning over time. In other words, the higher the baseline scenario is, the larger the absolute amount of emission reduction is (for the same percentage emission reduction). If technology costs will not vary significantly over time, a time-independent MAC curve can be a reasonable assumption (under a stable baseline scenario)."*

For minor comments:

1. In Figure 2, equations are not recommended to put in the caption text. Please move them to the methods.

[Response] We have moved these equations to a suitable place in Section 3 and changed to the following:

*"We also calculate the confidence intervals of the fitted curves using $\hat{y} \mp t_{\frac{\alpha}{2}} * S_{\varepsilon} *$*

$$\sqrt{1 + \frac{1}{n} + \frac{(x - \bar{x})^2}{\sum_{i=1}^{n} x_i^2 - \frac{(\sum_{i=1}^{n} x_i)^2}{n}}}$$ *(Thomson and Emery, 2014), where* $S_{\varepsilon} = \sqrt{\frac{\sum_{i=1}^{n} (y_i - \hat{y})^2}{n-2}}$, *n is the*

*sample size,* $t_{\frac{\alpha}{2}}$ *is the critical value of t-distribution,* $\bar{x}$ *is the mean of samples,* $\hat{y} = f(x)$, *and*

*$x_i, y_i$ are the original abatement level and carbon price result from the IAM, respectively."*

2. Please check the journal requirements for the layout of table. The table caption text is on top of the table, rather than under the table.

[Response] We have moved the caption text of all tables to the top of the tables.

3. For the summary of the results (lines 664-687), please re-order the lists (the estimation of MAC curves should come first, then the reproduction results), which can be consistent with the context of the manuscript.

[Response] We have re-ordered the first two summary points for the estimation of MAC curves first and the reproduction results later as below.

- *Certain data points were difficult to capture by MAC curves. In particular, PKB scenarios with low carbon budgets can give very large carbon prices in the near-term. Such data points tend to deviate from the trend of other data points and were manually removed from the MAC curve fitting where appropriate (Figure 1 and Table 1). Except for these "outliers," no discernible difference in the data trend was found between ECB scenarios and PKB scenarios, supporting the use of common MAC curves for ECB and PKB scenarios. Note also that certain data points from GET at high abatement levels do not follow the trend of other data points and were also removed from the MAC curve fitting where appropriate. We speculate that these data points are affected by the limit on CCS capacity assumed in GET.*
- *Some IAMs were more easily emulated than other IAMs, reflecting specific model features such as solution methods, technology assumptions, and abatement inertia. The emulator can usually reproduce the emission pathways of an IAM better if the model response to carbon price are well fitted with a MAC function.*

**#Reviewer 3**

I think the paper significantly improved, primarily with the revision of the figures and streamlining the content. Also, the new figure 10 and discussion of differences between estimates from the emulator and individual models, as well as the inclusion of confidence intervals and the generalization/discussion at the end improved the manuscript. Again, I do think the idea of this emulator is interesting and useful, but I still have a few concerns. I will

leave it to the editor to weigh these concerns for a final decision and I am open to go with the consensus of the other reviewers.

[Response] We appreciate the reviewer for the comments, which helped us to improve the quality of the manuscript.

General comments

1. Great that you included the confidence intervals in Figs. 2 and 6. However, how do these ranges propagate in the results of the emulator? In other words, could you also provide such ranges in the results of the emulator? Do they mean that the emulator output becomes very uncertain? I would propose, also to allow better comparison between the dots and the lines, that in Fig. 8 or 9 you omit some of the carbon budget levels (i.e., only focus on a few), and then also add confidence intervals of the emulator's output to get a feeling of how uncertain the output is.

[Response] We are thankful for the reviewer's comment. To estimate the uncertainty in the ACC2-emIAM due to the range of possible MAC curves derived from the IAMs, we utilized the confidence intervals at the 95% level for the fitted MAC curves, as represented as shaded bands in Figs. 2 and 6. In alignment with Fig. 8 in the main text, we take the REMIND MAC curves as an example. We, however, only report the reproducibility results based on the upper range of the MAC curve (95% confidence interval). We tried with the lower range of the MAC curve (95% confidence interval) as well, but we were not able to obtain reasonable results because of the negative segment of the lower MAC curve. The negative segment requires re-defining the problem as a new type of mathematical problem (a discontinuous nonlinear program (DNLP)), which either made it too complex to solve in our GAMS CONOPT3/4 computational environment or made the optimal solution unreliable (i.e., the solution becomes dependent on initial conditions).

[Figure]

**Figure R3. Validation results for ACC2-emIAM with mean and upper MAC curves from REMIND.** The points show the original emission pathways from REMIND obtained from the ENGAGE Scenario Explorer; the lines show the emission pathways reproduced from ACC2-emIAM by using the mean (the first column) and upper (the second column) MAC curves. The third column presents the uncertainty (shaded band within two emissions pathways) of ACC2-emIAM by using different MAC curves (only cases with the carbon budgets of 600, 1000, 2000, and 3000 GtCO$_2$ are shown here). The fourth column shows the errors from the reproduced scenarios (ACC2-emIAM) relative to the original scenarios (REMIND). Positive values indicate ACC2-emIAM gives higher estimates than REMIND and vice versa. The same color is used for each pair of original and reproduced pathways. For the sake of presentation, only the outcomes of the PKB scenarios without INDC are presented. This figure is added to Figure S243 in the Supplement.

The reproduced results using the default MAC curve and the upper MAC curve (95% confidence interval) are compared in Figure R3. We found that the use of the upper MAC curve weakens the emulator performance, as most clearly indicated by the abrupt emission declines for all gases, which did not occur in the original scenarios. This points to the need for assessing how to make use of the uncertainty in the MAC curve. It is also an issue of interpretation how the uncertainties in the MAC curves can propagate to the uncertainties in the reproduced scenarios gererated by new optimizations. The uncertainty propagation is different from more intuitive, forward uncertainty propagations, such as those along the cause-effect change of climate change: emissions → concentration → forcing → temperature change → impacts (e.g.,

Figure 8.27 of IPCC AR5 WGI Chapter 8). While analyses could be extended for other models and scenarios, such exploration falls beyond the primary focus of our paper.

We added the statement below in the main text and put Figure R3 as Figure S243 in Supplement.

"Uncertainty is reported in all MAC curves derived in this study. While such uncertainty is useful to indicate the confidence level of the MAC curve, it is not necessarily very obvious how to make use of the uncertainty range in reproducing scenarios from the IAM emulator (Figure S243)."

2. In your response to my question on time variance of MACs and percentage abatement, you quote the text "The behaviors of IAMs that contain various time-dependent processes were generally well captured by the time-independent MAC curves. A plausible explanation is that the use of percentage abatement levels relative to rising baseline can offset the effect of lowering mitigation costs through learning." I am not an expert on this particular matter, but could you elaborate on this? For example, has it been studied before to what extent, when merely looking at (percentage) abatement levels, time-invariant MACs are fine? I would expect that in the finer details (e.g., lifestyle changes, energy mix), this time invariance does not hold anymore. Also, see comment (3) below on the performance indicators you are using.

[Response] We appreciate the reviewer's thoughts. We agree with the reviewer's point that the time invariance may not hold for individual details. Our argument goes only at the aggregated level (global/regional total emissions from all sectors). While we hope to gain more insight at individual process levels, it is practically impossible for us to investigate how these are represented in each of the ten IAMs. (Note that we are just users of the data from the ENGAGE project. We are not part of the project, so our access to model details is limited.) This is also why we stick to the aggregated level and keep our scope to test how well our simple time-independent MAC representation works for different IAMs. Please also see the related comment from Reviewer #1 (first comment).

Time-independent MAC curves are not new as such and have been applied before. At the beginning of Section 3, we state that "*While MAC curves are more commonly time-dependent or for a specific point in time, time-independent MAC curves have also been used for long-term pathway calculations (Johansson et al., 2006; Tanaka and O'Neill, 2018; Tanaka et al., 2021) and short-term assessments (De Cara and Jayet, 2011)*". However, we argue that our study for the first time extensively applied the time-independent approach for capturing the behaviors of various IAMs. No other studies pushed this approach as far as our study did.

Given the reviewers' comment, we have modified and expanded the text. The revised text reads:

*"A plausible explanation is that the use of percentage abatement levels relative to rising baseline can offset the effect of lowering mitigation costs through learning over time. In other words, the higher the baseline scenario is, the larger the absolute amount of emission reduction is (for the same percentage emission reduction). If technology costs will not vary significantly over time, a time-independent MAC curve can be a reasonable assumption (under a stable baseline scenario)."*

3. The correlation metrics in Tab. 5 and 6 should be changed. These are correlations over time, I believe? If so, it makes perfect sense that you get high correlations purely because in both cases the emission pathways drop. I would prefer to use root-mean-square-error, to actually get an idea of the error (in average or cumulative Gt CO2, for example).

[Response] It is true that if two variables drop proportionally each time step, we only find perfect correlations. However, the data we are dealing with are not so idealistic and do deviate from this perfect setting, resulting in low correlations in certain cases. We maintain two indicators: i) ordinary Pearson's correlation coefficient $r_P$ and ii) Lin's concordance coefficient $r_C$. $r_P$ is a commonly used indicator that can be used to test the *strength of linear relationships*, which is a reference for our comparison. However, it is inappropriate for testing *agreement*, making $r_C$ a more appropriate choice for measuring agreement between two sequences.

Following the suggestion from the reviewer, we added two more indicators (i.e., the root-mean-square-error (*RMSE*) and mean-average-error (*MAE*) (see Figures S110-S128, S148-S166, S185-S202, and S222-S240 in the Supplement). We find that these two indicators provide added values as they capture the magnitude of the deviation. However, to avoid making the manuscript even longer, we keep the results of these two indicators in the Supplement.

Minor comments

• I still think that the paper is lengthy. You may want to consider moving some of the text and

figures to a supplementary file to streamline it even more.

[Response] We have thoroughly edited and streamlined the entire manuscript in response to the reviewer's comment for the previous review round. We have once again checked the manuscript to trim down the text for the current review round. However, due to the nature of this paper, we find it impossible to compress the text any more significantly. This is a methodological paper that requires a full description of our approach and underlying data, even

though the text becomes lengthy. We thought about moving some text to the Supplement, but we still prefer to keep them in the main paper and leave only additional figures in the Supplement (the text becomes not easily accessible for readers if it is put in the Supplement). It is also our observation that many other papers in Geoscientific Model Development are as long as or even longer than our manuscript. This is ultimately an editorial decision, but we hope that our current paper format is acceptable for the journal.

• Why is only REMIND shown in Fig. 8? Perhaps better to show the averages across all models?

[Response] We appreciate the reviewer for this suggestion. IAMs have very different baseline emissions and mitigation behaviors, which would make the averaged results challenging for interpretation. We take the results for REMIND only as an illustrative example for our verification processes. The results of other IAMs can be found in Figures S91-S109, S129-S147, S167-S184, and S203-S221 in the Supplement.

• Table 6 is unreadable. I propose to make a selection of things to show rather than everything.

[Response] We chose different colors to represent the intervals of specific values. That is, the darker the color, the higher the level of the reproducibility for ACC2-emIAM, thereby allowing the precise numerical values to become less central to the table's interpretation. Based on this visualization strategy, we consider the present format of the table to be suitable.

---

## Author Response (AR3)

LABORATOIRE DES SCIENCES DU CLIMAT ET DE L'ENVIRONNEMENT (LSCE)

**DR. KATSUMASA TANAKA**

Merisiers, bat 714, 91191 Gif-sur-Yvette, FRANCE

Phone: +33 (0)1 69 08 13 04

katsumasa.tanaka@lsce.ipsl.fr

31 July 2024

Dear Editor,

We thank the Editor for further considering our manuscript and providing additional feedback, in particular, to the issue of the time-dependency of marginal abatement cost (MAC) curves. Following the suggestions from the Editor, we revisited the analysis using time-dependent MAC curves.

The revised manuscript provides a more extended analysis using three IAMs (AIM, POLES, and WITCH), instead of the single IAM (AIM) used previously. The new analysis confirmed our earlier finding: the use of time-dependent MAC curves does not improve the reproducibility of emission scenarios, even though the time-dependent MAC curves provide a better fit to the price-quantity data generated from original IAMs than the time-independent MAC curves. Please see the revised manuscript for detailed discussions.

Overall, our revised analysis has led to a more comprehensive understanding of how our IAM emulator works and how various elements, such as MAC curves, carbon price pathways, and various model constraints, interact in generating least-cost emission pathways. In the attached document, we present the revised analysis using the time-dependent MAC curves, followed by our point-by-point responses to the Editor's comments. We then describe the changes made to the manuscript.

We believe that our revised manuscript has carefully addressed the Editor's remaining concerns and meets the high standards of *Geoscientific Model Development*. We look forward to the Editor's decision for publication.

Yours sincerely,

Katsumasa Tanaka and Weiwei Xiong, on behalf of the author team

**Deriving time-dependent MAC curves**

[revised manuscript text omitted]

remaining carbon budget to the IAM emulator as a constraint and calculated the least-cost pathway for $CO_2$. This approach is equivalent to Test 1 for $CO_2$ discussed earlier. We focus on this test because this is the most direct and simplest way to evaluate the performance of MAC curves. In this test, our emulator derives $CO_2$ emission pathways in the same way as the IAMs do (i.e., with the remaining carbon budget as the constraint for intertemporal optimization models and with the carbon price pathways (exogenously computed from the remaining carbon budget) as the constraint for recursive dynamic models). Other tests (Tests 2, 3, and 4) use the temperature target as the constraint, which is not used in the IAMs, although these tests are also useful for other purposes (e.g., to show how the MAC curve approach works in a climate-economy setting). Also note that the carbon cycle and climate modules in ACC2 are not used for Test 1 – Test 1 is only about emissions without considering their implications to concentrations and temperatures.

[revised manuscript text omitted]

**Our point-by-point responses to the Editor's comments**

I appreciate the author's effort in further revising the manuscript. Most of the remaining comments by the reviewers were addressed adequately.
However, I am not convinced by the discussion around time dependence, and have some further questions:

[Response]
We thank for the insightful feedback from the Editor. Our point-by-point responses to the Editor's comments are in blue text below.

• Fig R1: It is somewhat surprising that a fit with four (!) free parameters delivers such a poor fit, in particular for CO2, on seemingly nicely aligned data points. I would expect e.g. a simple polynomial fit like a*x^3 + b*x^2 + c*x + d to perform substantially better. (a,b,c can be restricted to positive values to ensure monotony). Have you tried such alternative fits?

[Response]
We now use a slightly more complex functional form, as described above in section "Deriving time-dependent MAC curves." Our revised time-dependent MAC curves have four free parameters (two free parameters for AIM), as opposed to just one free parameter in the previous version. The revised time-dependent MAC curves can better capture the original data from AIM, especially for $CO_2$ (compare Figures R1 and R1'). The revised time-dependent MAC curves also provide a good fit to the data from the other two IAMs (Figure R1).

The functional form of the time-dependent MAC curves is kept similar to that of the time-independent MAC curves. The parameters commonly used in both types of MAC curves take the same values as computed before. If we use a completely different functional form, it would introduce an additional factor in our comparison between the time-independent and time-dependent results, which can make our comparison less tractable. This point is also mentioned in section "Deriving time-dependent MAC curves."

• This poor fitting will affect of course the performance of the time-dependant emulators. I would expect that better fits affect the comparison of time-dependent vs. time-independent emulation substantially.

[Response]
This is a very important point, and we have given a lot of thought to it. Our initial hypothesis was also that the time-dependent MAC curves will substantially improve the emulation results; however, our results indicated otherwise. For this revision, we have refined the time-dependent MAC curves and tested with three IAMs (instead of just one IAM previously); however, the hypothesis still turned out to be negative. By thoroughly examining the results, including carbon price pathways, we now better understand why the time-dependent approach did not improve the emulation results. This is because the deviations from the

original IAM scenarios are primarily caused by the difference in carbon price pathways between the emulator and the IAMs. In other words, the potential benefit of using the time-dependent approach was not apparent due to the difference in carbon price pathways, most notably. See above for more detailed discussions.

• You only demonstrate the superiority of time-independent MACs for AIM but not for other models. Again, I find that unconvincing. You also state "REMIND is used in the rest of the manuscript as an illustrative case, but we found that REMIND is not suitable for this exploration because the outliers mainly originate from peak-budget scenarios, which may have caused by constraints associated with the net-zero target." There are no additional constraints in these scenarios other than the limitation of peak budget. If the problem realates to the INDC2030 fixing (i.e., delayed uniform climate policies) that would be understandable, but could be resolved by simply excluding these INDC2030 cases.

[Response]
As described above, we expanded the time-dependent analysis to three IAMs in the revised manuscript. To validate the scenario reproducibility, we focused on the end-of-century budget scenarios without INDC. In section "Reproducing the IAM scenarios with the time-dependent emulator: methods", we provide the rationales as below:

"We focus on the end-of-century budget scenarios without INDCs, among three other sets of scenarios (the peak budget scenarios with INDCs, peak budget scenarios without INDCs, and end-of-century budget scenarios with INDCs). This set of scenarios is most suitable for testing how well the MAC curves reproduce the original scenarios because this set of scenarios is free of constraints for net-zero emissions and INDC target levels, which cannot be captured by MAC curves."

• In respnse to the reviewer's comments on time-dependence, the authors added a paragraph pointing to learning as a key explanation. In fact, learning is not even represented in many IAMs. However, capital stock inertia is an even more important factor resulting in pathdependencies, e.g. via fossil carbon-lock in. Please add this aspect to the discussion

[Response]
We thank the Editor for pointing out this. We added "capital stock" to the discussion and further generalized the processes and factors that can cause inertia in IAMs. We revised the text at two places as follows (underlined text is the revised text):

"These barriers to rapid emission reductions and the associated costs could also be introduced by more complex functional forms internally in the MAC curves (Ha-Duong et al., 1997; Schwoon and Tol, 2006; De Cara and Jayet, 2011; Hof et al., 2021), but we applied such limits externally on the MAC curves. Processes and factors that can cause inertia in IAMs, including capital stock, growth rate constraints on technology expansion, availability of new technologies, learning by doing, and learning with time (Gambhir et al., 2019; Krey et al., 2019; Tong et al., 2019; Shiraki and Sugiyama, 2020), are not explicitly considered in our MAC curve approach, but are partially captured in our approach, which describes percentage reduction rates relative to rising baseline scenarios."

"Since most of the baseline scenarios are rising as noted above, the same amount of emission abatement in absolute terms can become smaller with time in percentage terms, which inadvertently but effectively captures the influences from time-dependent processes in IAMs.

Given the importance of near-term dynamics for any policy facing IAM research, I need to insist on clarifying and addressing these questions.

[Response]
We have addressed all of the Editor's points by using the refined formulation of the time-dependent MAC curves and the data from three IAMs. While the time-dependent MAC curves were shown to be more suited for capturing the relationship between the abatement level and the carbon price from the three IAMs than the time-independent MAC curves, we did not see an improvement in reproducing scenarios with the time-dependent MAC curves. As stated above, the potential benefit of using the time-dependent approach was not apparent due to other confounding factors, most notably carbon price pathways. Nevertheless, testing the time-dependent approach was useful in clarifying the dynamics of capturing the complex behavior of IAMs through the simplified emulation approach and in identifying issues for future improvements of the IAM emulator.

**Summary of the changes made in the manuscript**

The time-dependent MAC section (Section 4.5 of the previous version of our manuscript) has been replaced with a new Section 5 to present the new analysis above. Relevant figures and tables (Figures R1 to R3 and Tables R1 and R2) have been also added to the main text, instead of the Supplement, as the importance of this section has been elevated through the new analysis. We only slightly modified the text to make it suitable for the main text of the paper. For example, the text referring to changes from the previous version of our manuscript was not included in the main text. As shown in the manuscript with track changes, there are very minor changes in the figures and tables due to the update of color schemes and some corrections for the data presented, but these do not influence the results and discussion of this paper.

We further note that, through the additional analysis presented above, we realized the important role of carbon price pathways, which is also relevant to the rest of the analysis in our manuscript. However, we have decided to keep the remaining manuscript in the present form, so as not to further expand the already extensive manuscript. We will instead pursue this elsewhere, as part the ongoing study that aims to develop a new emulator of REMIND-MAgPIE and extended scenarios for the two Horizon Europe projects, as discussed in our previous review round.

New references

Gambhir, A., Butnar, I., Li, P.-H., Smith, P., and Strachan, N.: A Review of Criticisms of Integrated Assessment Models and Proposed Approaches to Address These, through the Lens of BECCS, Energies, 12, 1747, https://doi.org/10.3390/en12091747, 2019.

Krey, V., Guo, F., Kolp, P., Zhou, W., Schaeffer, R., Awasthy, A., Bertram, C., de Boer, H.-S., Fragkos, P., Fujimori, S., He, C., Iyer, G., Keramidas, K., Köberle, A. C., Oshiro, K., Reis, L. A., Shoai-Tehrani, B., Vishwanathan, S., Capros, P., Drouet, L., Edmonds, J. E., Garg, A., Gernaat, D. E. H. J., Jiang, K., Kannavou, M., Kitous, A., Kriegler, E., Luderer, G., Mathur, R., Muratori, M., Sano, F., and van Vuuren, D. P.: Looking under the hood: A comparison of techno-economic assumptions across national and global integrated assessment models, Energy, 172, 1254–1267, https://doi.org/10.1016/j.energy.2018.12.131, 2019.

Shiraki, H. and Sugiyama, M.: Back to the basic: toward improvement of technoeconomic representation in integrated assessment models, Clim. Change, 162, 13–24, https://doi.org/10.1007/s10584-020-02731-4, 2020.

Tong, D., Zhang, Q., Zheng, Y., Caldeira, K., Shearer, C., Hong, C., Qin, Y., and Davis, S. J.: Committed emissions from existing energy infrastructure jeopardize 1.5 °C climate target, Nature, 572, 373–377, https://doi.org/10.1038/s41586-019-1364-3, 2019.

---

## Author Response (AR4)

LABORATOIRE DES SCIENCES DU CLIMAT ET DE L'ENVIRONNEMENT (LSCE)

DR. KATSUMASA TANAKA
Merisiers, bat 714, 91191 Gif-sur-Yvette, FRANCE
Phone: +33 (0)1 69 08 13 04
katsumasa.tanaka@lsce.ipsl.fr

9 December 2024

Dear Editor,

We once again thank the Editor for further considering our manuscript and providing detailed and useful feedback, which prompted additional analyses as presented in our revised manuscript.

As suggested by the Editor, we further expanded the analysis by using "free-fitting" time-dependent MAC curves. We have confirmed that free-fitting time-dependent MAC curves can better fit to the relationship between the abatement level and the carbon price from IAMs than our previous time-dependent MAC curves (referred to as "transitional" time-dependent MAC curves in our revised manuscript).

We have also confirmed that the use of time-dependent MAC curves in our emulator can improve the reproducibility of the original IAM emission scenarios, compared to the use of time-independent MAC curves. At the same time, however, we found that this occurs only under certain conditions. Namely, if we prescribe carbon price pathways to the emulator (a new experiment presented in the revised manuscript), emission scenarios are better reproduced with time-dependent MAC curves than with time-independent MAC curves. On the other hand, if we endogenously optimize carbon price pathways for given carbon budgets in our emulator (our default approach), time-dependent MAC curves are no longer superior to time-independent MAC curves. We have examined this issue in detail and further confirmed that the difference in carbon price pathways between the emulator and the IAMs plays a major role here. We think that capturing carbon price pathways is a salient point for the future development of IAM emulators. Nevertheless, a notable advance over the previous manuscript is that we now have clear cases demonstrating that time-dependent MAC curves are better at reproducing scenarios, which we believe has helped improve the clarity of our paper.

Please see the attached document for our point-by-point responses to the Editor's comments. We believe that our revised manuscript has carefully addressed the Editor's remaining concerns and meets the high standards of *Geoscientific Model Development*. We look forward to the Editor's decision for publication.

Yours sincerely,

Katsumasa Tanaka and Weiwei Xiong, on behalf of the author team

--- Editor's comments

The authors put substantial into the revisions, adding to an already complex study.

However, some of the two key issues still are not fully satisfactorily addressed:

* The authors acknowledge that a time-dependent formulation of the MACs performs better than the time-independant one. Howerver, they argue, that when applied in the GET model, it does not make much of a difference. I don't find this convincing: Given that get has a different solution and/or optimization rationale (differences in foresight, different discount rates) between GET and the IAMs to be emulated, the MACCs but not the reproduction of emissions or carbon prices should be the relevant benchmark.

[Comment] We appreciate the Editor's comment, which prompted our new additional analysis. As stated in the cover letter, we found cases in which using time-dependent formulation of MAC curves can better reproduce emissions scenarios, but only under certain conditions. In light of this finding, we have substantially revised Section 5 as copied at the end of this document, which should address the Editor's comment above.

* I still maintain that constraining the time-dependent MAC to parameters derived from the time-independent leads to a poorer than necessary performance of the emulator. For instance, for t>2050, the emulator seems forced to the fit derived for the full period, instead of only 2050-2100. Why? For t<2050, the time-dependant emulator is formulated as a deviation of the time-independent fit. Fig 11 clearly shows that this formulation does not do very well in reproducing POLES and WITCH data points. The concluding comparison of time-dependent with time-independent emulation therefore seems biased against the time-dependent formulation. Have the authors tried to perform a free fit for each time step? I would expect this to result in a much better performance, and see it as a necessary basis for a meaningful discussion of time-dependent vs. time-independent.

[Comment] As stated in the cover letter, we have included the suggested formulation of the time-dependent MAC curves, which is referred to as "free-fitting" time-dependent MAC curves in the revised manuscript. Free-fitting time-dependent MAC curves indeed better capture the relationship between the abatement level and the carbon price from IAMs. Using free-fitting time-dependent MAC curves can also improve, as expected, the scenario reproducibility with prescribed carbon price pathways. For further details, please see Section 5 as copied at the end of this document.

* In the conclusions, the authors write "The behaviors of IAMs that contain various time-dependent processes were generally well captured by the time-independent MAC curves". What finding or test

is this based upon? At least regarding short-term emissions (until ~2050) I would need to be convinced that this holds. And these shorter time-scales matter for climate change policy and for overshoots. So it seems more accurate to condition this statement on the long-term bahavior. I would also ask the authors to add a caveat in the conclusions on the accuracy of the emulation for shorter timescales.

[Comment] We agree with the Editor that this statement needs to be made conditional. We have revised and expanded the text to the following. The new text also addresses the suggested point on the accuracy of the emulation for shorter timescales.

"The behaviors of IAMs that contain various time-dependent processes were generally well captured by the time-independent MAC curves in the second half of the century, although the goodness of fit varies considerably among IAMs. However, time-independent MAC curves can work only poorly on shorter timescales for many IAMs due to processes and factors that can cause inertia in IAMs, including capital stock, growth rate constraints on technology expansion, and availability of new technologies."

Furthermore, given the new insights from the additional analyses presented in the revised manuscript, we have revised the final bullet point in the Conclusions section to the following:

"For certain IAMs (AIM, POLES, and WITCH), time-dependent MAC curves provide a better fit to the price-quantity data generated from the original IAM than time-independent MAC curves. However, the use of time-dependent MAC curves improves the reproducibility of emission scenarios only when the equivalent carbon price pathway is prescribed to the emulator. When the carbon price pathway is endogenously optimized under the equivalent carbon budget in the emulator, it will differ from the carbon price pathway used for the IAMs. This difference in carbon prices can negate the benefit of using time-dependent MAC curves. The overall performance of the emulator is determined by a complex interplay of various factors, including the MAC curves, the upper bounds of the first and second derivative limits, and carbon price pathways. Reproducing carbon price pathways will be an important consideration for the future development of IAM emulators."

We have also checked through the entire manuscript once again. All other changes that were made in the manuscript (indicated in the manuscript with tracked changes) are very minor or editorial.

[revised manuscript text omitted]